# MSGNN: A Spectral Graph Neural Network Based on a Novel Magnetic Signed Laplacian

**Yixuan He**
University of Oxford
yixuan.he@stats.ox.ac.uk

**Michael Perlmutter**
University of California, Los Angeles
perlmutter@ucla.edu

**Gesine Reinert**
University of Oxford
and The Alan Turing Institute, London
reinert@stats.ox.ac.uk

**Mihai Cucuringu**
University of Oxford
and The Alan Turing Institute, London
mihai.cucuringu@stats.ox.ac.uk

## Abstract

Signed and directed networks are ubiquitous in real-world applications. However, there has been relatively little work proposing spectral graph neural networks (GNNs) for such networks. Here we introduce a signed directed Laplacian matrix, which we call the magnetic signed Laplacian, as a natural generalization of both the signed Laplacian on signed graphs and the magnetic Laplacian on directed graphs. We then use this matrix to construct a novel efficient spectral GNN architecture and conduct extensive experiments on both node clustering and link prediction tasks. In these experiments, we consider tasks related to signed information, tasks related to directional information, and tasks related to both signed and directional information. We demonstrate that our proposed spectral GNN is effective for incorporating both signed and directional information, and attains leading performance on a wide range of data sets. Additionally, we provide a novel synthetic network model, which we refer to as the Signed Directed Stochastic Block Model, and a number of novel real-world data sets based on lead-lag relationships in financial time series.

## 1 Introduction

Graph Neural Networks (GNNs) have emerged as a powerful tool for extracting information from graph-structured data and have achieved state-of-the-art performance on a variety of machine learning tasks. However, compared to research on constructing GNNs for unsigned and undirected graphs, and graphs with multiple types of edges, GNNs for graphs where the edges have a natural notion of sign, direction, or both, have received relatively little attention.

There is a demand for such tools because many important and interesting phenomena are naturally modeled as signed and/or directed graphs, i.e., graphs in which objects may have either positive or negative relationships, and/or in which such relationships are not necessarily symmetric [1]. For example, in the analysis of social networks, positive and negative edges could model friendship or enmity, and directional information could model the influence of one person on another [2, 3]. Signed/directed networks also arise when analyzing time-series data with lead-lag relationships [4], detecting influential groups in social networks [5], and computing rankings from pairwise comparisons [6]. Additionally, signed and directed networks are a natural model for group conflict analysis [7], modeling the interaction network of the agents during a rumor spreading process [8], and maximizing positive influence while formulating opinions [9].

In general, most GNNs are either spectral or spatial. Spatial methods typically define convolution on graphs as a localized aggregation whereas spectral methods rely on the eigen-decomposition of a suitable graph Laplacian. Our goal is to introduce a novel Laplacian and an associated GNN for

Y. He et al., MSGNN: A Spectral Graph Neural Network Based on a Novel Magnetic Signed Laplacian. *Proceedings of the First Learning on Graphs Conference (LoG 2022)*, PMLR 198, Virtual Event, December 9–12, 2022.

signed directed graphs. While several spatial GNNs exist, such as SDGNN [3], SiGAT [10], SNEA [11], and SSSNET [12] for signed (and possibly directed) networks, this is one of the first works to propose a spectral GNN for such networks. We devote special attention to the concurrent preprint SigMaNet [13] which also constructs a spectral GNN based on a different Laplacian.

A principal challenge in extending traditional spectral GNNs to this setting is to define a proper notion of the signed, directed graph Laplacian. Such a Laplacian should be positive semidefinite, have a bounded spectrum when properly normalized, and encode information about both the sign and direction of each edge. Here, we unify the magnetic Laplacian, which has been used in [14] to construct a GNN on an (unsigned) directed graph, with a signed Laplacian which has been used for a variety of data science tasks on (undirected) signed graphs [15–18]. Importantly, our proposed matrix, which we refer to as the *magnetic signed Laplacian*, reduces to either the magnetic Laplacian or the signed Laplacian when the graph is directed, but not signed, or signed, but not directed.

Although this magnetic signed Laplacian is fairly straightforward to obtain, it is novel and surprisingly powerful: We show that our proposed *Magnetic Signed GNN (MSGNN)* is effective for a variety of node clustering and link prediction tasks. Specifically, we consider several variations of the link prediction task, some of which prioritize signed information over directional information, some of which prioritize directional information over signed information, while others emphasize the method's ability to extract both signed and directional information simultaneously.

In addition to testing MSGNN on established data sets, we also devise a novel synthetic model which we call the *Signed Directed Stochastic Block Model (SDSBM)*, which generalizes both the (undirected) Signed Stochastic Block Model from [12] and the (unsigned) Directed Stochastic Block Model from [5]. Analogous to these previous models, our SDSBM can be defined by a meta-graph structure and additional parameters describing density and noise levels. We also introduce a number of signed directed networks for link prediction tasks using lead-lag relationships in real-world financial time series.

**Main Contributions.** The main contributions of our work are:

1. We devise a novel matrix called the magnetic signed Laplacian, which can naturally be applied to signed and directed networks. The magnetic signed Laplacian is Hermitian, positive semidefinite, and the eigenvalues of its normalized counterpart lie in $[0, 2]$. Our proposed Laplacian matrix and its counterpart reduce to existing Laplacians when the network is unsigned and/or undirected.

2. We propose an efficient spectral graph neural network architecture, MSGNN, based on this magnetic signed Laplacian, which attains leading performance on extensive node clustering and link prediction tasks, including novel tasks that consider edge sign and directionality jointly. To the best of our knowledge, this is the first work to evaluate GNNs on tasks that are related to both edge sign and directionality.[1]

3. We introduce a novel synthetic model for signed and directed networks, called Signed Directed Stochastic Block Model (SDSBM), and also contribute a number of new real-world data sets constructed from lead-lag relationships of financial time series data.

## 2 Related Work

In this section, we review related work constructing neural networks for directed graphs and signed graphs. We refer the reader to [1] for more background information.

Several works have aimed to define neural networks on directed graphs by constructing various directed graph Laplacians and defining convolution as multiplication in the associated eigenbasis. [19] defines a directed graph Laplacian by generalizing identities involving the undirected graph Laplacian and the stationary distribution of a random walk. [20] uses a similar idea, but with PageRank in place of a random walk. [21] constructs three different first- and second-order symmetric adjacency matrices and uses these adjacency matrices to define associated Laplacians. Similarly, [22] uses several different graph Laplacians based on various graph motifs.

Quite closely related to our work, [14] constructs a graph neural network using the magnetic Laplacian. Indeed, in the case where all edge weights are positive, our GNN exactly reduces to the one proposed

---

[1]Some previous work, such as [3], evaluates GNNs on signed and directed graphs. However, they focus on tasks where either only signed information is important, or where only directional information is important.

in [14]. Importantly, unlike the other directed graph Laplacians mentioned above, the magnetic Laplacian is a complex, Hermitian matrix rather than a real, symmetric matrix. We also note [5], which constructs a GNN for node clustering on directed graphs based on flow imbalance.

All of the above works are restricted to unsigned graphs, i.e., graphs with positive edge weights. However, there are also a number of neural networks introduced for signed (and possibly also directed) graphs, mostly focusing on the task of link sign prediction, i.e., predicting whether a link between two nodes will be positive or negative. SGCN by [23] is one of the first graph neural network methods to be applicable to signed networks, using an approach based on balance theory [24]. However, its design is mainly aimed at undirected graphs. SiGAT [10] utilizes a graph attention mechanism based on [25] to learn node embeddings for signed, directed graphs, using a novel motif-based GNN architecture based on balance theory and status theory [26]. Subsequently, SDGNN by [3] builds upon this work by increasing its efficiency and proposing a new objective function. In a similar vein, SNEA [11] proposes a signed graph neural network for link sign prediction based on a novel objective function. In a different line of work, [12] proposes SSSNET, a GNN not based on balance theory designed for semi-supervised node clustering in signed (and possibly directed) graphs. A concurrent preprint, SigMaNet [13], proposes a signed magnetic Laplacian to construct a spectral GNN.[2] Additionally, several GNNs [28–30] have been introduced for multi-relational graphs, i.e., graphs with different types of edges. In such networks, the number of learnable parameters typically increases linearly with the number of edge types. Signed graphs, at least if the graph is unweighted or the weighting function $w$ only takes finitely many values, can be thought of as special cases of multi-relational graphs. However, in the context of (possibly weighted) signed graphs, there is an implicit relationship between the different edge types, namely that a negative edge is interpreted as the opposite of a positive edge and that edges with large weights are deemed more important than edges with small weights. These relationships will allow us to construct a network with significantly fewer trainable parameters than if we were considering an arbitrary multi-relational graph.

## 3   Proposed Method

### 3.1   Problem Formulation

Let $\mathcal{G} = (\mathcal{V}, \mathcal{E}, w, \mathbf{X}_\mathcal{V})$ denote a signed, and possibly directed, weighted graph with node attributes, where $\mathcal{V}$ is the set of nodes (or vertices), $\mathcal{E}$ is the set of (directed) edges (or links), and $w : \mathcal{E} \to (-\infty, \infty) \setminus \{0\}$ is the weighting function. Let $\mathcal{E}^+ = \{e \in \mathcal{E} : w(e) > 0\}$ denote the set of positive edges and let $\mathcal{E}^- = \{e \in \mathcal{E} : w(e) < 0\}$ denote the set of negative edges so that $\mathcal{E} = \mathcal{E}^+ \cup \mathcal{E}^-$. Here, we do allow self-loops but not multiple edges; if $v_i, v_j \in \mathcal{V}$, there is at most one edge $e \in \mathcal{E}$ from $v_i$ to $v_j$. Let $n = |\mathcal{V}|$, and let $d_{\text{in}}$ be the number of attributes at each node, so that $\mathbf{X}_\mathcal{V}$ is an $n \times d_{\text{in}}$ matrix whose rows are the attributes of each node. We let $\mathbf{A} = (A_{ij})_{i,j \in \mathcal{V}}$ denote the weighted, signed adjacency matrix where $\mathbf{A}_{i,j} = w_{i,j}$ if $(v_i, v_j) \in \mathcal{E}$, and $\mathbf{A}_{i,j} = 0$ otherwise.

### 3.2   Magnetic Signed Laplacian

In this section, we define Hermitian matrices $\mathbf{L}_U^{(q)}$ and $\mathbf{L}_N^{(q)}$ which we refer to as the unnormalized and normalized magnetic signed Laplacian matrices, respectively. We first define a symmetrized adjacency matrix and an absolute degree matrix by

$$\tilde{\mathbf{A}}_{i,j} \coloneqq \frac{1}{2}(\mathbf{A}_{i,j} + \mathbf{A}_{j,i}), \ \ 1 \le i,j \le n, \ \ \tilde{\mathbf{D}}_{i,i} \coloneqq \frac{1}{2}\sum_{j=1}^{n}(|\mathbf{A}_{i,j}| + |\mathbf{A}_{j,i}|), \ \ 1 \le i \le n,$$

with $\tilde{\mathbf{D}}_{i,j} = 0$ for $i \ne j$. Importantly, the use of absolute values ensures that the entries of $\tilde{\mathbf{D}}$ are non-negative. Furthermore, it ensures that all $\tilde{\mathbf{D}}_{i,i}$ will be strictly positive if the graph is connected. This is in contrast to the construction in [13] which will give a node degree zero if it has an equal number of positive and negative neighbors (for unweighted networks). To capture directional information, we next define a phase matrix $\mathbf{\Theta}^{(q)}$ by $\mathbf{\Theta}_{i,j}^{(q)} \coloneqq 2\pi q(\mathbf{A}_{i,j} - \mathbf{A}_{j,i})$, where $q \in \mathbb{R}$ is the so-called "charge parameter." In our experiments, for simplicity, we set $q = 0$ when the task at hand is unrelated to directionality, or when the underlying graph is undirected, and we set

---

[2]After the initial submission of this work we became aware of a concurrent work [27] by another research group which developed a network similar to ours independently at the same time.

$q = q_0 := 1/[2 \max_{i,j}(\mathbf{A}_{i,j} - \mathbf{A}_{j,i})]$ (so that $\Theta^{(q)}$ has entries $\in [0, \pi]$) for all the other tasks (except in an ablation study on the role of $q$). With $\odot$ denoting elementwise multiplication, and $\mathtt{i}$ denoting the imaginary unit, we now construct a complex Hermitian matrix $\mathbf{H}^{(q)}$ by

$$\mathbf{H}^{(q)} := \tilde{\mathbf{A}} \odot \exp(\mathtt{i}\Theta^{(q)})$$

where $\exp(\mathtt{i}\Theta^{(q)})$ is defined elementwise by $\exp(\mathtt{i}\Theta^{(q)})_{i,j} := \exp(\mathtt{i}\Theta_{i,j}^{(q)})$.

Note that $\mathbf{H}^{(q)}$ is Hermitian, as $\tilde{\mathbf{A}}$ is symmetric and $\Theta^{(q)}$ is skew-symmetric. In particular, when $q = 0$, we have $\mathbf{H}^{(0)} = \tilde{\mathbf{A}}$. Therefore, setting $q = 0$ is equivalent to making the input graph symmetric and discarding directional information. In general, however, $\mathbf{H}^{(q)}$ captures information about a link's sign, through $\tilde{\mathbf{A}}$, and about its direction, through $\Theta^{(q)}$.

We observe that flipping the direction of an edge, i.e., replacing a positive or negative link from $v_i$ to $v_j$ with a link of the same sign from $v_j$ to $v_i$ corresponds to complex conjugation of $\mathbf{H}_{i,j}^{(q)}$ (assuming either that there is not already a link from $v_j$ to $v_i$ or that we also flip the direction of that link if there is one). We also note that if $q = 0.25$, $\mathbf{A}_{i,j} = \pm 1$, and $\mathbf{A}_{j,i} = 0$, we have

$$\mathbf{H}_{i,j}^{(0.25)} = \pm\frac{i}{2} = -\mathbf{H}_{j,i}^{(0.25)}.$$

Thus, a unit-weight edge from $v_i$ to $v_j$ is treated as the opposite of a unit-weight edge from $v_j$ to $v_i$.

Given $\mathbf{H}^{(q)}$, we next define the unnormalized magnetic signed Laplacian by

$$\mathbf{L}_U^{(q)} := \tilde{\mathbf{D}} - \mathbf{H}^{(q)} = \tilde{\mathbf{D}} - \tilde{\mathbf{A}} \odot \exp(\mathtt{i}\Theta^{(q)}), \tag{1}$$

and also define the normalized magnetic signed Laplacian by

$$\mathbf{L}_N^{(q)} := \mathbf{I} - \left(\tilde{\mathbf{D}}^{-1/2}\tilde{\mathbf{A}}\tilde{\mathbf{D}}^{-1/2}\right) \odot \exp(\mathtt{i}\Theta^{(q)}). \tag{2}$$

When the graph $\mathcal{G}$ is directed, but not signed, $\mathbf{L}_U^{(q)}$ and $\mathbf{L}_N^{(q)}$ reduce to the magnetic Laplacians utilized in works such as [14, 31, 32] and [33]. Similarly, when $\mathcal{G}$ is signed, but not directed, $\mathbf{L}_U^{(q)}$ and $\mathbf{L}_N^{(q)}$ reduce to the signed Laplacian matrices considered in e.g., [15, 18] and [34]. Additionally, when the graph is neither signed nor directed, they reduce to the standard normalized and unnormalized graph Laplacians [35]. The following theorems show that $\mathbf{L}_U^{(q)}$ and $\mathbf{L}_N^{(q)}$ satisfy properties analogous to the traditional graph Laplacians. The proofs are in Appendix A.

**Theorem 1.** *For any signed directed graph $\mathcal{G}$ defined in Sec. 3.1, $\forall q \in \mathbb{R}$, both the unnormalized magnetic signed Laplacian $\mathbf{L}_U^{(q)}$ and its normalized counterpart $\mathbf{L}_N^{(q)}$ are positive semidefinite.*

**Theorem 2.** *For any signed directed graph $\mathcal{G}$ defined in Sec. 3.1, $\forall q \in \mathbb{R}$, the eigenvalues of the normalized magnetic signed Laplacian $\mathbf{L}_N^{(q)}$ are contained in the interval $[0, 2]$.*

By construction, $\mathbf{L}_U^{(q)}$ and $\mathbf{L}_N^{(q)}$ are Hermitian, and Theorem 1 shows they are positive semidefinite. In particular, they are diagonalizable by an orthonormal basis of complex eigenvectors $\mathbf{u}_1, \ldots, \mathbf{u}_n$ associated to real, nonnegative eigenvalues $\lambda_1 \leq \ldots \leq \lambda_n = \lambda_{\max}$. Thus, similar to the traditional normalized Laplacian, we may factor $\mathbf{L}_N^{(q)} = \mathbf{U}\Lambda\mathbf{U}^\dagger$, where $\mathbf{U}$ is an $n \times n$ matrix whose $k$-th column is $\mathbf{u}_k$, for $1 \leq k \leq n$, $\Lambda$ is a diagonal matrix with $\Lambda_{k,k} = \lambda_k$, and $\mathbf{U}^\dagger$ is the conjugate transpose of $\mathbf{U}$. A similar formula holds for $\mathbf{L}_U^{(q)}$.

We conclude this subsection with a comparison to SigMaNet, proposed in the concurrent preprint [13]. SigMaNet also constructs a GNN based on a signed magnetic Laplacian, which is different from the magnetic signed Laplacian proposed here. The claimed advantage of SigMaNet is that it does not require the tuning of a charge parameter $q$ and is invariant to, e.g., doubling the weight of every edge. In our work, for the sake of simplicity, we usually set $q = 0.25$, except for when the graph is undirected (in which case we set $q = 0$). However, a user may choose to also tune $q$ through a standard cross-validation procedure as in [14]. Moreover, one can readily address the latter issue by normalizing the adjacency matrix via a preprocessing step (e.g., [36]). In contrast to our magnetic signed Laplacian, in the case where the graph is not signed but is weighted and directed, the matrix proposed in [13] does not reduce to the magnetic Laplacian considered in [14]. For example, denoting

the graph adjacency matrix by $\mathbf{A}$, consider the case where $0 < \mathbf{A}_{j,i} < \mathbf{A}_{i,j}$. Let $m = \frac{1}{2}(\mathbf{A}_{i,j} + \mathbf{A}_{j,i})$, $\delta = \mathbf{A}_{i,j} - \mathbf{A}_{j,i}$, and let $\mathtt{i}$ denote the imaginary unit. Then the $(i,j)$-th entry of the matrix $\mathbf{L}^\sigma$ proposed in [13] is given by $\mathbf{L}^\sigma_{i,j} = m\mathtt{i}$, whereas the corresponding entry of the unnormalized magnetic Laplacian is given by $(\mathbf{L}^{(q)}_U)_{i,j} = m\exp(2\pi\mathtt{i}q\delta)$. Moreover, while SigMaNet is in principle well-defined on signed and directed graphs, the experiments in [13] are restricted to tasks where only signed or directional information is important (but not both). In our experiments, we find that our proposed method outperforms SigMaNet on a variety of tasks on signed and/or directed networks. Moreover, we observe that the signed magnetic Laplacian $\mathbf{L}^\sigma$ proposed in [13] has an undesirable property when the graph is unweighted — a node is assigned to have degree zero if it has an equal number of positive and negative connections. Our proposed Laplacian does not suffer from this issue.

### 3.3 Spectral Convolution via the Magnetic Signed Laplacian

In this section, we show how to use a Hermitian, positive semidefinite matrix $\mathbf{L}$ such as the normalized or unnormalized magnetic signed Laplacian introduced in Sec. 3.2, to define convolution on a signed directed graph. This method is similar to the ones proposed for unsigned (possibly directed) graphs in, e.g., [37–39] and [14], but we provide details in order to keep our work reasonably self-contained.

Given $\mathbf{L}$, let $\mathbf{u}_1 \ldots, \mathbf{u}_n$ be an orthonormal basis of eigenvectors such that $\mathbf{L}\mathbf{u}_k = \lambda_k \mathbf{u}_k$, and let $\mathbf{U}$ be an $n \times n$ matrix whose $k$-th column is $\mathbf{u}_k$, for $1 \le k \le n$. For a signal $\mathbf{x} : \mathcal{V} \to \mathbb{C}$, we define its Fourier transform $\widehat{\mathbf{x}} \in \mathbb{C}^n$ by $\widehat{\mathbf{x}}(k) = \langle \mathbf{x}, \mathbf{u}_k \rangle := \mathbf{u}_k^\dagger \mathbf{x}$, and equivalently, $\widehat{\mathbf{x}} = \mathbf{U}^\dagger \mathbf{x}$. Since $\mathbf{U}$ is unitary, we readily obtain the Fourier inversion formula

$$\mathbf{x} = \mathbf{U}\widehat{\mathbf{x}} = \sum_{k=1}^n \widehat{\mathbf{x}}(k)\mathbf{u}_k\,. \tag{3}$$

Analogous to the well-known convolution theorem in Euclidean domains, we define the convolution of $\mathbf{x}$ with a filter $\mathbf{y}$ as multiplication in the Fourier domain, i.e., $\widehat{\mathbf{y} * \mathbf{x}}(k) = \widehat{\mathbf{y}}(k)\widehat{\mathbf{x}}(k)$. By (3), this implies $\mathbf{y} * \mathbf{x} = \mathbf{U}\mathrm{Diag}(\widehat{\mathbf{y}})\widehat{\mathbf{x}} = (\mathbf{U}\mathrm{Diag}(\widehat{\mathbf{y}})\mathbf{U}^\dagger)\mathbf{x}$, where $\mathrm{Diag}(\mathbf{z})$ denotes a diagonal matrix with the vector $\mathbf{z}$ on its diagonal. Therefore, we say that $\mathbf{Y}$ is a *generalized convolution matrix* if

$$\mathbf{Y} = \mathbf{U}\boldsymbol{\Sigma}\mathbf{U}^\dagger\,, \tag{4}$$

for a diagonal matrix $\boldsymbol{\Sigma}$. This is a natural generalization of the class of convolutions used in [40].

A main purpose of using a graph filter as in (4) is to reduce the number of parameters while maintaining permutation invariance. Some potential drawbacks exist when defining a convolution via (4). First, it requires one to compute the eigen-decomposition of $\mathbf{L}$ which is expensive for large graphs. Second, the number of trainable parameters equals the size of the graph (the number of nodes), rendering GNNs constructed via (4) prone to overfitting. To remedy these issues, we follow [38] (see also [37]) and observe that spectral convolution may also be implemented in the spatial domain via polynomials of $\mathbf{L}$ by setting $\boldsymbol{\Sigma}$ equal to a polynomial of $\boldsymbol{\Lambda}$. This reduces the number of trainable parameters from the size of the graph to the degree of the polynomial and also enhances robustness to perturbations [41]. As in [38], we let $\widetilde{\boldsymbol{\Lambda}} = \frac{2}{\lambda_{\max}}\boldsymbol{\Lambda} - \mathbf{I}$ denote the normalized eigenvalue matrix (with entries in $[-1, 1]$) and choose $\boldsymbol{\Sigma} = \sum_{k=0}^K \theta_k T_k(\widetilde{\boldsymbol{\Lambda}})$, for some $\theta_1, \ldots, \theta_k \in \mathbb{R}$ where for $0 \le k \le K$, $T_k$ is the Chebyshev polynomials defined by $T_0(x) = 1, T_1(x) = x$, and $T_k(x) = 2xT_{k-1}(x) + T_{k-2}(x)$ for $k \ge 2$. Since $\mathbf{U}$ is unitary, we have $(\mathbf{U}\widetilde{\boldsymbol{\Lambda}}\mathbf{U}^\dagger)^k = \mathbf{U}\widetilde{\boldsymbol{\Lambda}}^k\mathbf{U}^\dagger$, and thus, letting $\widetilde{\mathbf{L}} := \frac{2}{\lambda_{\max}}\mathbf{L} - \mathbf{I}$, we have

$$\mathbf{Y}\mathbf{x} = \mathbf{U}\sum_{k=0}^K \theta_k T_k(\widetilde{\boldsymbol{\Lambda}})\mathbf{U}^\dagger\mathbf{x} = \sum_{k=0}^K \theta_k T_k(\widetilde{\mathbf{L}})\mathbf{x}\,. \tag{5}$$

This is the class of convolutional filters we will use in our experiments. However, one could also imitate Sec. 3.1 on [14] to produce a class of filters based on [39] rather than [38].

It is important to note that $\widetilde{\mathbf{L}}$ is constructed so that, in (5), $(\mathbf{Y}\mathbf{x})_i$ depends on all nodes within $K$-hops from $v_i$ on the undirected, unsigned counterpart of $\mathcal{G}$, i.e. the graph whose adjacency matrix is given by $\mathbf{A}'_{i,j} = \frac{1}{2}(|\mathbf{A}_{i,j}| + |\mathbf{A}_{j,i}|)$. Therefore, this notion of convolution does not favor "outgoing neighbors" $\{v_j \in \mathcal{V} : (v_i, v_j) \in \mathcal{E}\}$ over "incoming neighbors" $\{v_j \in V : (v_j, v_i) \in \mathcal{E}\}$ (or vice versa). This is important since for a given node $v_i$, both sets may contain different, useful information. Furthermore, since the phase matrix $\boldsymbol{\Theta}^{(q)}$ encodes an outgoing edge and an incoming edge differently,

the filter matrix $\mathbf{Y}$ is also able to aggregate information from these two sets in different ways. Regarding computational complexity, we note that while the matrix $\exp(\mathrm{i}\boldsymbol{\Theta}^{(q)})$ is dense in theory, in practice, one only needs to compute a small fraction of its entries corresponding to the nonzero entries of $\tilde{\mathbf{A}}$ (which is sparse for most real-world data sets). Thus, the computational complexity of the convolution proposed here is equivalent to that of its undirected, unsigned counterparts.

### 3.4 The MSGNN architecture

We now define our network, MSGNN. Let $\mathbf{X}^{(0)}$ be an $n \times F_0$ input matrix with columns $\mathbf{x}_1^{(0)}, \ldots \mathbf{x}_{F_0}^{(0)}$, and $L$ denote the number of convolution layers. As in [14], we use a complex version of the Rectified Linear Unit defined by $\sigma(z) = z$, if $-\pi/2 \le \arg(z) < \pi/2$, and $\sigma(z) = 0$ otherwise, where $\arg(\cdot)$ is the complex argument of $z \in \mathbb{C}$. Let $F_\ell$ be the number of channels in the $\ell$-th layer. For $1 \le \ell \le L$, $1 \le i \le F_\ell$, and $1 \le j \le F_{\ell-1}$, let $\mathbf{Y}_{ij}^{(\ell)}$ be a convolution matrix defined by (4) or (5). Given the $(\ell-1)$-st layer hidden representation matrix $\mathbf{X}^{(\ell-1)}$, we define $\mathbf{X}^{(\ell)}$ columnwise by

$$\mathbf{x}_j^{(\ell)} = \sigma \left( \sum_{i=1}^{F_{\ell-1}} \mathbf{Y}_{ij}^{(\ell)} \mathbf{x}_i^{(\ell-1)} + \mathbf{b}_j^{(\ell)} \right), \tag{6}$$

where $\mathbf{b}_j^{(\ell)}$ is a bias vector with equal real and imaginary parts, $\mathrm{Real}(\mathbf{b}_j^{(\ell)}) = \mathrm{Imag}(\mathbf{b}_j^{(\ell)})$. In matrix form we write $\mathbf{X}^{(\ell)} = \mathbf{Z}^{(\ell)}\left(\mathbf{X}^{(\ell-1)}\right)$, where $\mathbf{Z}^{(\ell)}$ is a hidden layer of the form (6). In our experiments, we utilize convolutions of the form (5) with $\mathbf{L} = \mathbf{L}_N^{(q)}$ and set $K = 1$, in which case we obtain

$$\mathbf{X}^{(\ell)} = \sigma \left( \mathbf{X}^{(\ell-1)}\mathbf{W}_{\mathrm{self}}^{(\ell)} + \widetilde{\mathbf{L}}_N^{(q)}\mathbf{X}^{(\ell-1)}\mathbf{W}_{\mathrm{neigh}}^{(\ell)} + \mathbf{B}^{(\ell)} \right),$$

where $\mathbf{W}_{\mathrm{self}}^{(\ell)}$ and $\mathbf{W}_{\mathrm{neigh}}^{(\ell)}$ are learned weight matrices corresponding to the filter weights of different channels and $\mathbf{B}^{(\ell)} = (\mathbf{b}_1^{(\ell)}, \ldots, \mathbf{b}_{F_\ell}^{(\ell)})$. After the convolutional layers, we unwind the complex matrix $\mathbf{X}^{(L)}$ into a real-valued $n \times 2F_L$ matrix. For node clustering, we then apply a fully connected layer followed by the softmax function. By default, we set $L = 2$, in which case, our network is given by

$$\mathrm{softmax}(\mathrm{unwind}(\mathbf{Z}^{(2)}(\mathbf{Z}^{(1)}(\mathbf{X}^{(0)})))\mathbf{W}^{(3)}).$$

For link prediction, we apply the same method, except we concatenate rows corresponding to pairs of nodes after the unwind layer before applying the linear layer and softmax.

## 4 Experiments

### 4.1 Tasks and Evaluation Metrics

**Node Clustering.** In the node clustering task, one aims to partition the nodes of the graph into the disjoint union of $C$ sets $\mathcal{C}_0, \ldots, \mathcal{C}_{C-1}$. Typically in an unsigned, undirected network, one aims to choose the $\mathcal{C}_i$'s so that there are many links within each cluster and comparably few links between clusters, in which case nodes within each cluster are *similar* due to dense connections. In general, however, similarity could be defined differently [42]. In a signed graph, clusters can be formed by grouping together nodes with positive links and separating nodes with negative links (see [12]). In a directed graph, clusters can be determined by a directed flow on the network (see [5]). More generally, we can define clusters based on an underlying meta-graph, where meta-nodes, each of which corresponds to a cluster in the network, can be distinguished based on either signed or directional information (e.g., flow imbalance [5]). This general meta-graph idea motivates our introduction of a novel synthetic network model, which we will define in Sec. 4.2, driven by both link sign and directionality. All of our node clustering experiments are done in the semi-supervised setting, where one selects a fraction of the nodes in each cluster as seed nodes, with known cluster membership labels. In all of our node clustering tasks, we measure our performance using the Adjusted Rand Index (ARI) [43].

**Link Prediction.** On undirected, unsigned graphs, link prediction is simply the task of predicting whether or not there is a link between a pair of nodes. Here, we consider five different variations of the link prediction task for *signed and/or directed* networks. In our first task, link sign prediction (SP), one assumes that there is a link from $v_i$ to $v_j$ and aims to predict whether that link is positive or

negative, i.e., whether $(v_i, v_j) \in \mathcal{E}^+$ or $(v_i, v_j) \in \mathcal{E}^-$. Our second task, direction prediction (DP), one aims to predict whether $(v_i, v_j) \in \mathcal{E}$ or $(v_j, v_i) \in \mathcal{E}$ under the assumption that exactly one of these two conditions holds. We also consider three-, four-, and five-class prediction problems. In the three-class problem (3C), the possibilities are $(v_i, v_j) \in \mathcal{E}$, $(v_j, v_i) \in \mathcal{E}$, or that neither $(v_i, v_j)$ nor $(v_j, v_i)$ are in $\mathcal{E}$. For the four-class problem (4C), the possibilities are $(v_i, v_j) \in \mathcal{E}^+$, $(v_i, v_j) \in \mathcal{E}^-$, $(v_j, v_i) \in \mathcal{E}^+$, and $(v_j, v_i) \in \mathcal{E}^-$. For the five-class problem (5C), we also add in the possibility that neither $(v_i, v_j)$ nor $(v_j, v_i)$ are in $\mathcal{E}$. For all tasks, we evaluate the performance with classification accuracy. Notably, while (SP), (DP), and (3C) only require a method to be able to extract signed *or* directed information, the tasks (4C) and (5C) require it to be able to effectively process both sign *and* directional information. Also, we discard those edges that satisfy more than one condition in the possibilities for training and evaluation, but these edges are kept in the input network which is observed during training.

## 4.2 Synthetic Data for Node Clustering

**Established Synthetic Models.** We conduct experiments on the Signed Stochastic Block Models (SSBMs) and polarized SSBMs (POL-SSBMs) introduced in [12], which are signed but undirected. In the SSBM$(n, C, p, \rho, \eta)$ model, $n$ represents the number of nodes, $C$ is the number of clusters, $p$ is the probability that there is a link (of either sign) between two nodes, $\rho$ is the approximate ratio between the largest cluster size and the smallest cluster size, and $\eta$ is the probability that an edge will have the "wrong" sign, i.e., that an intra-cluster edge will be negative or an inter-cluster edge will be positive. POL-SSBM $(n, r, p, \rho, \eta, N)$ is a hierarchical variation of the SSBM model consisting of $r$ communities, each of which is itself an SSBM. We refer the reader to [12] for details of both models.

**A novel Synthetic Model: Signed Directed Stochastic Block Model (SDSBM).** Given a meta-graph adjacency matrix $\mathbf{F} = (\mathbf{F}_{k,l})_{k,l=0,\ldots,C-1}$, an edge sparsity level $p$, a number of nodes $n$, and a sign flip noise level parameter $0 \le \eta \le 0.5$, we defined a SDSBM model, denoted by SDSBM $(\mathbf{F}, n, p, \rho, \eta)$, as follows: (1) Assign block sizes $n_0 \le n_1 \le \cdots \le n_{C-1}$ based on a parameter $\rho \ge 1$, which approximately represents the ratio between the size of largest block and the size of the smallest block, using the same method as in [12]. (2) Assign each node to one of the $C$ blocks, so that each block $C_i$ has size $n_i$. (3) For nodes $v_i \in \mathcal{C}_k$, and $v_j \in \mathcal{C}_l$, independently sample an edge from $v_i$ to $v_j$ with probability $p \cdot |\mathbf{F}_{k,l}|$. Give this edge weight 1 if $F_{k,l} \ge 0$ and weight $-1$ if $F_{k,l} < 0$. (4) Flip the sign of all the edges in the generated graph with sign-flip probability $\eta$.

In our experiments, we use two sets of specific meta-graph structures $\{\mathbf{F}_1(\gamma)\}$, $\{\mathbf{F}_2(\gamma)\}$, with three and four clusters, respectively, where $0 \le \gamma \le 0.5$ is the directional noise level. Specifically, we are interested in SDSBM $(\mathbf{F}_1(\gamma), n, p, \rho, \eta)$ and SDSBM $(\mathbf{F}_2(\gamma), n, p, \rho, \eta)$ models with varying $\gamma$ where

$$\mathbf{F}_1(\gamma) = \begin{bmatrix} 0.5 & \gamma & -\gamma \\ 1-\gamma & 0.5 & -0.5 \\ -1+\gamma & -0.5 & 0.5 \end{bmatrix}, \mathbf{F}_2(\gamma) = \begin{bmatrix} 0.5 & \gamma & -\gamma & -\gamma \\ 1-\gamma & 0.5 & -0.5 & -\gamma \\ -1+\gamma & -0.5 & 0.5 & -\gamma \\ -1+\gamma & -1+\gamma & -1+\gamma & 0.5 \end{bmatrix}.$$

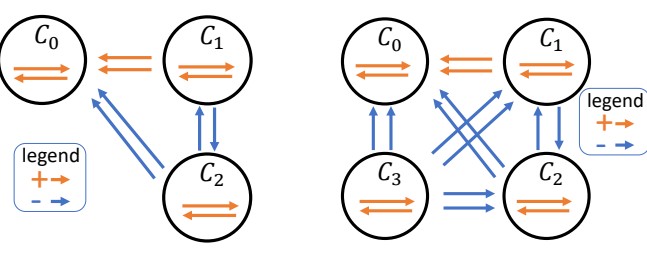

(a) Three-cluster model $\mathbf{F}_1$.      (b) Four-cluster model $\mathbf{F}_2$.

**Figure 1:** SDSBM illustration.

To better understand the above SDSBM models, toy examples are provided. We consider the following toy examples for our proposed synthetic data in Figure 1, which models groups of athletes and sports fans on social media. Here, signed, directed edges represent positive or negative mentions. In Figure 1(a), $\mathcal{C}_0$ are the players of a sports team, $\mathcal{C}_1$ is a group of their fans who typically say positive

things about the players, and $\mathcal{C}_2$ is a group of fans of a rival team, who typically say negative things about the players. Since they are fans of rival teams, the members of $\mathcal{C}_1$ and $\mathcal{C}_2$ both say negative things about each other. In general, fans mention the players more than players mention fans, which leads to net flow imbalance. In Figure 1(b), we add in $\mathcal{C}_3$, a group of fans of a third, less important team. This group dislikes the other two teams and disseminates negative content about $\mathcal{C}_0$, $\mathcal{C}_1$, and $\mathcal{C}_2$. However, since this third team is quite unimportant, no one comments anything back.

Notably, in both examples, as the expected edge density is identical both within and across clusters, discarding either signed or directional information will ruin the clustering structure. For instance, in both examples, if we discard directional information, then $\mathcal{C}_0$ will look identical to $\mathcal{C}_1$ in the resulting meta-graph. On the other hand, if we discard signed information, $\mathcal{C}_1$ will look identical to $\mathcal{C}_2$.

We also note that the SDSBM model proposed here is a generalization of both the SSBM model from [12] and the Directed Stochastic Block Model from [5] when we have suitable meta-graph structures.

### 4.3 Real-World Data for Link Prediction

**Standard Real-World Data Sets.** We consider four standard real-world signed and directed data sets. *BitCoin-Alpha* and *BitCoin-OTC* [2] describe bitcoin trading. *Slashdot* [44] is related to a technology news website, and *Epinions* [45] describes consumer reviews. These networks range in size from 3783 to 131580 nodes. Only *Slashdot* and *Epinions* are unweighted ($|w_{i,j}| = 1, \forall (v_i, v_j) \in \mathcal{E}$).

**Novel Financial Data Sets from Stock Returns.** Using financial time series data, we build signed directed networks where the weighted edges encode lead-lag relationships inherent in the financial market, for each year in the interval 2000-2020. The lead-lag matrices are built from time series of daily price returns.[3] We refer to these networks as our **Fi**ancial **L**ead-**L**ag (FiLL) data sets. For each year in the data set, we build a signed directed graph (*FiLL-pvCLCL*) based on the price return of 444 stocks at market close times on consecutive days. We also build another graph (*FiLL-OPCL*), based on the price return of 430 stocks from market open to close. The difference between 444 versus 430 stems from the non-availability of certain open and close prices on some days for certain stocks. The lead-lag metric that is captured by the entry $\mathbf{A}_{i,j}$ in each network encodes a measure that quantifies the extent to which stock $v_i$ leads stock $v_j$, and is obtained by computing the linear regression coefficient when regressing the time series (of length 245) of daily returns of stock $v_i$ against the lag-one version of the time series (of length 245) of the daily returns of stock $v_j$. Specifically, we use the beta coefficient of the corresponding simple linear regression, to serve as the one-day lead-lag metric. The resulting matrix is asymmetric and signed, rendering it amenable to a signed, directed network interpretation. The initial matrix is dense, with nonzero entries outside the main diagonal, since we do not consider the own auto-correlation of each stock. Note that an alternative approach to building the directed network could be based on Granger causality [46, 47], or other measures that quantify the lead-lag between a pair of time series, potentially while accounting for nonlinearity, such as second-order log signatures from rough paths theory as in [4].

Next, we sparsify each network, keeping only 20% of the edges with the largest magnitudes. We also report the average results across the all the yearly data sets (a total of 42 networks) where the data set is denoted by *FiLL (avg.)*. To facilitate future research using these data sets as benchmarks, both the dense lead-lag matrices and their sparsified counterparts have been made publicly available.

### 4.4 Experimental Results

We compare MSGNN against representative GNNs which are described in Section 2. The six methods we consider are (1) SGCN [23], (2) SDGNN [3], (3) SiGAT [10], (4) SNEA [11], (5) SSSNET [12], and (6) SigMaNet [13]. For all link prediction tasks, comparisons are carried out on all baselines; for the node clustering tasks, we only compare MSGNN against SSSNET and SigMaNet as adapting the other methods to this task is nontrivial. In all of our experiments, we use the normalized Magnetic signed Laplacian, $\mathbf{L}_N^q$, unless otherwise stated. Implementation details are provided in Appendix B, along with a runtime comparison which shows that MSGNN is generally the fastest method, see Table 2 in Appendix B. Extended results are in Appendix C and D. Code and preprocessed data are available at https://github.com/SherylHYX/MSGNN and have been included in the open-source library *PyTorch Geometric Signed Directed* [1].

---

[3]Raw CRSP data accessed through https://wrds-www.wharton.upenn.edu/.

### 4.4.1 Node Clustering

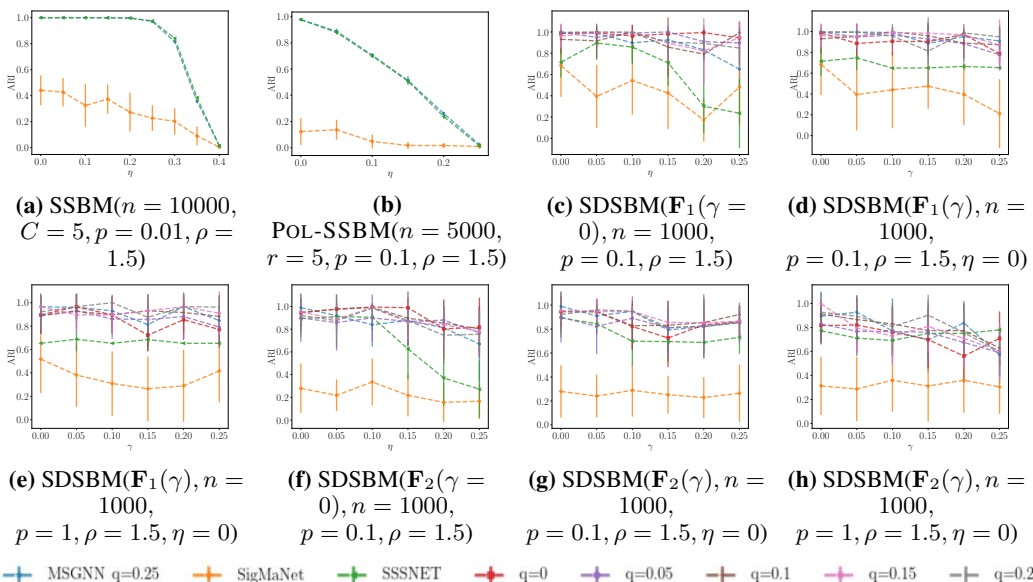

**Figure 2:** Node clustering test ARI comparison on synthetic data. Error bars indicate one standard error. Results are averaged over ten runs — five different networks, each with two distinct data splits.

Figure 2 compares the node clustering performance of MSGNN with two other signed GNNs on synthetic data, and against variants of MSGNN on SDSBMs. For signed, undirected networks $q$ does not has an effect, and hence we only report one MSGNN variant. Error bars are given by one standard error. We conclude that MSGNN outperforms SigMaNet on all data sets and is competitive with SSSNET.

On the majority of data sets, MSGNN achieves leading performance, whereas on some signed undirected networks (SSBM and Pol-SSBM) it is slightly outperformed by SSSNET. On these relatively small data sets, MSGNN and SSSNET have comparable runtime and are faster than SigMaNet. Comparing the MSGNN variants, we conclude that the directional information in these SDSBM models plays a vital role since MSGNN usually performs better with nonzero $q$.

### 4.4.2 Link Prediction

Our results for link prediction in Table 1 indicate that MSGNN is the top performing method, achieving the highest accuracy in **all** 25 cases. SNEA is among the best performing methods, but is the least efficient in speed due to its use of graph attention, see runtime comparison in Appendix B. Specifically, the "avg." results for the novel financial data sets first average the accuracy values across all individual networks (a total of 42 networks), then report the mean and standard deviation over the five runs. Results for individual *FiLL* networks are reported in Appendix D. Note that $\pm 0.0$ in the result tables indicates that the standard deviation is less than 0.05%.

### 4.4.3 Ablation Study and Discussion

Table 3 in Appendix C compares different variants of MSGNN on the link prediction tasks, with respect to (1) whether we set $q = 0$ or use a value $q = q_0 := 1/[2 \max_{i,j}(\mathbf{A}_{i,j} - \mathbf{A}_{j,i})]$) which strongly emphaszies directional information; (2) whether to include sign in input node features (if False, then only in- and out-degrees are computed like in [14] regardless of edge signs, otherwise features are constructed based on the positive and negative subgraphs separately); and (3) whether we take edge weights into account (if False, we view all edge weights as having magnitude one). Taking the standard errors into account, we find that incorporating directionality into the Laplacian matrix (i.e., having nonzero $q$) typically leads to slightly better performance in the directionality-related tasks (DP, 3C, 4C, 5C). Although, for *FiLL*, the $q = q_0$ values are within one standard deviation of the $q = 0$ ones for (T,T). The only example where $q = 0$ is clearly better is *Epinions* with (T,T) and

**Table 1:** Test accuracy (%) comparison the signed and directed link prediction tasks introduced in Sec. 4.1. The best is marked in **bold red** and the second best is marked in underline blue.

| Data Set | Link Task | SGCN | SDGNN | SiGAT | SNEA | SSSNET | SigMaNet | MSGNN |
|---|---|---|---|---|---|---|---|---|
| *BitCoin-Alpha* | SP | 64.7±0.9 | 64.5±1.1 | 62.9±0.9 | 64.1±1.3 | 67.4±1.1 | 47.8±3.9 | **71.3±1.2** |
| | DP | 60.4±1.7 | 61.5±1.0 | 61.9±1.9 | 60.9±1.7 | 68.1±2.3 | 49.4±3.1 | **72.5±1.5** |
| | 3C | 81.4±0.5 | 79.2±0.9 | 77.1±0.7 | 83.2±0.5 | 78.3±4.7 | 37.4±16.7 | **84.4±0.6** |
| | 4C | 51.1±0.8 | 52.5±1.1 | 49.3±0.7 | 52.4±1.8 | 54.3±2.9 | 20.6±6.3 | **58.5±0.7** |
| | 5C | 79.5±0.3 | 78.2±0.5 | 76.5±0.3 | 81.1±0.3 | 77.9±0.3 | 34.2±6.5 | **81.9±0.9** |
| *BitCoin-OTC* | SP | 65.6±0.9 | 65.3±1.2 | 62.8±1.3 | 67.7±0.5 | 70.1±1.2 | 50.0±2.3 | **73.0±1.4** |
| | DP | 63.8±1.2 | 63.2±1.5 | 64.0±2.0 | 65.3±1.2 | 69.6±1.0 | 48.4±4.9 | **71.8±1.1** |
| | 3C | 79.0±0.7 | 77.3±0.7 | 73.6±0.7 | 82.2±0.4 | 76.9±1.1 | 26.8±10.9 | **83.3±0.7** |
| | 4C | 51.5±0.4 | 55.3±0.8 | 51.2±1.8 | 56.9±0.7 | 57.0±2.0 | 23.3±7.4 | **59.8±0.7** |
| | 5C | 77.4±0.7 | 77.3±0.8 | 74.1±0.5 | 80.5±0.5 | 74.0±1.6 | 25.9±6.2 | **80.9±0.9** |
| *Slashdot* | SP | 74.7±0.5 | 74.1±0.7 | 64.0±1.3 | 70.6±1.0 | 86.6±2.2 | 57.9±5.3 | **92.4±0.2** |
| | DP | 74.8±0.9 | 74.2±1.4 | 62.8±0.9 | 71.1±1.1 | 87.8±1.0 | 53.0±4.0 | **93.1±0.1** |
| | 3C | 69.7±0.3 | 66.3±1.8 | 49.1±1.2 | 72.5±0.7 | 79.3±1.2 | 42.0±7.9 | **86.1±0.3** |
| | 4C | 63.2±0.3 | 64.0±0.7 | 53.4±0.2 | 60.5±0.6 | 72.7±0.6 | 25.7±8.9 | **78.2±0.3** |
| | 5C | 64.4±0.3 | 62.6±2.0 | 44.4±1.4 | 66.4±0.5 | 70.4±0.7 | 19.3±8.6 | **76.8±0.6** |
| *Epinions* | SP | 62.9±0.5 | 67.7±0.8 | 63.6±0.5 | 66.5±1.0 | 78.5±2.1 | 53.3±10.6 | **85.4±0.5** |
| | DP | 61.7±0.5 | 67.9±0.6 | 63.6±0.8 | 66.4±1.2 | 73.9±6.2 | 49.0±3.2 | **86.3±0.3** |
| | 3C | 70.3±0.8 | 73.2±0.8 | 52.3±1.3 | 72.8±0.2 | 72.7±2.0 | 30.5±8.3 | **83.1±0.5** |
| | 4C | 66.7±1.2 | 71.0±0.6 | 62.3±0.5 | 69.5±0.7 | 70.2±5.2 | 29.9±6.4 | **78.7±0.9** |
| | 5C | 73.5±0.8 | 76.6±0.7 | 52.9±0.7 | 74.2±0.1 | 70.3±4.6 | 22.1±6.1 | **80.5±0.5** |
| *FiLL (avg.)* | SP | 88.4±0.0 | 82.0±0.3 | 76.9±0.1 | 90.0±0.0 | 88.7±0.3 | 50.4±1.8 | **90.8±0.0** |
| | DP | 88.5±0.1 | 82.0±0.2 | 76.9±0.1 | 90.0±0.0 | 88.8±0.3 | 48.0±2.7 | **90.9±0.0** |
| | 3C | 63.0±0.1 | 59.3±0.0 | 55.3±0.1 | 64.3±0.1 | 62.2±0.3 | 33.7±1.3 | **66.1±0.1** |
| | 4C | 81.7±0.0 | 78.8±0.1 | 70.5±0.1 | 83.2±0.1 | 80.0±0.3 | 24.9±0.9 | **83.3±0.0** |
| | 5C | 63.8±0.0 | 61.1±0.1 | 55.5±0.1 | **64.8±0.1** | 60.4±0.4 | 19.8±1.1 | 64.8±0.1 |

task 4C. Hence, recommending $q = q_0$ is sensible. A further comparison of the role of $q$ is provided in Table 4 and shows that nonzero $q$ values usually deliver superior performance.

Moreover, signed features are in general helpful for tasks involving sign prediction. For constructing weighted features we see no significant difference in simply summing up entries in the adjacency matrix compared to summing the absolute values of the entries. Besides, calculating degrees regardless of edge weight magnitudes could be helpful for the first four data sets but not for *FiLL*. In the first four data sets the standard errors are much larger than the averages of the sums of the features, whereas in the *FiLL* data sets, the standard errors are much smaller than the average, see Table 17. Hence this feature may not show enough variability in the *FiLL* data sets to be very informative. Treating negative edge weights as the negation of positive ones is also not helpful (by not having separate degree features for the positive and negative subgraphs), which may explain why SigMaNet performs poorly in most scenarios due to its undesirable property. Surprisingly often, including only signed information but not weighted features does well. To conclude, constructing features based on the positive and negative subgraphs separately is helpful, and including directional information is generally beneficial.

More discussions on using instead the unnormalized Laplacian, or including more layers, and exploration of some properties of our proposed Laplacian, are provided in Appendix C.

## 5 Conclusion and Outlook

In this paper, we propose a spectral GNN based on a novel magnetic signed Laplacian matrix, introduce a novel synthetic network model and new real-world data sets, and conduct experiments on node clustering and link prediction tasks that are not restricted to considering either link sign or directionality alone. MSGNN performs as well or better than leading GNNs, while being considerably faster on real-world data sets. Future plans include investigating more properties of the proposed Laplacian, and an extension to temporal/dynamic graphs, where node features and/or edge information could evolve over time [48, 49]. We are also interested in extending our work to being able to encode nontrivial edge features, to develop objectives which explicitly handle heterogeneous edge densities throughout the graph, and to extend our approach to hypergraphs and other complex network structures.

## Author Contributions

Y.H.: Conceptualization, Data curation, Formal analysis, Funding acquisition, Investigation, Methodology, Project administration, Software, Visualization, Writing – original draft; M.P.: Conceptualization, Formal analysis, Methodology, Project administration, Writing – review & editing; G.R.: Conceptualization, Formal analysis, Funding acquisition, Methodology, Project administration, Supervision, Validation, Writing – review & editing; M.C.: Conceptualization, Data curation, Formal analysis, Funding acquisition, Methodology, Project administration, Resources, Supervision, Writing – review & editing.

## Acknowledgements

Y.H. is supported by a Clarendon scholarship. G.R. is supported in part by EPSRC grants EP/T018445/1, EP/W037211/1 and EP/R018472/1. M.C. acknowledges support from the EPSRC grants EP/N510129/1 and EP/W037211/1 at The Alan Turing Institute.

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

# A Proof of Theorems

## A.1 Proof of Theorem 1

*Proof.* Let $\mathbf{x} \in \mathbb{C}^n$. Since $\mathbf{L}_U^{(q)}$ is Hermitian, we have $\text{Imag}(\mathbf{x}^\dagger \mathbf{L}_U^{(q)} \mathbf{x}) = 0$. Next, we note by the triangle inequality that $\tilde{\mathbf{D}}_{i,i} = \frac{1}{2}\sum_{j=1}^n (|\mathbf{A}_{i,j}| + |\mathbf{A}_{j,i}|) \geq \sum_{j=1}^n |\tilde{\mathbf{A}}_{i,j}|$. Therefore, we may use the fact that $\tilde{\mathbf{A}}$ is symmetric to obtain

$$2\text{Real}\left(\mathbf{x}^\dagger \mathbf{L}_U^{(q)} \mathbf{x}\right)$$
$$=2\sum_{i=1}^n \tilde{\mathbf{D}}_{i,i}|\mathbf{x}(i)|^2 - 2\sum_{i,j=1}^n \tilde{\mathbf{A}}_{i,j}\mathbf{x}(i)\overline{\mathbf{x}(j)}\cos(\mathbf{\Theta}_{i,j}^{(q)})$$
$$\geq 2\sum_{i,j=1}^n |\tilde{\mathbf{A}}_{i,j}||\mathbf{x}(i)|^2 - 2\sum_{i,j=1}^n |\tilde{\mathbf{A}}_{i,j}||\mathbf{x}(i)||\mathbf{x}(j)|$$
$$=\sum_{i,j=1}^n |\tilde{\mathbf{A}}_{i,j}||\mathbf{x}_i|^2 + \sum_{i,j=1}^n |\tilde{\mathbf{A}}_{i,j}||\mathbf{x}_j|^2 - 2\sum_{i,j=1}^n |\tilde{\mathbf{A}}_{i,j}||\mathbf{x}_i||\mathbf{x}_j|$$
$$=\sum_{i,j=1}^n |\tilde{\mathbf{A}}_{i,j}|\left(|\mathbf{x}(i)| - |\mathbf{x}(j)|\right)^2 \geq 0.$$

Thus, $\mathbf{L}_U^{(q)}$ is positive semidefinite. For the normalized magnetic Laplacian, one may verify $\left(\tilde{\mathbf{D}}^{-1/2}\tilde{\mathbf{A}}\tilde{\mathbf{D}}^{-1/2}\right) \odot \exp(\mathrm{i}\mathbf{\Theta}^{(q)}) = \tilde{\mathbf{D}}^{-1/2}\left(\tilde{\mathbf{A}} \odot \exp(\mathrm{i}\mathbf{\Theta}^{(q)})\right)\tilde{\mathbf{D}}^{-1/2}$, and hence

$$\mathbf{L}_N^{(q)} = \tilde{\mathbf{D}}^{-1/2}\mathbf{L}_U^{(q)}\tilde{\mathbf{D}}^{-1/2}. \tag{7}$$

Thus, letting $\mathbf{y} = \tilde{\mathbf{D}}^{-1/2}\mathbf{x}$, the fact that $\tilde{\mathbf{D}}$ is diagonal implies

$$\mathbf{x}^\dagger \mathbf{L}_N^{(q)} \mathbf{x} = \mathbf{x}^\dagger \tilde{\mathbf{D}}^{-1/2}\mathbf{L}_U^{(q)}\tilde{\mathbf{D}}^{-1/2}\mathbf{x} = \mathbf{y}^\dagger \mathbf{L}_U^{(q)}\mathbf{y} \geq 0. \qquad \square$$

## A.2 Proof of Theorem 2

*Proof.* By Theorem 1, it suffices to show that the lead eigenvalue, $\lambda_n$, is less than or equal to 2. The Courant-Fischer theorem shows that

$$\lambda_n = \max_{\mathbf{x}\neq 0} \frac{\mathbf{x}^\dagger \mathbf{L}_N^{(q)} \mathbf{x}}{\mathbf{x}^\dagger \mathbf{x}}.$$

Therefore, using (7) and setting $\mathbf{y} = \tilde{\mathbf{D}}^{-1/2}\mathbf{x}$, we have

$$\lambda_n = \max_{\mathbf{x}\neq 0} \frac{\mathbf{x}^\dagger \tilde{\mathbf{D}}^{-1/2}\mathbf{L}_U^{(q)}\tilde{\mathbf{D}}^{-1/2}\mathbf{x}}{\mathbf{x}^\dagger \mathbf{x}} = \max_{\mathbf{y}\neq 0} \frac{\mathbf{y}^\dagger \mathbf{L}_U^{(q)}\mathbf{y}}{\mathbf{y}^\dagger \tilde{\mathbf{D}}\mathbf{y}}.$$

First, we observe that since $\tilde{\mathbf{D}}$ is diagonal, we have

$$\mathbf{y}^\dagger \tilde{\mathbf{D}}\mathbf{y} = \sum_{i,j=1}^n \tilde{\mathbf{D}}_{i,j}\mathbf{y}_i\overline{\mathbf{y}_j} = \sum_{i=1}^n \tilde{\mathbf{D}}_{i,i}|\mathbf{y}(i)|^2 = \frac{1}{2}\sum_{i,j=1}^n (|\mathbf{A}_{i,j}| + |\mathbf{A}_{j,i}|)|\mathbf{y}(i)|^2.$$

**Table 2:** Runtime (seconds) comparison on link tasks. The fastest is marked in **bold red** and the second fastest is marked in underline blue.

| Data Set | Task | SGCN | SDGNN | SiGAT | SNEA | SSSNET | SigMaNet | MSGNN |
|---|---|---|---|---|---|---|---|---|
| *BitCoin-Alpha* | SP | 352 | 124 | 277 | 438 | 59 | 151 | **29** |
| | DP | 328 | 196 | 432 | 498 | 78 | 152 | **37** |
| | 3C | 403 | 150 | 288 | 446 | 77 | 245 | **37** |
| | 4C | 385 | 133 | 293 | 471 | 57 | 143 | **36** |
| | 5C | 350 | 373 | 468 | 570 | 82 | 182 | **37** |
| *BitCoin-OTC* | SP | 340 | 140 | 397 | 584 | 68 | 222 | **30** |
| | DP | 471 | 243 | 426 | 941 | 80 | 155 | **38** |
| | 3C | 292 | 252 | 502 | 551 | 92 | 230 | **37** |
| | 4C | 347 | 143 | 487 | 607 | 68 | 209 | **37** |
| | 5C | 460 | 507 | 500 | 959 | 86 | 326 | **38** |
| *Slashdot* | SP | 4218 | 3282 | 1159 | 5792 | 342 | 779 | **227** |
| | DP | 4231 | 3129 | 1200 | 5773 | 311 | 817 | **222** |
| | 3C | 3686 | 6517 | 1117 | 6628 | **263** | 642 | 322 |
| | 4C | 4038 | 5296 | 948 | 7349 | **202** | 535 | 232 |
| | 5C | 4269 | 7394 | 904 | 8246 | 424 | 390 | **327** |
| *Epinions* | SP | 6436 | 4725 | 2527 | 8734 | **300** | 1323 | 370 |
| | DP | 6437 | 4605 | 2381 | 8662 | 404 | 1319 | **369** |
| | 3C | 6555 | 8746 | 2779 | 10536 | **471** | 885 | 510 |
| | 4C | 6466 | 6923 | 2483 | 10380 | **272** | 727 | 384 |
| | 5C | 7974 | 9310 | 2719 | 11780 | **460** | 551 | 517 |
| *FiLL (avg.)* | SP | 591 | 320 | 367 | 617 | 61 | 63 | **32** |
| | DP | 387 | 316 | 363 | 386 | 53 | 38 | **36** |
| | 3C | 542 | 471 | 298 | 657 | 79 | 114 | **43** |
| | 4C | 608 | 384 | 343 | 642 | 56 | 78 | **35** |
| | 5C | 318 | 534 | 266 | 521 | 63 | 66 | **44** |

The triangle inequality implies that $|\tilde{\mathbf{A}}_{i,j}| \leq \frac{1}{2}(|\mathbf{A}_{i,j}| + |\mathbf{A}_{j,i}|)$. Therefore, we may repeatedly expand the sums and interchange the roles of $i$ and $j$ to obtain

$$
\begin{aligned}
&\mathbf{y}^\dagger \mathbf{L}_U^{(q)} \mathbf{y} \\
\leq &\frac{1}{2} \sum_{i,j=1}^n (|\mathbf{A}_{i,j}| + |\mathbf{A}_{j,i}|)|\mathbf{y}(i)|^2 + \frac{1}{2} \sum_{i,j=1}^n (|\mathbf{A}_{i,j}| + |\mathbf{A}_{j,i}|)|\mathbf{y}(i)||\mathbf{y}(j)| \\
= &\frac{1}{2} \sum_{i,j=1}^n |\mathbf{A}_{i,j}|(|\mathbf{y}_i|^2 + |\mathbf{y}_j|^2 + 2|\mathbf{y}_i||\mathbf{y}_j|) \\
= &\frac{1}{2} \sum_{i,j=1}^n |\mathbf{A}_{i,j}|(|\mathbf{y}(i)| + |\mathbf{y}(j)|)^2 \leq \sum_{i,j=1}^n |\mathbf{A}_{i,j}|(|\mathbf{y}(i)|^2 + |\mathbf{y}(j)|^2) \\
= &\sum_{i,j=1}^n (|\mathbf{A}_{i,j}| + |\mathbf{A}_{j,i}|)|\mathbf{y}(i)|^2 = 2\mathbf{y}^\dagger \tilde{\mathbf{D}} \mathbf{y}. \qquad \square
\end{aligned}
$$

# B  Implementation Details

Experiments were conducted on two compute nodes, each with 8 Nvidia Tesla T4, 96 Intel Xeon Platinum 8259CL CPUs @ 2.50GHz and 378GB RAM. Table 2 reports total runtime after data preprocessing (in seconds) on link tasks for competing methods. We conclude that for large graphs SNEA is the least efficient method in terms of speed due to the attention mechanism employed, followed by SDGNN, which needs to count motifs. MSGNN is generally the fastest. Averaged results are reported with error bars representing one standard deviation in the figures, and plus/minus one standard deviation in the tables. For all the experiments, we use Adam [50] as our optimizer

with a learning rate of 0.01 and employ $\ell_2$ regularization with weight decay parameter $5 \cdot 10^{-4}$ to avoid overfitting. We use the open-source library PyTorch Geometric Signed Directed [1] for data loading, node and edge splitting, node feature preparation, and implementation of some baselines. For SSSNET [12], we use hidden size 16, 2 hops, and $\tau = 0.5$, and we adapt the architecture so that SSSNET is suitable for link prediction tasks. For SigMaNet [13], we use the code and parameter settings from `https://anonymous.4open.science/r/SigMaNet`. We set the number of layers to two for all methods.

## B.1 Node Clustering

We conduct semi-supervised node clustering, with 10% of all nodes from each cluster as test nodes, 10% as validation nodes to select the model, and the remaining 80% as training nodes (10% of which are seed nodes). For each of the synthetic models, we first generate five different networks, each with two different data splits, then conduct experiments on them and report average performance over these 10 runs. To train the GNNs on the signed undirected data sets (SSBMs and POL-SSBMs), we use the semi-supervised loss function $\mathcal{L}_1 = \mathcal{L}_{\text{PBNC}} + \gamma_s(\mathcal{L}_{\text{CE}} + \gamma_t \mathcal{L}_{\text{triplet}})$ as in [12], with the same hyperparameter setting $\gamma_s = 50, \gamma_t = 0.1$, where $\mathcal{L}_{\text{PBNC}}$ is the self-supervised probablistic balanced normalized cut loss function penalizing unexpected signs. For these signed undirected graphs, we use validation ARI for early stopping. For the SDSBMs, our loss function is the sum of $\mathcal{L}_1$ and the imbalance loss function $\mathcal{L}_{\text{vol\_sum}}^{\text{sort}}$ from [5] (absolute edge weights as input), i.e., $\mathcal{L}_2 = \mathcal{L}_{\text{vol\_sum}}^{\text{sort}} + \mathcal{L}_1$, and we use the self-supervised part of the validation loss ($\mathcal{L}_{\text{PBNC}} + \mathcal{L}_{\text{vol\_sum}}^{\text{sort}}$) for early stopping. We further restrict the GNNs to be trained on the subgraph induced by only the training nodes while applying the training loss function. For MSGNN on SDSBMs, we set $q = 0.25$ to emphasize directionality. The input node feature matrix $\mathbf{X}_{\mathcal{V}}$ for undirected signed networks in our experiments is generated by stacking the eigenvectors corresponding to the largest $C$ eigenvalues of a regularized version of the symmetrized adjacency matrix $\tilde{\mathbf{A}}$. For signed, directed networks, we calculate the in- and out-degrees based on both signs to obtain a four-dimensional feature vector for each node. We train all GNNs for the node clustering task for at most 1000 epochs with a 200-epoch early-stopping scheme.

## B.2 Link Prediction

We train all GNNs for each link prediction task for 300 epochs. We use the proposed loss functions from their original papers for SGCN [23], SNEA [11], SiGAT [10], and SDGNN [3], and we use cross-entropy loss $\mathcal{L}_{\text{CE}}$ for SigMaNet [13], SSSNET [12] and MSGNN. For all link prediction experiments, we sample 20% edges as test edges, and use the rest of the edges for training. Five splits were generated randomly for each input graph. We calculate the in- and out-degrees based on both signs from the observed input graph (removing test edges) to obtain a four-dimensional feature vector for each node for training SigMaNet [13], SSSNET [12], and MSGNN, and we use the default settings from [1] for SGCN [23], SNEA [11], SiGAT [10], and SDGNN [3].

## C Ablation Study and Discussion

Table 3 compares different variants of MSGNN on the link prediction tasks (Table 10 reports runtime), with respect to whether we use a traditional signed Laplacian that is initially designed for undirected networks (in which case we have $q = 0$) and a magnetic signed Laplacian, with $q = q_0 := 1/[2 \max_{i,j}(\mathbf{A}_{i,j} - \mathbf{A}_{j,i})]$, which strongly emphasizes directionality. We also assess whether to include sign in input node features, and whether we take edge weights into account. Note that, by default, the degree calculated given signed edge weights are net degrees, meaning that we sum the edge weights up without taking absolute values, which means that -1 and 1 would cancel out during calculation. The features that sum up absolute values of edge weights are denoted with T' in the table. Taking the standard error into account, we find no significant difference between the two options $T$ and $T'$ as weight features when signed features are not considered. We provide a toy example here to help better understand what the tuples mean. Consider a signed directed graph with adjacency matrix

$$\begin{bmatrix} 0 & 0.5 & -0.1 & 3 \\ -3 & 0 & 0 & 3 \\ 3 & 0 & 0 & 0 \\ 0 & -1 & 10 & 0 \end{bmatrix}$$

**Table 3:** Link prediction test performance (accuracy in percentage) comparison for variants of MSGNN. Each variant is denoted by a $q$ value and a 2-tuple: (whether to include signed features, whether to include weighted features), where "T" and "F" stand for "True" and "False", respectively. "T" for weighted features means simply summing up entries in the adjacency matrix while "T'" means summing the absolute values of the entries. The best is marked in **bold red** and the second best is marked in underline blue.

| q value → | | 0 | | | | | $q_0 := 1/[2\max_{i,j}(\mathbf{A}_{i,j} - \mathbf{A}_{j,i})]$ | | | | |
| Data Set | Link Task | (F, F) | (F, T) | (F, T') | (T, F) | (T, T) | (F, F) | (F, T) | (F, T') | (T, F) | (T, T) |
|---|---|---|---|---|---|---|---|---|---|---|---|
| *BitCoin-Alpha* | SP | 71.3±1.2 | 70.5±1.4 | 70.1±1.3 | **71.9±0.8** | 71.3±1.2 | 71.6±0.9 | 71.6±1.6 | 70.2±1.1 | 71.8±1.1 | 71.3±1.0 |
| | DP | 73.8±1.5 | 72.5±0.8 | 73.5±0.7 | 73.2±1.4 | 69.9±1.6 | 74.8±1.0 | 71.8±1.6 | 71.6±1.8 | **75.3±1.3** | 72.5±1.5 |
| | 3C | 85.6±0.3 | 84.4±0.5 | 84.3±0.6 | 85.7±0.5 | 84.3±0.5 | **85.8±0.9** | 84.2±0.9 | 84.4±0.6 | 85.6±0.6 | 84.4±0.6 |
| | 4C | 59.3±2.4 | 56.4±2.2 | 56.5±1.9 | 58.6±1.0 | 58.9±0.6 | 58.8±1.1 | 55.2±2.3 | 56.6±1.6 | **59.4±1.4** | 58.5±0.7 |
| | 5C | 83.8±0.4 | 82.3±0.6 | 81.3±1.7 | **83.9±0.6** | 82.1±0.5 | 83.3±0.6 | 82.0±0.5 | 81.9±0.5 | 83.2±0.3 | 81.9±0.9 |
| *BitCoin-OTC* | SP | 74.0±0.6 | 72.9±0.9 | 71.8±1.9 | 73.7±1.2 | 73.0±1.4 | 73.7±1.5 | 73.0±0.6 | 73.3±0.8 | **74.1±1.1** | 72.1±2.5 |
| | DP | 73.4±2.2 | 72.3±1.4 | 72.3±0.6 | 73.8±0.9 | 73.6±0.8 | 74.8±0.7 | 73.6±1.1 | 72.6±1.4 | **75.2±1.4** | 71.8±1.1 |
| | 3C | 83.7±0.8 | 82.9±0.7 | 82.4±1.0 | 84.2±0.4 | 83.0±0.6 | **85.0±0.5** | 83.9±0.4 | 83.3±1.0 | 84.8±0.9 | 83.3±0.7 |
| | 4C | 60.5±1.2 | 59.6±0.9 | 59.4±2.6 | 63.0±1.4 | 61.7±0.7 | 61.4±0.4 | 58.4±0.9 | 55.9±2.1 | **63.3±1.4** | 59.8±0.7 |
| | 5C | 81.1±0.6 | 79.8±0.6 | 79.0±0.9 | 82.4±0.3 | 80.0±1.3 | 82.4±0.8 | 78.0±2.5 | 80.0±0.7 | **82.6±0.7** | 80.9±0.9 |
| *Slashdot* | SP | 92.3±0.2 | 92.3±0.2 | 92.3±0.2 | 92.4±0.2 | 92.4±0.2 | 93.0±0.1 | 93.0±0.1 | 93.0±0.1 | **93.1±0.1** | **93.1±0.1** |
| | DP | 92.2±0.3 | 92.3±0.1 | 92.4±0.2 | 92.4±0.2 | 92.4±0.2 | **93.1±0.1** | **93.1±0.1** | **93.1±0.1** | 93.0±0.1 | **93.1±0.1** |
| | 3C | 86.0±0.1 | 85.8±0.2 | 86.0±0.1 | 85.9±0.3 | 85.7±0.4 | 86.2±0.2 | **86.3±0.4** | 86.2±0.3 | **86.3±0.2** | 86.1±0.3 |
| | 4C | 77.1±0.6 | 76.9±0.7 | 76.3±0.8 | 78.1±0.4 | 77.7±0.6 | 77.7±0.1 | 70.3±1.1 | 71.5±1.1 | 70.7±1.2 | **78.2±0.3** |
| | 5C | 77.1±0.7 | 77.4±0.4 | 77.7±0.3 | **78.1±0.4** | 77.8±0.3 | 72.8±0.3 | 73.1±0.3 | 72.8±0.3 | 77.5±0.6 | 76.8±0.6 |
| *Epinions* | SP | 85.2±0.7 | 85.5±0.4 | 85.2±0.7 | 85.9±0.3 | 85.4±0.5 | 86.4±0.1 | **86.6±0.1** | 86.4±0.1 | **86.6±0.2** | 86.3±0.1 |
| | DP | 85.1±0.8 | 85.3±0.7 | 85.4±0.4 | 85.0±0.5 | 85.3±0.7 | 86.2±0.2 | 86.1±0.7 | 86.1±0.4 | **86.3±0.1** | **86.3±0.3** |
| | 3C | 83.0±0.6 | 83.0±0.6 | 82.7±0.6 | 83.2±0.3 | 82.9±0.4 | 83.5±0.2 | 83.3±0.3 | **83.6±0.4** | **83.6±0.3** | 83.1±0.5 |
| | 4C | 78.3±0.6 | 78.2±1.6 | 79.1±1.2 | 79.7±1.2 | **80.0±1.0** | 76.6±1.3 | 76.7±1.5 | 76.5±1.5 | 79.3±0.5 | 78.7±0.9 |
| | 5C | 79.7±1.4 | 77.6±3.9 | 80.4±0.4 | 80.5±0.2 | **80.9±0.5** | 78.3±0.9 | 78.6±1.4 | 78.5±0.7 | 80.1±0.8 | 80.5±0.5 |
| *FiLL (avg.)* | SP | 74.0±0.1 | 75.5±0.1 | 75.5±0.1 | 75.1±0.1 | **76.2±0.1** | 73.8±0.1 | 75.5±0.0 | 75.5±0.1 | 75.1±0.1 | 76.1±0.1 |
| | DP | 74.0±0.1 | 75.3±0.1 | 75.3±0.3 | 75.0±0.1 | **75.9±0.1** | 73.5±0.3 | 75.2±0.0 | 75.2±0.1 | 75.0±0.0 | **75.9±0.1** |
| | 3C | 74.1±0.1 | 75.5±0.0 | 75.6±0.0 | 75.2±0.1 | **76.2±0.1** | 73.8±0.1 | 75.4±0.1 | 75.5±0.1 | 75.1±0.0 | 76.1±0.0 |
| | 4C | 74.0±0.2 | 75.6±0.1 | 75.6±0.0 | 75.2±0.1 | **76.3±0.1** | 74.0±0.3 | 75.5±0.1 | 75.6±0.1 | 75.1±0.1 | 76.2±0.1 |
| | 5C | 75.0±0.1 | 76.4±0.1 | 76.4±0.1 | 76.0±0.1 | **77.0±0.1** | 74.6±0.3 | 76.3±0.1 | 76.2±0.2 | 75.8±0.2 | **77.0±0.1** |

Corresponding to our tuple definition, we have, for the node corresponding to the first row and first column, $[2, 3]$ for (F,F), $[0, 3.4]$ for (F,T), $[6, 3.6]$ for (F,T'), $[1, 2, 1, 1]$ for (T,F), and $[3, 3.5, 3, 0.1]$ for (T,T).

To see the effect of using edge weights as input instead of treating all weights as having unit-magnitude, we report the main results as well as ablation study results correspondingly in Tables 5,6 and 7, and their runtimes are reported in Tables 12,13 and 14. We conclude that using unit-magnitude weights could be either beneficial or harmful depending on the data set and task at hand. However, in general, edge weights are important for *FiLL* but might not be helpful for the bitcoin data sets. Besides, we could draw similar conclusions as in Sec. 4.4.3 for the influence of input features as well as the $q$ value.

Table 8 compares tue performance of MSGNN with and without symmetric normalization (i.e., whether we use $\mathbf{L}_U^{(q)}$ or $\mathbf{L}_N^{(q)}$) and Table 9 shows how MSGNN's performance varies with the number of layers. The corresponding runtimes are reported in Tables 15 and 16, respectively. In general, we see that there are not significant differences in performance between normalizing and not normalizing; and in the vast majority of cases these differences are less than one standard deviation. We also note that in most cases, adding slightly more layers (from 2 to 4) yields a modest increase in performance. However, we again note that these increases in performance are typically quite small and often less than one standard deviation. When we further increase the number of layers (up to 10), performance of MSGNN begin to drop slightly. Overall, we do not see much evidence of severe oversmoothing. However, there does not seem to be significant advantages to very deep networks. Therefore, in our experiments, to be both effective in performance and efficient in computational expense we stick to two layers. (See Tables 15 and 16 for runtime comparison.)

It is of interest to explore the behavior of our proposed magnetic signed Laplacian matrix further. Hence here we assess its capability of separating clusters based on its top eigenvectors. Figures 3 and 4 show the top eigenvector of our proposed magnetic signed Laplacian matrix with $q = 0.25$ and symmetric normalization, where the x-axis denotes the real parts and the y-axis denotes the imaginary parts, for the two synthetic SDSBM models $\mathbf{F_1}(\gamma)$ and $\bar{\mathbf{F}}_{\mathbf{1}}(\gamma)$ from Subsection 4.2. Our Laplacian clearly picks up a signal using the top eigenvector, but does not detect all four clusters. Including information beyond the top eigenvector Figure 5 reports ARI values when we apply K-means algorithm to the stacked top eigenvectors for clustering. Specifically, for $\mathbf{F}_1$ with three clusters,

**Table 4:** Link prediction test performance (accuracy in percentage) comparison for MSGNN with different $q$ values (multiples of $q_0 := 1/[2 \max_{i,j}(\mathbf{A}_{i,j} - \mathbf{A}_{j,i})]$). The best is marked in **bold red** and the second best is marked in underline blue.

| Data Set | Link Task | $q=0$ | $q=0.2q_0$ | $q=0.4q_0$ | $q=0.6q_0$ | $q=0.8q_0$ | $q=q_0$ |
|---|---|---|---|---|---|---|---|
| *BitCoin-Alpha* | SP | 71.3±1.2 | 70.8±1.7 | 71.4±2.3 | **71.5±1.6** | 70.8±1.8 | 71.3±1.0 |
| | DP | 69.9±1.6 | 72.3±2.7 | 71.1±2.4 | 72.4±1.6 | 72.1±1.7 | **72.5±1.5** |
| | 3C | 84.3±0.5 | 84.6±0.4 | 84.5±0.9 | 84.4±0.5 | **84.8±0.9** | 84.4±0.6 |
| | 4C | 58.9±0.6 | 55.7±1.8 | 58.1±1.2 | 58.5±1.5 | **59.0±1.3** | 58.5±0.7 |
| | 5C | 82.1±0.5 | 82.3±0.7 | **82.8±0.4** | 82.4±0.5 | 81.9±1.1 | 81.9±0.9 |
| *BitCoin-OTC* | SP | **73.0±1.4** | 70.9±2.2 | 72.9±1.4 | 72.6±1.2 | 72.9±0.8 | 72.1±2.5 |
| | DP | **73.6±0.8** | **73.6±2.1** | 72.3±1.5 | 72.9±1.2 | 72.8±0.4 | 71.8±1.1 |
| | 3C | 83.0±0.6 | **83.7±0.5** | 83.5±0.4 | 83.1±0.6 | 83.6±0.6 | 83.3±0.7 |
| | 4C | **61.7±0.7** | 60.3±0.6 | 59.6±0.9 | 61.5±0.7 | 59.7±1.6 | 59.8±0.7 |
| | 5C | 80.0±1.3 | **81.1±1.3** | 80.9±0.8 | 81.0±0.8 | **81.1±0.6** | 80.9±0.9 |
| *Slashdot* | SP | 92.4±0.2 | 92.6±0.2 | 92.9±0.1 | 92.9±0.1 | 93.0±0.1 | **93.1±0.1** |
| | DP | 92.4±0.2 | 92.7±0.1 | 92.9±0.1 | 92.9±0.1 | **93.1±0.1** | **93.1±0.1** |
| | 3C | 85.7±0.4 | 86.0±0.2 | **86.3±0.2** | 86.2±0.2 | 86.2±0.2 | 86.1±0.3 |
| | 4C | 77.7±0.5 | 77.7±0.3 | 77.9±0.4 | 78.5±0.4 | **78.6±0.2** | 78.2±0.3 |
| | 5C | 77.8±0.3 | **78.2±0.3** | 78.1±0.1 | 77.6±0.5 | 77.6±0.4 | 76.8±0.6 |
| *Epinions* | SP | 85.4±0.5 | 85.7±0.3 | 86.0±0.4 | **86.5±0.1** | 86.2±0.1 | 86.3±0.1 |
| | DP | 85.3±0.7 | 85.9±0.3 | 86.2±0.1 | 86.2±0.1 | **86.5±0.2** | 86.3±0.3 |
| | 3C | 82.9±0.4 | 83.5±0.2 | **83.6±0.3** | 83.5±0.2 | 83.2±0.3 | 83.1±0.5 |
| | 4C | 80.0±1.0 | 80.8±0.5 | **81.1±0.5** | 80.1±0.8 | 79.7±0.7 | 78.7±0.9 |
| | 5C | 80.9±0.5 | 80.7±0.2 | **81.2±0.4** | 80.3±0.6 | 80.8±0.6 | 80.5±0.5 |
| *FiLL (avg.)* | SP | **76.2±0.1** | **76.2±0.1** | **76.2±0.1** | 76.1±0.0 | 76.1±0.0 | 76.1±0.1 |
| | DP | **75.9±0.1** | **75.9±0.1** | **75.9±0.1** | **75.9±0.0** | **75.9±0.0** | **75.9±0.1** |
| | 3C | **76.2±0.1** | **76.2±0.0** | **76.2±0.0** | 76.1±0.0 | 76.1±0.1 | 76.1±0.0 |
| | 4C | **76.3±0.1** | **76.3±0.1** | **76.3±0.0** | **76.3±0.0** | 76.2±0.0 | 76.2±0.1 |
| | 5C | **77.0±0.1** | **77.0±0.1** | **77.0±0.1** | **77.0±0.1** | **77.0±0.1** | **77.0±0.1** |

**Table 5:** Test accuracy (%) comparison the signed and directed link prediction tasks introduced in Sec. 4.1 where all networks are treated as unweighted. The best is marked in **bold red** and the second best is marked in underline blue.

| Data Set | Link Task | SGCN | SDGNN | SiGAT | SNEA | SSSNET | SigMaNet | MSGNN |
|---|---|---|---|---|---|---|---|---|
| *BitCoin-Alpha* | SP | 64.7±0.9 | 64.5±1.1 | 62.9±0.9 | 64.1±1.3 | 64.3±2.9 | 50.2±0.9 | **70.3±1.5** |
| | DP | 60.4±1.7 | 61.5±1.0 | 61.9±1.9 | 60.9±1.7 | 71.8±1.3 | 51.9±5.6 | **74.7±1.5** |
| | 3C | 81.4±0.5 | 79.2±0.9 | 77.1±0.7 | 83.2±0.5 | 79.4±1.3 | 34.9±17.5 | **86.1±0.5** |
| | 4C | 51.1±0.8 | 52.5±1.1 | 49.3±0.7 | 52.4±1.8 | 56.3±1.4 | 28.6±7.6 | **59.5±2.2** |
| | 5C | 79.5±0.3 | 78.2±0.5 | 76.5±0.3 | 81.1±0.3 | 78.8±0.8 | 25.8±17.0 | **83.8±0.8** |
| *BitCoin-OTC* | SP | 65.6±0.9 | 65.3±1.2 | 62.8±1.3 | 67.7±0.5 | 68.3±2.5 | 50.4±5.4 | **73.7±1.3** |
| | DP | 63.8±1.2 | 63.2±1.5 | 64.0±2.0 | 65.3±1.2 | 70.4±1.7 | 47.3±4.2 | **75.6±1.0** |
| | 3C | 79.0±0.7 | 77.3±0.7 | 73.6±0.7 | 82.2±0.4 | 78.0±0.5 | 35.3±15.3 | **85.3±0.4** |
| | 4C | 51.5±0.4 | 55.3±0.8 | 51.2±1.8 | 56.9±0.7 | 60.4±0.9 | 24.3±6.6 | **62.8±1.0** |
| | 5C | 77.4±0.7 | 77.3±0.8 | 74.1±0.5 | 80.5±0.5 | 76.8±0.5 | 18.9±11.1 | **83.0±0.9** |
| *FiLL (avg.)* | SP | 88.4±0.0 | 82.0±0.3 | 76.9±0.1 | 90.0±0.0 | 88.7±0.2 | 51.1±0.7 | **90.8±0.0** |
| | DP | 88.5±0.1 | 82.0±0.2 | 76.9±0.1 | 90.0±0.0 | 86.9±0.6 | 50.7±1.1 | **90.9±0.0** |
| | 3C | 63.0±0.1 | 59.3±0.0 | 55.3±0.1 | 64.3±0.1 | 57.3±0.4 | 34.1±0.4 | **64.6±0.1** |
| | 4C | 81.7±0.0 | 78.8±0.1 | 70.5±0.1 | **83.2±0.1** | 76.8±0.1 | 25.5±1.3 | 82.1±0.1 |
| | 5C | 63.8±0.0 | 61.1±0.1 | 55.5±0.1 | **64.8±0.1** | 55.8±0.5 | 20.5±0.7 | 62.3±0.2 |

**Table 6:** Link prediction test performance (accuracy in percentage) comparison for variants of MSGNN where input networks are treated as unweighted. Each variant is denoted by a $q$ value and a 2-tuple: (whether to include signed features, whether to include weighted features), where "T" and "F" stand for "True" and "False", respectively. "T" for weighted features means simply summing up entries in the adjacency matrix while "T'" means summing the absolute values of the entries. The best is marked in **bold red** and the second best is marked in underline blue.

| $q$ value Data Set | Link Task | 0 (F, F) | (F, T) | (F, T') | (T, F) | (T, T) | $q_0 := 1/[2\max_{i,j}(\mathbf{A}_{i,j} - \mathbf{A}_{j,i})]$ (F, F) | (F, T) | (F, T') | (T, F) | (T, T) |
|---|---|---|---|---|---|---|---|---|---|---|---|
| *BitCoin-Alpha* | SP | 72.0±0.9 | 70.2±1.1 | 70.9±1.4 | 72.1±0.4 | 70.3±1.5 | 72.0±1.2 | 70.4±0.6 | 70.2±2.0 | **72.2±1.2** | 70.9±1.7 |
| | DP | 73.9±0.5 | 73.9±1.0 | 72.8±2.3 | 74.1±1.1 | 74.5±1.5 | 73.9±1.2 | 73.8±1.0 | 73.3±1.1 | 73.6±2.4 | **74.7±1.5** |
| | 3C | 85.4±0.5 | 85.4±0.3 | 85.2±0.8 | 85.5±0.5 | 85.7±0.3 | 85.9±0.4 | 85.9±0.5 | 85.9±0.5 | 86.0±0.4 | **86.1±0.5** |
| | 4C | 57.9±1.9 | 58.8±1.5 | 57.7±1.2 | 58.9±1.1 | 58.4±1.8 | 53.6±1.7 | 54.6±2.2 | 55.2±1.8 | **59.6±2.2** | 59.5±2.2 |
| | 5C | 83.4±0.3 | 83.1±0.3 | 83.5±0.5 | **84.2±0.5** | 83.7±0.5 | 82.6±0.6 | 82.8±0.5 | 82.8±0.7 | 83.7±0.4 | 83.8±0.8 |
| *BitCoin-OTC* | SP | 74.5±1.0 | 71.7±2.3 | 73.8±1.0 | 74.1±1.0 | 73.7±1.3 | 74.2±1.1 | 73.4±0.8 | 73.1±0.9 | **74.9±1.0** | 73.5±0.6 |
| | DP | 75.0±0.5 | 75.2±1.8 | 74.7±1.1 | 75.2±1.4 | 74.8±2.1 | 75.6±1.4 | 75.2±1.1 | **75.7±0.9** | 74.8±0.8 | 75.6±1.0 |
| | 3C | 85.2±0.6 | 84.7±1.0 | 85.0±0.5 | 84.7±0.9 | 85.1±0.6 | 85.2±0.6 | 85.4±0.6 | 85.3±0.7 | **85.5±0.4** | 85.3±0.4 |
| | 4C | 61.0±1.1 | 61.4±1.9 | 60.6±1.7 | **64.5±2.4** | 63.7±1.8 | 57.8±1.5 | 56.8±1.4 | 55.6±1.1 | 63.3±0.5 | 62.8±1.0 |
| | 5C | 82.0±0.6 | 82.3±0.7 | 82.1±0.8 | 82.9±0.7 | 82.7±0.5 | 81.0±0.3 | 80.9±0.5 | 80.5±0.9 | 82.6±0.8 | **83.0±0.9** |
| *FiLL (avg.)* | SP | 74.1±0.2 | 74.0±0.3 | 74.0±0.4 | **75.2±0.1** | **75.2±0.1** | 69.3±0.6 | 69.6±0.2 | 69.6±0.5 | 74.8±0.2 | 74.8±0.2 |
| | DP | 73.9±0.1 | 74.0±0.1 | 73.9±0.2 | 75.0±0.1 | **75.1±0.1** | 70.0±0.2 | 70.2±0.3 | 70.3±0.4 | 74.6±0.1 | 74.7±0.1 |
| | 3C | 74.1±0.1 | 74.1±0.1 | 74.0±0.2 | **75.3±0.1** | 75.2±0.1 | 69.8±0.3 | 69.5±0.4 | 69.4±0.5 | 74.8±0.1 | 74.8±0.1 |
| | 4C | 74.1±0.3 | 74.2±0.2 | 74.3±0.1 | **75.3±0.2** | 75.2±0.1 | 69.1±0.2 | 68.8±0.4 | 68.8±0.5 | 74.8±0.3 | 74.9±0.3 |
| | 5C | 75.1±0.1 | 75.1±0.2 | 75.0±0.2 | 76.1±0.1 | **76.2±0.1** | 70.0±0.2 | 70.1±0.5 | 70.3±0.3 | 75.7±0.2 | 75.6±0.3 |

**Table 7:** Link prediction test performance (accuracy in percentage) comparison for MSGNN with different $q$ values (multiples of $q_0 := 1/[2\max_{i,j}(\mathbf{A}_{i,j} - \mathbf{A}_{j,i})]$) when input networks are treated as unweighted. The best is marked in **bold red** and the second best is marked in underline blue.

| Data Set | Link Task | $q = 0$ | $q = 0.2q_0$ | $q = 0.4q_0$ | $q = 0.6q_0$ | $q = 0.8q_0$ | $q = q_0$ |
|---|---|---|---|---|---|---|---|
| *BitCoin-Alpha* | SP | 70.3±1.5 | 71.0±0.7 | 70.9±1.5 | **71.1±0.9** | 69.2±1.4 | 70.9±1.7 |
| | DP | 74.5±1.5 | 74.0±1.7 | 74.3±1.2 | 74.0±1.6 | **74.7±0.7** | 74.7±1.5 |
| | 3C | 85.7±0.3 | 86.1±0.4 | **86.4±0.3** | 85.8±0.6 | 86.0±0.5 | 86.1±0.5 |
| | 4C | 58.4±1.8 | 59.1±2.0 | 60.0±1.4 | **60.2±1.3** | 60.1±1.2 | 59.5±2.2 |
| | 5C | 83.7±0.5 | 84.0±0.5 | **84.1±0.4** | **84.1±0.4** | 83.8±0.2 | 83.8±0.8 |
| *BitCoin-OTC* | SP | **73.7±1.3** | 73.2±0.7 | **73.7±0.9** | 73.0±1.5 | 73.3±1.1 | 73.5±0.6 |
| | DP | 74.8±2.1 | 75.1±1.2 | **75.8±1.1** | 75.3±0.8 | 75.6±1.0 | 75.6±1.0 |
| | 3C | 85.1±0.6 | 85.1±0.5 | **85.3±0.5** | 85.3±0.8 | 85.3±0.6 | **85.3±0.4** |
| | 4C | 63.7±1.8 | **64.4±0.6** | 63.1±2.2 | 64.3±2.1 | 63.0±1.2 | 62.8±1.0 |
| | 5C | 82.7±0.5 | 82.8±0.9 | 82.7±0.7 | **83.2±0.8** | 82.7±0.9 | 83.0±0.9 |
| *FiLL (avg.)* | SP | **75.2±0.1** | 75.2±0.2 | **75.2±0.1** | 75.0±0.3 | 74.8±0.2 | 74.8±0.2 |
| | DP | **75.1±0.1** | 75.0±0.2 | 74.9±0.1 | 74.9±0.1 | 74.6±0.1 | 74.7±0.1 |
| | 3C | **75.2±0.1** | 75.2±0.1 | 75.2±0.1 | **75.2±0.1** | 74.9±0.2 | 74.8±0.1 |
| | 4C | 75.2±0.1 | **75.3±0.1** | 75.2±0.1 | 75.1±0.2 | 74.9±0.2 | 74.9±0.3 |
| | 5C | **76.2±0.1** | **76.2±0.1** | 76.1±0.1 | 75.9±0.2 | 75.8±0.1 | 75.6±0.3 |

we stack the real and imaginary parts of the top $3 + 1 = 4$ eigenvectors as input features for K-means, while for $\mathbf{F}_2$ we employ the top $4 + 1 = 5$ eigenvectors. We conclude that the top eigenvectors of our proposed magnetic signed Laplacian can separate some of the clusters while it confuses a pair, and that the separation ability decreases as we increase the noise level ($\gamma$ and/or $\eta$). In particular, simply using K-means on the top eigenvectors is not competitive compared to the performance of MSGNN after training.

Table 17 reports input feature sum statistics on real-world data sets: $m_1$ and $s_1$ denote the average and one standard error of the sum of input features for each node corresponding to the (T, F) tuple in Table 3, respectively, while $m_2$ and $s_2$ correspond to (T,T). We conclude that in the first four data sets the standard errors are much larger than the averages of the sums of the features, whereas in the *FiLL* data sets, the standard errors are much smaller than the average, see Table 17. Hence in the *FiLL* data sets these features may not show enough variability around the average to be very informative.

**Table 8:** Link prediction test performance (accuracy in percentage) comparison for MSGNN with various number of layers. The better variant is marked in **bold red** and the worse variant is marked in underline blue.

| Data Set | Link Task | no normalization | symmetric normalization |
|----------|-----------|------------------|-------------------------|
| BitCoin-Alpha | SP | **71.9±1.4** | 71.3±1.2 |
|  | DP | **73.4±1.0** | 72.5±1.5 |
|  | 3C | **84.8±0.8** | 84.4±0.6 |
|  | 4C | 57.9±2.2 | **58.5±0.7** |
|  | 5C | **82.2±0.6** | 81.9±0.9 |
| BitCoin-OTC | SP | **74.1±0.7** | 73.0±1.4 |
|  | DP | **73.5±0.8** | 71.8±1.1 |
|  | 3C | **83.5±0.7** | 83.3±0.7 |
|  | 4C | 59.5±1.8 | **59.8±0.7** |
|  | 5C | 80.4±0.8 | **80.9±0.9** |
| Slashdot | SP | **92.7±0.1** | 92.4±0.2 |
|  | DP | 92.8±0.1 | **93.1±0.1** |
|  | 3C | **86.6±0.1** | 86.1±0.3 |
|  | 4C | 77.9±0.9 | **78.2±0.3** |
|  | 5C | **78.1±0.5** | 76.8±0.6 |
| Epinions | SP | **86.1±0.5** | 85.4±0.5 |
|  | DP | 85.8±0.2 | **86.3±0.3** |
|  | 3C | 82.8±1.0 | **83.1±0.5** |
|  | 4C | **79.7±1.1** | 78.7±0.9 |
|  | 5C | **80.6±0.6** | 80.5±0.5 |
| FiLL (avg.) | SP | **76.2±0.1** | 76.1±0.1 |
|  | DP | **76.0±0.0** | 75.9±0.1 |
|  | 3C | **76.2±0.1** | 76.1±0.0 |
|  | 4C | **76.3±0.1** | 76.2±0.1 |
|  | 5C | **77.0±0.1** | **77.0±0.1** |

**Table 9:** Link prediction test performance (accuracy in percentage) comparison for MSGNN with various number of layers. The best is marked in **bold red** and the second best is marked in underline blue.

| Data Set | Link Task | 2 layers | 3 layers | 4 layers | 5 layers | 6 layers | 7 layers | 8 layers | 9 layers | 10 layers |
|----------|-----------|----------|----------|----------|----------|----------|----------|----------|----------|-----------|
| BitCoin-Alpha | SP | 71.3±1.2 | 71.2±0.6 | 71.9±1.0 | **72.7±0.5** | 72.2±0.8 | 70.9±2.1 | 70.3±0.7 | 70.5±0.6 | 69.4±1.7 |
|  | DP | 72.5±1.5 | 72.0±0.9 | **73.4±1.0** | 73.4±1.6 | 71.0±2.6 | 70.6±1.2 | 70.4±1.5 | 68.3±1.7 | 69.8±1.7 |
|  | 3C | 84.4±0.6 | 84.8±0.8 | 85.0±0.4 | **85.1±0.8** | 84.8±0.3 | 84.7±0.2 | 84.2±0.2 | 84.6±1.1 | 84.1±1.1 |
|  | 4C | 58.5±0.7 | 58.2±2.0 | **59.8±2.3** | 58.9±1.5 | 58.5±1.9 | 58.2±1.2 | 57.5±2.0 | 57.7±1.3 | 57.0±2.0 |
|  | 5C | 81.9±0.9 | 81.6±1.2 | **83.2±0.4** | 82.9±0.3 | 82.4±0.2 | 82.3±0.6 | 81.9±0.7 | 82.2±0.6 | 81.7±0.4 |
| BitCoin-OTC | SP | 73.0±1.4 | 71.5±1.3 | **74.1±1.1** | 73.2±1.4 | 73.2±2.2 | 71.3±1.1 | 71.6±0.9 | 70.3±1.0 | 69.2±1.9 |
|  | DP | 71.8±1.1 | 72.7±0.6 | **73.9±1.0** | 73.0±1.2 | 73.7±0.6 | 72.6±0.7 | 71.9±1.3 | 72.2±2.0 | 71.4±1.7 |
|  | 3C | 83.3±0.7 | 82.5±1.1 | **83.4±0.5** | 83.2±0.8 | 83.2±0.8 | 83.0±0.7 | 83.0±0.5 | 83.0±1.0 | 82.8±0.2 |
|  | 4C | 59.8±0.7 | 59.2±1.6 | 61.2±1.1 | **61.6±0.6** | 58.7±1.2 | 57.9±1.7 | 58.1±1.7 | 57.9±1.9 | 54.4±1.9 |
|  | 5C | **80.9±0.9** | 80.1±0.8 | 80.8±0.6 | 80.3±1.0 | 79.9±0.5 | 79.4±0.9 | 79.7±0.3 | 79.5±0.5 | 79.6±0.5 |
| Slashdot | SP | **92.4±0.2** | 92.0±0.3 | 92.3±0.2 | 92.0±0.3 | 91.7±0.3 | 91.8±0.2 | 91.3±0.3 | 91.4±0.3 | 90.9±0.7 |
|  | DP | 93.1±0.1 | 93.0±0.1 | **93.2±0.1** | 93.1±0.1 | 93.0±0.2 | 92.9±0.1 | 93.1±0.1 | 93.0±0.2 | 93.0±0.2 |
|  | 3C | 86.1±0.3 | 86.0±0.5 | 86.2±0.1 | 86.1±0.1 | 85.8±0.2 | 85.9±0.2 | **86.2±0.2** | 85.9±0.2 | 85.8±0.2 |
|  | 4C | 78.2±0.3 | 78.2±0.4 | **78.5±0.5** | 78.0±0.2 | 77.7±0.3 | 76.9±0.3 | 77.4±1.1 | 77.6±0.7 | 76.9±0.7 |
|  | 5C | 76.8±0.6 | 77.5±0.2 | **77.6±0.4** | 77.5±0.4 | 77.1±0.8 | 76.8±0.8 | 76.9±0.5 | 76.8±0.5 | 76.4±0.3 |
| Epinions | SP | **85.4±0.5** | 85.1±0.9 | **85.4±0.5** | **85.4±0.5** | 83.7±1.0 | 83.3±0.9 | 82.8±0.9 | 82.8±1.2 | 80.8±1.0 |
|  | DP | 86.3±0.3 | 86.1±0.2 | **86.7±0.1** | 86.5±0.1 | **86.7±0.2** | 86.5±0.4 | 86.6±0.2 | **86.7±0.1** | 86.5±0.1 |
|  | 3C | 83.1±0.5 | 83.2±0.2 | **83.6±0.2** | **83.6±0.3** | **83.6±0.2** | 83.2±0.5 | 83.4±0.3 | 83.5±0.3 | 83.3±0.2 |
|  | 4C | 78.7±0.9 | 79.8±0.3 | 79.3±1.2 | **80.1±0.4** | **80.1±0.5** | 79.0±0.8 | 79.6±0.5 | 79.8±0.5 | 78.8±1.1 |
|  | 5C | 80.5±0.5 | 79.9±1.0 | 80.8±0.5 | 80.7±0.3 | 80.2±0.8 | 80.5±0.2 | 80.8±0.2 | **80.9±0.3** | 80.2±0.5 |
| FiLL (avg.) | SP | 76.1±0.1 | 76.2±0.0 | **76.4±0.0** | 76.2±0.1 | 76.0±0.1 | 76.0±0.1 | 76.0±0.0 | 76.0±0.1 | 75.8±0.1 |
|  | DP | 75.9±0.1 | 76.0±0.1 | **76.1±0.1** | 76.0±0.1 | 75.8±0.1 | 75.8±0.2 | 75.8±0.1 | 75.7±0.1 | 75.6±0.0 |
|  | 3C | 76.1±0.0 | 76.2±0.1 | **76.4±0.1** | 76.2±0.0 | 76.0±0.0 | 76.0±0.0 | 76.0±0.1 | 75.9±0.1 | 75.8±0.1 |
|  | 4C | 76.2±0.1 | 76.3±0.0 | **76.5±0.1** | 76.3±0.0 | 76.1±0.1 | 76.1±0.0 | 76.1±0.0 | 76.0±0.1 | 75.8±0.1 |
|  | 5C | 77.0±0.1 | 77.0±0.1 | **77.2±0.1** | 77.0±0.1 | 76.9±0.1 | 76.8±0.1 | 76.9±0.1 | 76.7±0.1 | 76.6±0.1 |

**Table 10:** Runtime (seconds) comparison on link tasks for variants of MSGNN. Each variant is denoted by a $q$ value and a 2-tuple: (whether to include signed features, whether to include weighted features), where "T" and "F" stand for "True" and "False", respectively. "T" for weighted features means simply summing up entries in the adjacency matrix while "T'" means summing the absolute values of the entries. The fastest is marked in **bold red** and the second fastest is marked in underline blue.

| $q$ value
Data Set | Link Task | 0
(F, F) | (F, T) | (F, T') | (T, F) | (T, T) | $q_0 := 1/[2\max_{i,j}(\mathbf{A}_{i,j} - \mathbf{A}_{j,i})]$
(F, F) | (F, T) | (F, T') | (T, F) | (T, T) |
|---|---|---|---|---|---|---|---|---|---|---|---|
| *BitCoin-Alpha* | SP | 23 | **21** | 25 | 25 | 29 | 23 | **21** | 25 | 25 | 29 |
| | DP | 26 | **25** | 26 | **25** | 26 | 36 | 38 | 37 | 37 | 37 |
| | 3C | **28** | **28** | 29 | 29 | 29 | 37 | 37 | 37 | 36 | 37 |
| | 4C | _25_ | _25_ | **24** | 26 | _25_ | 36 | 36 | 36 | 36 | 36 |
| | 5C | 29 | _28_ | _28_ | **27** | 29 | 37 | 37 | 37 | 37 | 37 |
| *BitCoin-OTC* | SP | **26** | 27 | 27 | 28 | 30 | **26** | 27 | 27 | 28 | 30 |
| | DP | 28 | _27_ | 28 | 28 | **26** | 38 | 37 | 38 | 37 | 38 |
| | 3C | 33 | _32_ | _32_ | 33 | **30** | 38 | 38 | 38 | 39 | 37 |
| | 4C | _26_ | **24** | 27 | 27 | _26_ | 37 | 37 | 36 | 36 | 37 |
| | 5C | 32 | _30_ | 31 | 31 | **29** | 38 | 38 | 38 | 37 | 38 |
| *Slashdot* | SP | 226 | 227 | **221** | 224 | 227 | 226 | 227 | **221** | 224 | 227 |
| | DP | 223 | 223 | **222** | 227 | **222** | 223 | 223 | **222** | 227 | **222** |
| | 3C | 327 | 327 | 327 | 325 | **322** | 327 | 327 | 327 | 325 | **322** |
| | 4C | 231 | **227** | 232 | 229 | 232 | 232 | **227** | 232 | 229 | 232 |
| | 5C | 330 | **324** | 325 | 334 | 326 | 330 | **324** | 325 | 334 | 327 |
| *Epinions* | SP | 368 | 374 | **367** | 370 | 370 | 368 | 374 | **367** | 370 | 370 |
| | DP | **363** | 366 | _364_ | 376 | 369 | _364_ | 365 | _364_ | 376 | 369 |
| | 3C | **506** | 510 | 510 | 510 | 509 | **506** | 510 | 510 | 510 | 510 |
| | 4C | **374** | 377 | 382 | 381 | 384 | **374** | 376 | 382 | 380 | 384 |
| | 5C | 511 | 511 | **508** | 518 | 517 | 511 | 511 | _509_ | 518 | 517 |
| *FiLL (avg.)* | SP | 36 | 36 | 35 | 36 | **32** | 36 | 36 | 35 | 36 | **32** |
| | DP | 36 | 36 | **35** | **35** | 36 | 36 | 36 | 36 | 37 | 36 |
| | 3C | 44 | **42** | 43 | **42** | 43 | 43 | 44 | 44 | 44 | 43 |
| | 4C | 35 | 36 | _34_ | _34_ | **31** | 35 | 35 | 35 | 36 | 35 |
| | 5C | **43** | 45 | **43** | **43** | **43** | 44 | 44 | 44 | 45 | 44 |

# D   Experimental Results on Individual Years for *FiLL*

Table 18, 19, 20 and 21 provide full experimental results on financial data sets for individual years for the main experiments, while Table 22, 23, 24, 25, 26, 27, 28 and 29 contain individual results for the years for the ablation study.

**Table 11:** Runtime (seconds) comparison on link tasks for variants of MSGNN with different $q$ values (multiples of $q_0 := 1/[2\max_{i,j}(\mathbf{A}_{i,j} - \mathbf{A}_{j,i})]$). The fastest is marked in **bold red** and the second fastest is marked in underline blue.

| Data Set | Link Task | $q = 0$ | $q = 0.2q_0$ | $q = 0.4q_0$ | $q = 0.6q_0$ | $q = 0.8q_0$ | $q = q_0$ |
|---|---|---|---|---|---|---|---|
| *BitCoin-Alpha* | SP | 29 | 29 | 29 | 29 | 30 | **26** |
|  | DP | **26** | 31 | 31 | 31 | 33 | 37 |
|  | 3C | **29** | 37 | 37 | 37 | 37 | 37 |
|  | 4C | **25** | 31 | 31 | 32 | 32 | 36 |
|  | 5C | **29** | 36 | 36 | 36 | 36 | 37 |
| *BitCoin-OTC* | SP | 30 | 30 | 30 | **27** | **27** | **27** |
|  | DP | **26** | 36 | 36 | 36 | 35 | 38 |
|  | 3C | **30** | 45 | 45 | 45 | 45 | 37 |
|  | 4C | **26** | 35 | 35 | 35 | 35 | 37 |
|  | 5C | **29** | 45 | 45 | 45 | 45 | 38 |
| *Slashdot* | SP | 227 | **226** | **226** | **226** | **226** | **226** |
|  | DP | **222** | **222** | **222** | **222** | **222** | **222** |
|  | 3C | **322** | **322** | **322** | **322** | **322** | **322** |
|  | 4C | **232** | **232** | **232** | **232** | **232** | **232** |
|  | 5C | **326** | 327 | 327 | 327 | 327 | 327 |
| *Epinions* | SP | **370** | **370** | **370** | **370** | **370** | **370** |
|  | DP | **369** | **369** | **369** | **369** | **369** | **369** |
|  | 3C | **509** | 510 | 510 | 510 | 510 | 510 |
|  | 4C | **384** | **384** | **384** | **384** | **384** | **384** |
|  | 5C | **517** | **517** | **517** | **517** | **517** | **517** |
| *FiLL (avg.)* | SP | **32** | 36 | 36 | 36 | 36 | 36 |
|  | DP | **36** | **36** | **36** | **36** | **36** | **36** |
|  | 3C | 43 | 44 | 44 | 43 | **42** | 43 |
|  | 4C | **31** | 36 | 36 | 36 | 36 | 35 |
|  | 5C | **43** | 45 | 45 | 44 | 44 | 44 |

**Table 12:** Runtime (seconds) comparison the signed and directed link prediction tasks introduced in Sec. 4.1 where all networks are treated as unweighted. The fastest is marked in **bold red** and the second fastest is marked in underline blue.

| Data Set | Link Task | SGCN | SDGNN | SiGAT | SNEA | SSSNET | SigMaNet | MSGNN |
|---|---|---|---|---|---|---|---|---|
| *BitCoin-Alpha* | SP | 352 | 124 | 277 | 438 | 49 | 56 | **32** |
|  | DP | 328 | 196 | 432 | 498 | 53 | 59 | **34** |
|  | 3C | 403 | 150 | 288 | 446 | 50 | 64 | **40** |
|  | 4C | 385 | 133 | 293 | 471 | 49 | 73 | **33** |
|  | 5C | 350 | 373 | 468 | 570 | 45 | 60 | **40** |
| *BitCoin-OTC* | SP | 340 | 140 | 397 | 584 | 50 | 52 | **34** |
|  | DP | 471 | 243 | 426 | 941 | 48 | 77 | **36** |
|  | 3C | 292 | 252 | 502 | 551 | 53 | 58 | **44** |
|  | 4C | 347 | 143 | 487 | 607 | 47 | 77 | **35** |
|  | 5C | 460 | 507 | 500 | 959 | **45** | 52 | **45** |
| *FiLL (avg.)* | SP | 591 | 320 | 367 | 617 | 99 | 122 | **36** |
|  | DP | 387 | 316 | 363 | 386 | 95 | 122 | **35** |
|  | 3C | 542 | 471 | 298 | 657 | 76 | 76 | **43** |
|  | 4C | 608 | 384 | 343 | 642 | 76 | 111 | **35** |
|  | 5C | 318 | 534 | 266 | 521 | 108 | 119 | **44** |

**Table 13:** Runtime (seconds) comparison for variants of MSGNN where input networks are treated as unweighted. Each variant is denoted by a $q$ value and a 2-tuple: (whether to include signed features, whether to include weighted features), where "T" and "F" stand for "True" and "False", respectively. "T" for weighted features means simply summing up entries in the adjacency matrix while "T'" means summing the absolute values of the entries. The fastest is marked in **bold red** and the second fastest is marked in underline blue.

| $q$ value Data Set | Link Task | 0 (F, F) | (F, T) | (F, T') | (T, F) | (T, T) | $q_0 := 1/[2 \max_{i,j}(\mathbf{A}_{i,j} - \mathbf{A}_{j,i})]$ (F, F) | (F, T) | (F, T') | (T, F) | (T, T) |
|---|---|---|---|---|---|---|---|---|---|---|---|
| *BitCoin-Alpha* | SP | 33 | 33 | 34 | 34 | **32** | 33 | 33 | 33 | 33 | 33 |
| | DP | 34 | 34 | 34 | 34 | **32** | 34 | 34 | 34 | 35 | 34 |
| | 3C | 39 | 40 | 40 | 40 | **36** | 40 | 40 | 40 | 40 | 40 |
| | 4C | 33 | 33 | 33 | 33 | **31** | 33 | 34 | 33 | 34 | 33 |
| | 5C | 40 | 40 | 40 | 39 | **36** | 40 | 39 | 40 | 40 | 40 |
| *BitCoin-OTC* | SP | **33** | 34 | **33** | 34 | 34 | **33** | 34 | 34 | 34 | 35 |
| | DP | 35 | **34** | **34** | 35 | 35 | 35 | **34** | **34** | **34** | 36 |
| | 3C | 41 | 41 | **40** | 41 | **40** | 41 | **40** | 41 | 41 | 44 |
| | 4C | **32** | 33 | 33 | 34 | 33 | 34 | 34 | 34 | 34 | 35 |
| | 5C | **40** | 41 | 41 | **40** | 41 | 41 | 41 | 41 | 41 | 45 |
| *FiLL (avg.)* | SP | 37 | 37 | 37 | 39 | **36** | 37 | **36** | **36** | 38 | **36** |
| | DP | 41 | 37 | 37 | 37 | **35** | 41 | 36 | 37 | 37 | **35** |
| | 3C | 68 | 46 | 45 | 45 | **43** | 69 | 45 | 45 | 44 | **43** |
| | 4C | 36 | 36 | 36 | 37 | **35** | **35** | **35** | **35** | 36 | **35** |
| | 5C | 63 | 46 | 46 | 46 | **44** | 65 | **44** | 45 | 45 | **44** |

**Table 14:** Runtime (seconds) comparison for MSGNN with different $q$ values (multiples of $q_0 := 1/[2 \max_{i,j}(\mathbf{A}_{i,j} - \mathbf{A}_{j,i})]$) when input networks are treated as unweighted. The fastest is marked in **bold red** and the second fastest is marked in underline blue.

| Data Set | Link Task | $q = 0$ | $q = 0.2q_0$ | $q = 0.4q_0$ | $q = 0.6q_0$ | $q = 0.8q_0$ | $q = q_0$ |
|---|---|---|---|---|---|---|---|
| *BitCoin-Alpha* | SP | 32 | **31** | 32 | **31** | 34 | 33 |
| | DP | **32** | **32** | **32** | **32** | 33 | 34 |
| | 3C | **36** | **36** | **36** | **36** | 38 | 40 |
| | 4C | **31** | 32 | 32 | **31** | 33 | 33 |
| | 5C | **36** | **36** | 37 | 37 | 40 | 40 |
| *BitCoin-OTC* | SP | 34 | **33** | 35 | 34 | 36 | 35 |
| | DP | 35 | **34** | 36 | 36 | 36 | 36 |
| | 3C | **40** | 41 | 43 | 44 | 45 | 44 |
| | 4C | **33** | **33** | 35 | 35 | 35 | 35 |
| | 5C | **41** | **41** | 45 | 44 | 45 | 45 |
| *FiLL (avg.)* | SP | **36** | **36** | **36** | **36** | **36** | **36** |
| | DP | **35** | 36 | 36 | **35** | 36 | **35** |
| | 3C | **43** | 44 | 44 | **43** | **43** | **43** |
| | 4C | 35 | 36 | 36 | **34** | 35 | 35 |
| | 5C | **44** | 45 | 45 | **44** | 45 | **44** |

**Table 15:** Runtime (seconds) comparison for MSGNN with various number of layers. The faster variant is marked in **bold red** and the slower variant is marked in underline blue.

| Data Set | Link Task | no normalization | symmetric normalization |
|---|---|---|---|
| *BitCoin-Alpha* | SP | 46 | **29** |
| | DP | 89 | **37** |
| | 3C | 101 | **37** |
| | 4C | 63 | **36** |
| | 5C | 81 | **37** |
| *BitCoin-OTC* | SP | 47 | **30** |
| | DP | 158 | **38** |
| | 3C | 147 | **37** |
| | 4C | 63 | **37** |
| | 5C | 106 | **38** |
| *Slashdot* | SP | 1209 | **227** |
| | DP | 1365 | **222** |
| | 3C | 1475 | **322** |
| | 4C | 1378 | **232** |
| | 5C | 1464 | **327** |
| *Epinions* | SP | 1095 | **370** |
| | DP | 1103 | **369** |
| | 3C | 950 | **510** |
| | 4C | 956 | **384** |
| | 5C | 1194 | **517** |
| *FiLL (avg.)* | SP | 64 | **32** |
| | DP | 76 | **36** |
| | 3C | 78 | **43** |
| | 4C | 60 | **35** |
| | 5C | 81 | **44** |

**Table 16:** Runtime (seconds) comparison for MSGNN with various number of layers. The fastest is marked in **bold red** and the second fastest is marked in underline blue.

| Data Set | Link Task | 2 layers | 3 layers | 4 layers | 5 layers | 6 layers | 7 layers | 8 layers | 9 layers | 10 layers |
|---|---|---|---|---|---|---|---|---|---|---|
| *BitCoin-Alpha* | SP | **29** | 33 | 36 | 52 | 46 | 50 | 54 | 58 | 61 |
| | DP | **37** | 38 | **37** | 51 | 46 | 51 | 55 | 58 | 61 |
| | 3C | 37 | **32** | 38 | 46 | 44 | 52 | 57 | 60 | 61 |
| | 4C | 36 | **26** | **26** | 48 | 43 | 49 | 53 | 58 | 61 |
| | 5C | 37 | **32** | 38 | 50 | 47 | 52 | 57 | 60 | 61 |
| *BitCoin-OTC* | SP | 30 | 29 | **28** | 44 | 46 | 51 | 55 | 59 | 60 |
| | DP | 38 | **31** | 37 | 43 | 46 | 48 | 55 | 59 | 61 |
| | 3C | 37 | **36** | 41 | 46 | 49 | 54 | 58 | 59 | 62 |
| | 4C | 37 | **27** | 29 | 45 | 45 | 49 | 56 | 60 | 61 |
| | 5C | 38 | **35** | 41 | 47 | 50 | 53 | 58 | 61 | 62 |
| *Slashdot* | SP | **227** | 360 | 374 | 403 | 403 | 500 | 501 | 608 | 610 |
| | DP | **222** | 298 | 299 | 402 | 403 | 497 | 498 | 603 | 604 |
| | 3C | **322** | 399 | 399 | 497 | 497 | 597 | 599 | 707 | 708 |
| | 4C | **232** | 304 | 314 | 412 | 413 | 503 | 505 | 615 | 617 |
| | 5C | **327** | 399 | 402 | 504 | 504 | 600 | 601 | 711 | 713 |
| *Epinions* | SP | **370** | 524 | 536 | 696 | 704 | 822 | 837 | 980 | 987 |
| | DP | **369** | 509 | 535 | 687 | 695 | 821 | 838 | 983 | 984 |
| | 3C | **510** | 647 | 650 | 814 | 819 | 962 | 968 | 1122 | 1127 |
| | 4C | **384** | 508 | 519 | 676 | 684 | 812 | 816 | 980 | 986 |
| | 5C | **517** | 649 | 655 | 810 | 813 | 967 | 985 | 1110 | 1110 |
| *FiLL (avg.)* | SP | **32** | 42 | 44 | 55 | 58 | 61 | 64 | 70 | 72 |
| | DP | **36** | 43 | 45 | 55 | 59 | 62 | 65 | 70 | 72 |
| | 3C | **43** | 50 | 51 | 62 | 65 | 68 | 71 | 77 | 78 |
| | 4C | **35** | 41 | 43 | 54 | 57 | 60 | 64 | 70 | 71 |
| | 5C | **44** | 50 | 52 | 64 | 66 | 69 | 72 | 77 | 79 |

**Table 17:** Input feature sum statistics: $m_1$ and $s_1$ denote the average and one standard error of the sum of input features for each node corresponding to the (T, F) tuple in Table 3, respectively, while $m_2$ and $s_2$ correspond to (T,T).

| Data Set | $m_1$ | $s_1$ | $\frac{m_1}{s_1}$ | $m_2$ | $s_2$ | $\frac{m_2}{s_2}$ |
|---|---|---|---|---|---|---|
| *BitCoin-Alpha* | 29.076 | 84.322 | 0.345 | 12.787 | 34.446 | 0.152 |
| *BitCoin-OTC* | 30.564 | 102.996 | 0.297 | 12.104 | 38.299 | 0.118 |
| *Slashdot* | 13.372 | 44.756 | 0.299 | 13.372 | 44.756 | 0.299 |
| *Epinions* | 12.765 | 60.549 | 0.211 | 12.765 | 60.549 | 0.211 |
| *FiLL-pvCLCL (2000)* | 29.878 | 9.906 | 3.016 | 177.599 | 40.650 | 17.928 |
| *FiLL-OPCL (2000)* | 28.182 | 8.834 | 3.190 | 172.000 | 38.860 | 19.471 |
| *FiLL-pvCLCL (2001)* | 31.404 | 11.624 | 2.702 | 177.599 | 44.367 | 15.278 |
| *FiLL-OPCL (2001)* | 30.026 | 10.243 | 2.931 | 172.000 | 40.412 | 16.792 |
| *FiLL-pvCLCL (2002)* | 32.865 | 19.055 | 1.725 | 177.599 | 64.000 | 9.320 |
| *FiLL-OPCL (2002)* | 30.154 | 15.255 | 1.977 | 172.000 | 59.662 | 11.275 |
| *FiLL-pvCLCL (2003)* | 28.622 | 12.286 | 2.330 | 177.599 | 59.288 | 14.456 |
| *FiLL-OPCL (2003)* | 27.085 | 11.552 | 2.345 | 172.000 | 55.738 | 14.889 |
| *FiLL-pvCLCL (2004)* | 26.031 | 8.965 | 2.904 | 177.599 | 45.693 | 19.810 |
| *FiLL-OPCL (2004)* | 24.633 | 8.222 | 2.996 | 172.000 | 43.912 | 20.921 |
| *FiLL-pvCLCL (2005)* | 26.047 | 8.885 | 2.932 | 177.599 | 47.953 | 19.988 |
| *FiLL-OPCL (2005)* | 23.884 | 7.106 | 3.361 | 172.000 | 40.651 | 24.206 |
| *FiLL-pvCLCL (2006)* | 28.621 | 11.214 | 2.552 | 177.599 | 50.197 | 15.837 |
| *FiLL-OPCL (2006)* | 26.219 | 8.764 | 2.992 | 172.000 | 43.565 | 19.626 |
| *FiLL-pvCLCL (2007)* | 33.365 | 18.674 | 1.787 | 177.599 | 80.030 | 9.510 |
| *FiLL-OPCL (2007)* | 30.564 | 14.449 | 2.115 | 172.000 | 60.453 | 11.904 |
| *FiLL-pvCLCL (2008)* | 45.040 | 31.693 | 1.421 | 177.599 | 97.070 | 5.604 |
| *FiLL-OPCL (2008)* | 42.205 | 26.869 | 1.571 | 172.000 | 84.361 | 6.401 |
| *FiLL-pvCLCL (2009)* | 43.435 | 31.176 | 1.393 | 177.599 | 109.911 | 5.697 |
| *FiLL-OPCL (2009)* | 36.304 | 19.429 | 1.869 | 172.000 | 75.508 | 8.853 |
| *FiLL-pvCLCL (2010)* | 25.883 | 14.252 | 1.816 | 177.599 | 67.060 | 12.461 |
| *FiLL-OPCL (2010)* | 26.908 | 12.924 | 2.082 | 172.000 | 61.325 | 13.309 |
| *FiLL-pvCLCL (2011)* | 41.301 | 31.316 | 1.319 | 177.604 | 116.559 | 5.671 |
| *FiLL-OPCL (2011)* | 36.046 | 26.133 | 1.379 | 172.000 | 102.552 | 6.582 |
| *FiLL-pvCLCL (2012)* | 29.015 | 13.159 | 2.205 | 177.599 | 54.519 | 13.496 |
| *FiLL-OPCL (2012)* | 26.120 | 9.786 | 2.669 | 172.000 | 46.006 | 17.576 |
| *FiLL-pvCLCL (2013)* | 26.538 | 13.247 | 2.003 | 177.599 | 69.771 | 13.407 |
| *FiLL-OPCL (2013)* | 25.754 | 13.107 | 1.965 | 172.000 | 53.168 | 13.123 |
| *FiLL-pvCLCL (2014)* | 25.220 | 9.101 | 2.771 | 177.599 | 48.146 | 19.514 |
| *FiLL-OPCL (2014)* | 25.366 | 10.319 | 2.458 | 172.000 | 50.838 | 16.668 |
| *FiLL-pvCLCL (2015)* | 25.859 | 14.186 | 1.823 | 177.599 | 57.031 | 12.519 |
| *FiLL-OPCL (2015)* | 27.853 | 15.188 | 1.834 | 172.000 | 63.501 | 11.325 |
| *FiLL-pvCLCL (2016)* | 30.540 | 18.321 | 1.667 | 177.599 | 54.241 | 9.694 |
| *FiLL-OPCL (2016)* | 29.418 | 18.629 | 1.579 | 172.000 | 48.518 | 9.233 |
| *FiLL-pvCLCL (2017)* | 28.700 | 10.860 | 2.643 | 177.599 | 39.566 | 16.353 |
| *FiLL-OPCL (2017)* | 27.020 | 9.549 | 2.830 | 172.000 | 33.605 | 18.012 |
| *FiLL-pvCLCL (2018)* | 25.543 | 10.128 | 2.522 | 177.599 | 51.628 | 17.535 |
| *FiLL-OPCL (2018)* | 25.859 | 11.772 | 2.197 | 172.000 | 64.979 | 14.611 |
| *FiLL-pvCLCL (2019)* | 28.781 | 13.533 | 2.127 | 177.599 | 47.949 | 13.123 |
| *FiLL-OPCL (2019)* | 26.188 | 11.474 | 2.282 | 172.000 | 43.134 | 14.990 |

**Table 18:** Full link prediction test accuracy (%) comparison for directions (and signs) on *FiLL-pvCLCL* data sets on individual years 2000-2010. The best is marked in **bold red** and the second best is marked in underline blue. The link prediction tasks are introduced in Sec. 4.1.

| Year | Link Task | SGCN | SDGNN | SiGAT | SNEA | SSSNET | SigMaNet | MSGNN |
|------|-----------|------|-------|-------|------|--------|----------|-------|
| 2000 | SP | 87.3±0.3 | 78.2±2.0 | 70.9±0.6 | 88.9±0.2 | 87.3±2.1 | 59.1±11.8 | **89.0±0.4** |
|      | DP | 87.1±0.2 | 78.6±1.1 | 70.6±0.7 | 88.9±0.3 | 87.7±0.9 | 53.5±9.4 | **89.1±0.5** |
|      | 3C | 59.8±0.4 | 53.0±1.2 | 47.9±0.5 | 60.8±0.5 | 58.5±2.2 | 31.9±7.4 | **61.5±0.7** |
|      | 4C | 71.1±0.4 | 66.4±1.3 | 56.5±0.4 | **72.5±0.5** | 69.0±1.2 | 23.6±8.0 | 71.9±0.5 |
|      | 5C | 53.6±0.3 | 49.8±0.8 | 43.5±0.2 | **54.0±0.3** | 51.3±1.5 | 20.7±7.6 | 53.9±0.4 |
| 2001 | SP | 88.0±0.3 | 80.2±1.4 | 71.5±1.0 | 90.3±0.2 | 85.5±3.4 | 46.2±10.7 | **90.7±0.2** |
|      | DP | 88.3±0.2 | 78.9±2.6 | 70.6±0.5 | 90.2±0.3 | 88.4±1.3 | 47.1±5.6 | **90.7±0.1** |
|      | 3C | 60.3±0.3 | 54.1±1.0 | 48.6±0.6 | 61.7±0.3 | 58.8±4.8 | 35.9±6.6 | **63.1±0.5** |
|      | 4C | 74.1±0.3 | 69.7±1.5 | 58.4±0.9 | **76.4±0.4** | 72.1±0.9 | 23.2±7.9 | 75.7±0.3 |
|      | 5C | 55.9±0.3 | 52.5±0.3 | 45.2±0.4 | **57.0±0.2** | 54.5±1.4 | 16.4±4.6 | 56.8±0.2 |
| 2002 | SP | 88.7±0.2 | 83.8±1.2 | 79.1±0.5 | 90.7±0.2 | 89.1±1.3 | 48.4±7.6 | **91.4±0.2** |
|      | DP | 88.8±0.2 | 84.4±0.9 | 78.9±0.7 | 90.7±0.3 | 89.8±1.1 | 51.8±11.7 | **91.5±0.2** |
|      | 3C | 62.1±0.4 | 60.6±0.6 | 56.3±0.8 | 64.2±0.4 | 63.8±0.2 | 31.0±1.8 | **66.2±0.2** |
|      | 4C | 84.3±0.5 | 82.8±0.7 | 75.3±0.7 | 85.6±0.3 | 80.0±2.4 | 19.7±4.1 | **85.7±0.4** |
|      | 5C | 65.7±0.3 | 64.2±0.1 | 58.0±0.9 | **67.1±0.1** | 57.4±3.4 | 21.5±3.6 | 66.7±0.4 |
| 2003 | SP | 86.7±0.6 | 80.7±1.1 | 76.4±0.7 | 87.9±0.5 | 86.6±1.3 | 54.8±7.7 | **89.5±0.4** |
|      | DP | 87.2±0.4 | 80.9±1.1 | 76.9±0.6 | 88.6±0.4 | 87.6±1.3 | 44.4±5.4 | **89.6±0.4** |
|      | 3C | 59.5±0.3 | 56.6±0.9 | 53.2±0.5 | 61.1±0.6 | 58.0±2.2 | 35.1±7.1 | **63.1±0.5** |
|      | 4C | 80.9±0.3 | 78.4±0.8 | 69.6±0.4 | 82.2±0.3 | 77.9±3.0 | 28.9±9.3 | **82.7±0.4** |
|      | 5C | 61.4±0.1 | 59.6±0.4 | 53.4±0.6 | 62.5±0.1 | 56.4±1.3 | 17.9±7.3 | **62.7±0.4** |
| 2004 | SP | 86.3±0.3 | 76.8±2.7 | 72.4±0.5 | 88.0±0.3 | 86.2±1.7 | 46.3±9.6 | **88.7±0.3** |
|      | DP | 86.1±0.4 | 75.2±1.5 | 72.8±0.5 | 87.9±0.5 | 87.2±0.8 | 50.0±6.7 | **88.8±0.3** |
|      | 3C | 58.7±0.2 | 50.8±1.2 | 49.1±0.3 | 59.7±0.4 | 59.5±0.9 | 34.4±2.1 | **61.6±0.4** |
|      | 4C | 77.1±0.3 | 71.9±1.6 | 61.1±1.4 | **78.9±0.4** | 75.9±0.9 | 19.9±2.1 | 78.7±0.4 |
|      | 5C | 57.7±0.4 | 53.8±1.0 | 47.9±0.6 | **58.8±0.4** | 55.0±0.7 | 21.7±3.4 | 58.7±0.6 |
| 2005 | SP | 85.1±0.2 | 76.3±1.4 | 74.9±0.6 | 86.5±0.6 | 85.5±1.5 | 53.0±13.7 | **87.8±0.4** |
|      | DP | 84.9±0.3 | 76.3±2.3 | 74.1±0.7 | 86.7±0.3 | 85.9±0.9 | 47.8±5.2 | **87.8±0.4** |
|      | 3C | 57.1±0.2 | 53.3±1.2 | 50.1±0.5 | 59.1±0.3 | 57.9±1.0 | 30.7±4.4 | **60.9±0.6** |
|      | 4C | 79.2±0.4 | 75.3±1.0 | 67.4±0.3 | **80.5±0.1** | 77.6±0.7 | 20.3±7.3 | 80.5±0.2 |
|      | 5C | 60.4±0.5 | 57.6±0.4 | 52.2±0.2 | **61.4±0.4** | 57.8±1.0 | 21.1±5.6 | 61.0±0.2 |
| 2006 | SP | 88.7±0.2 | 82.7±1.2 | 75.1±1.0 | 90.4±0.1 | 90.0±0.7 | 38.8±6.5 | **91.0±0.2** |
|      | DP | 88.8±0.3 | 83.2±0.9 | 75.7±0.7 | 90.6±0.5 | 89.1±1.2 | 46.4±7.6 | **91.1±0.1** |
|      | 3C | 61.9±0.3 | 56.9±1.4 | 52.8±0.4 | 63.1±0.4 | 61.5±1.9 | 37.4±7.1 | **64.1±0.3** |
|      | 4C | 81.2±0.2 | 77.8±0.9 | 66.9±0.6 | 83.0±0.4 | 80.6±0.4 | 25.7±7.3 | **83.1±0.4** |
|      | 5C | 62.1±0.4 | 58.4±0.7 | 53.2±0.2 | **63.3±0.1** | 58.7±1.8 | 16.2±7.3 | 62.8±0.3 |
| 2007 | SP | 87.8±0.3 | 83.9±0.8 | 79.2±0.3 | 89.4±0.3 | 89.3±0.5 | 55.4±13.3 | **90.4±0.5** |
|      | DP | 88.3±0.4 | 83.6±0.4 | 79.7±0.4 | 89.7±0.4 | 88.4±2.0 | 42.0±12.9 | **90.6±0.4** |
|      | 3C | 65.4±0.6 | 64.0±0.8 | 60.6±0.2 | 67.0±0.4 | 66.5±0.6 | 29.6±6.9 | **69.0±0.3** |
|      | 4C | 86.5±0.4 | 84.2±0.9 | 77.5±0.2 | 87.7±0.3 | 86.2±0.9 | 23.6±5.9 | **88.4±0.2** |
|      | 5C | 68.8±0.4 | 66.9±0.6 | 61.5±0.5 | **69.8±0.1** | 65.7±2.3 | 25.5±11.1 | 69.7±0.2 |
| 2008 | SP | 94.9±0.2 | 92.5±0.8 | 83.4±0.8 | 95.7±0.3 | 94.2±2.1 | 45.0±16.6 | **96.4±0.2** |
|      | DP | 95.2±0.2 | 93.2±0.4 | 82.2±0.3 | 95.9±0.1 | 95.1±0.8 | 39.9±18.4 | **96.3±0.2** |
|      | 3C | 75.4±0.5 | 76.6±0.5 | 68.3±0.5 | 77.2±0.5 | 73.3±3.2 | 22.4±9.5 | **78.9±0.2** |
|      | 4C | 95.7±0.3 | 95.5±0.3 | 87.0±1.0 | **96.2±0.2** | 95.4±0.4 | 25.0±16.6 | 96.2±0.3 |
|      | 5C | 80.9±0.2 | 80.4±0.4 | 71.1±0.4 | 81.9±0.1 | 75.0±4.4 | 19.4±10.4 | **82.0±0.4** |
| 2009 | SP | 96.0±0.3 | 91.2±1.4 | 87.0±0.5 | 96.9±0.2 | 96.3±1.1 | 45.9±13.0 | **97.8±0.2** |
|      | DP | 96.3±0.3 | 91.6±0.5 | 87.2±0.5 | 97.2±0.1 | 96.5±0.6 | 43.0±14.2 | **97.6±0.1** |
|      | 3C | 75.3±0.2 | 73.1±0.5 | 70.0±0.3 | 76.5±0.2 | 74.5±2.1 | 37.3±8.5 | **78.6±0.3** |
|      | 4C | 94.1±0.3 | 92.4±0.4 | 87.7±0.9 | 94.8±0.2 | 93.4±0.4 | 30.2±10.5 | **95.0±0.2** |
|      | 5C | 78.8±0.3 | 77.0±0.4 | 72.8±0.4 | 79.5±0.3 | 74.6±3.2 | 16.7±11.3 | **79.8±0.2** |
| 2010 | SP | 90.9±0.4 | 85.1±0.7 | 79.2±0.9 | 92.1±0.3 | 90.4±2.4 | 52.5±10.5 | **92.8±0.3** |
|      | DP | 91.0±0.2 | 86.0±1.1 | 78.4±0.8 | 91.9±0.3 | 90.5±0.8 | 45.8±6.1 | **92.7±0.4** |
|      | 3C | 64.5±0.3 | 63.1±0.6 | 56.2±0.9 | 65.7±0.3 | 61.6±2.3 | 33.8±2.2 | **68.5±0.4** |
|      | 4C | 89.8±0.2 | 88.3±0.4 | 79.6±0.9 | 90.3±0.3 | 87.0±1.2 | 28.4±5.5 | **91.1±0.4** |
|      | 5C | 71.5±0.3 | 69.7±0.4 | 62.6±0.8 | 72.3±0.2 | 68.1±1.1 | 17.8±5.7 | **72.4±0.1** |

**Table 19:** Full link prediction test accuracy (%) comparison for directions (and signs) on *FiLL-pvCLCL* data sets on individual years 2011-2020. The best is marked in **bold red** and the second best is marked in underline blue. The link prediction tasks are introduced in Sec. 4.1.

| Year | Link Task | SGCN | SDGNN | SiGAT | SNEA | SSSNET | SigMaNet | MSGNN |
|------|-----------|------|-------|-------|------|--------|----------|-------|
| 2011 | SP | 97.3±0.2 | 94.5±0.9 | 89.3±0.3 | 97.7±0.2 | 98.3±0.1 | 66.2±16.0 | **98.7±0.2** |
|      | DP | 97.4±0.3 | 95.4±0.9 | 89.8±0.5 | 97.8±0.2 | 98.1±0.2 | 45.0±20.6 | **98.7±0.1** |
|      | 3C | 84.3±0.2 | 82.0±0.4 | 77.7±0.4 | 84.7±0.3 | 82.3±3.1 | 33.0±17.4 | **86.2±0.3** |
|      | 4C | 97.2±0.1 | 96.7±0.5 | 90.0±0.3 | 97.7±0.1 | 98.1±0.2 | 13.9±7.9 | **98.3±0.2** |
|      | 5C | 84.9±0.3 | 83.2±0.8 | 78.2±0.4 | 85.5±0.2 | 82.0±3.6 | 18.6±12.3 | **87.2±0.4** |
| 2012 | SP | 90.9±0.4 | 83.4±1.3 | 74.5±1.1 | **92.7±0.2** | 89.0±4.3 | 42.3±6.0 | 92.7±0.3 |
|      | DP | 90.8±0.2 | 83.7±1.8 | 72.8±1.2 | **92.8±0.1** | 91.1±1.4 | 38.1±9.7 | 92.6±0.3 |
|      | 3C | 64.4±0.2 | 58.6±1.3 | 52.1±0.8 | 65.9±0.3 | 62.7±0.7 | 33.1±5.0 | **67.4±0.3** |
|      | 4C | 86.1±0.3 | 82.0±0.6 | 69.8±1.2 | **87.2±0.4** | 85.2±0.4 | 26.7±19.2 | 86.9±0.4 |
|      | 5C | 66.2±0.3 | 62.4±0.7 | 53.7±0.8 | **67.2±0.5** | 63.7±1.9 | 15.2±2.7 | 67.0±0.2 |
| 2013 | SP | 88.1±0.2 | 82.1±0.8 | 80.7±0.4 | 89.2±0.3 | 88.8±1.0 | 43.5±9.2 | **90.5±0.3** |
|      | DP | 87.5±0.4 | 82.4±0.7 | 80.6±0.6 | 88.7±0.4 | 87.7±0.6 | 56.6±14.4 | **90.4±0.3** |
|      | 3C | 63.2±0.2 | 61.5±0.7 | 59.1±0.3 | 64.5±0.2 | 64.7±0.8 | 33.1±2.8 | **66.1±0.3** |
|      | 4C | 84.6±0.3 | 81.6±0.4 | 75.8±0.4 | 85.5±0.4 | 84.2±0.4 | 26.1±19.6 | **86.1±0.3** |
|      | 5C | 65.7±0.2 | 64.0±0.4 | 59.9±0.2 | 66.5±0.2 | 64.0±0.6 | 16.3±7.7 | **66.8±0.3** |
| 2014 | SP | 84.5±0.2 | 75.9±1.7 | 70.4±0.6 | 86.4±0.4 | 85.5±1.2 | 49.5±6.1 | **87.3±0.2** |
|      | DP | 84.3±0.5 | 75.3±1.2 | 70.9±0.6 | 86.4±0.1 | 84.8±1.6 | 42.7±11.6 | **87.2±0.3** |
|      | 3C | 57.5±0.2 | 53.3±0.5 | 48.7±0.5 | 59.6±0.3 | 57.8±2.0 | 31.6±2.5 | **60.6±0.2** |
|      | 4C | 77.9±0.4 | 74.5±0.9 | 63.7±0.8 | 79.9±0.1 | 76.3±1.2 | 29.2±7.7 | **80.2±0.2** |
|      | 5C | 58.7±0.5 | 56.1±0.9 | 50.1±1.0 | 60.1±0.5 | 56.2±0.8 | 16.6±4.0 | **60.3±0.4** |
| 2015 | SP | 87.0±0.3 | 81.6±1.3 | 75.2±0.8 | 88.2±0.4 | 84.5±2.8 | 50.1±8.5 | **89.1±0.4** |
|      | DP | 86.9±0.3 | 80.8±1.4 | 74.6±0.7 | 88.0±0.3 | 86.3±1.4 | 49.1±12.6 | **89.3±0.3** |
|      | 3C | 60.1±0.2 | 57.5±0.8 | 51.8±0.8 | 60.1±0.8 | 61.0±1.2 | 32.6±6.2 | **63.8±0.4** |
|      | 4C | 83.1±0.4 | 81.0±0.7 | 69.6±1.0 | **84.7±0.5** | 79.0±1.8 | 25.5±11.1 | 84.4±0.5 |
|      | 5C | 64.6±0.2 | 62.8±0.5 | 55.1±0.3 | **65.9±0.1** | 57.1±5.4 | 16.7±3.4 | 65.7±0.4 |
| 2016 | SP | 87.9±0.5 | 81.6±1.8 | 73.5±0.9 | 89.7±0.5 | 86.5±2.5 | 49.0±6.5 | **90.2±0.5** |
|      | DP | 87.4±0.4 | 81.1±1.4 | 73.1±0.9 | 89.6±0.3 | 87.0±2.7 | 56.2±6.7 | **90.0±0.5** |
|      | 3C | 60.6±0.3 | 57.2±0.5 | 50.9±0.9 | 62.4±0.4 | 59.0±1.3 | 34.4±4.0 | **63.9±0.2** |
|      | 4C | 80.7±0.6 | 77.5±0.8 | 67.5±0.3 | **82.2±0.6** | 74.1±4.8 | 24.3±3.8 | 82.1±0.5 |
|      | 5C | 60.8±0.2 | 58.0±0.8 | 52.0±0.5 | **62.6±0.3** | 54.8±2.7 | 19.7±4.0 | 62.0±0.4 |
| 2017 | SP | 87.7±0.4 | 81.7±1.2 | 78.0±0.4 | 89.8±0.3 | 88.3±1.5 | 57.8±6.8 | **90.2±0.3** |
|      | DP | 87.5±0.3 | 82.0±1.2 | 77.6±0.3 | 89.8±0.2 | 87.6±1.7 | 50.2±4.8 | **90.2±0.3** |
|      | 3C | 59.7±0.4 | 56.2±0.4 | 52.8±0.1 | 61.1±0.3 | 59.9±0.5 | 31.3±4.6 | **62.3±0.4** |
|      | 4C | 68.3±0.5 | 65.4±0.6 | 59.1±0.6 | **70.5±0.3** | 66.4±0.5 | 28.6±7.0 | 69.6±0.4 |
|      | 5C | 51.7±0.6 | 50.1±1.0 | 45.8±0.5 | **53.4±0.4** | 50.1±1.4 | 22.3±3.8 | 53.2±0.1 |
| 2018 | SP | 83.6±0.3 | 78.7±1.1 | 72.6±0.5 | 86.2±0.3 | 84.3±1.7 | 47.9±4.2 | **87.0±0.4** |
|      | DP | 83.7±0.5 | 78.8±0.8 | 72.7±0.7 | 86.3±0.2 | 84.9±1.9 | 44.1±5.0 | **87.0±0.4** |
|      | 3C | 57.7±0.3 | 54.4±0.8 | 50.7±0.5 | 58.9±0.6 | 56.9±2.7 | 33.4±4.0 | **61.4±0.3** |
|      | 4C | 77.0±0.4 | 75.8±0.7 | 65.7±0.7 | **79.5±0.4** | 77.3±0.6 | 23.5±5.7 | **79.5±0.5** |
|      | 5C | 59.4±0.1 | 57.6±0.6 | 51.6±0.7 | **61.0±0.4** | 56.4±1.7 | 21.8±4.8 | 60.5±0.3 |
| 2019 | SP | 88.5±0.4 | 80.6±2.0 | 73.3±0.4 | 90.3±0.3 | 87.6±1.6 | 52.0±10.8 | **90.8±0.3** |
|      | DP | 88.6±0.4 | 81.5±1.0 | 72.6±1.3 | 90.6±0.1 | 88.7±0.9 | 46.4±4.9 | **90.9±0.3** |
|      | 3C | 61.1±0.4 | 56.0±1.5 | 50.1±0.9 | 63.0±0.2 | 59.6±2.3 | 33.2±3.1 | **64.3±0.2** |
|      | 4C | 75.3±0.4 | 71.8±1.2 | 62.1±0.4 | **78.0±0.4** | 74.3±1.1 | 23.1±7.8 | 77.5±0.4 |
|      | 5C | 57.9±0.3 | 54.9±0.8 | 48.1±0.4 | **59.6±0.5** | 54.3±1.5 | 21.5±2.6 | 58.7±0.2 |
| 2020 | SP | 95.7±0.2 | 93.5±0.8 | 90.4±0.5 | 95.6±0.4 | 96.2±0.5 | 50.9±27.3 | **97.3±0.2** |
|      | DP | 95.9±0.1 | 92.6±1.2 | 90.2±0.4 | 95.5±0.4 | 95.7±0.7 | 37.1±13.3 | **97.2±0.1** |
|      | 3C | 81.9±0.3 | 77.8±1.0 | 76.3±0.3 | 81.4±0.2 | 76.0±6.1 | 43.2±8.1 | **82.9±0.5** |
|      | 4C | 95.1±0.2 | 93.0±0.9 | 90.5±0.5 | 94.7±0.3 | 95.6±0.5 | 44.1±20.8 | **96.5±0.1** |
|      | 5C | 82.1±0.2 | 77.8±1.1 | 76.4±0.7 | 81.4±0.4 | 77.8±3.8 | 21.4±16.7 | **82.8±0.4** |

**Table 20:** Full link prediction test accuracy (%) comparison for directions (and signs) on *FiLL-OPCL* data sets on individual years 2000-2010. The best is marked in **bold red** and the second best is marked in _underline blue_. The link prediction tasks are introduced in Sec. 4.1.

| Year | Link Task | SGCN | SDGNN | SiGAT | SNEA | SSSNET | SigMaNet | MSGNN |
|---|---|---|---|---|---|---|---|---|
| 2000 | SP | 85.7±0.4 | 77.1±3.1 | 69.0±1.0 | _87.5±0.5_ | 87.1±0.5 | 49.5±8.1 | **87.9±0.5** |
|  | DP | 85.6±0.2 | 77.2±1.5 | 68.4±1.4 | _87.4±0.6_ | 87.0±0.4 | 53.6±12.4 | **87.9±0.6** |
|  | 3C | 58.6±0.5 | 50.7±1.6 | 46.6±0.8 | _59.8±0.3_ | 58.9±0.8 | 32.9±4.9 | **60.7±0.4** |
|  | 4C | 70.2±0.5 | 65.2±0.8 | 54.7±1.0 | **71.8±0.4** | 68.7±0.9 | 34.8±4.0 | _71.3±0.3_ |
|  | 5C | 52.5±0.5 | 47.9±0.6 | 42.8±0.5 | _53.3±0.7_ | 51.2±0.9 | 22.6±5.2 | **53.4±0.5** |
| 2001 | SP | 87.6±0.3 | 79.6±1.8 | 69.9±1.6 | _89.8±0.3_ | 88.2±0.9 | 60.3±6.6 | **90.0±0.5** |
|  | DP | 87.3±0.5 | 81.2±1.1 | 69.6±0.8 | _89.3±0.5_ | 86.9±2.8 | 44.0±14.9 | **90.2±0.3** |
|  | 3C | 59.4±0.4 | 54.8±0.7 | 47.4±0.3 | _61.0±0.3_ | 60.5±1.2 | 32.5±3.6 | **62.2±0.4** |
|  | 4C | 74.2±0.5 | 69.2±1.7 | 57.5±0.9 | **76.0±0.3** | 70.6±2.6 | 20.8±5.0 | _75.4±0.6_ |
|  | 5C | 54.5±0.2 | 51.1±0.2 | 44.5±0.5 | **55.9±0.1** | 51.4±2.5 | 18.8±3.8 | _55.8±0.4_ |
| 2002 | SP | 87.3±0.4 | 82.4±0.3 | 75.5±0.6 | _89.6±0.2_ | 85.4±3.2 | 49.9±12.3 | **90.5±0.2** |
|  | DP | 87.2±0.2 | 82.4±1.2 | 75.8±0.4 | _89.3±0.3_ | 85.5±3.7 | 49.1±7.1 | **90.5±0.3** |
|  | 3C | 60.9±0.4 | 59.4±0.9 | 53.1±0.8 | _63.0±0.5_ | 60.5±3.2 | 35.9±2.8 | **65.0±0.4** |
|  | 4C | 82.7±0.3 | 81.1±0.3 | 70.9±0.7 | _84.1±0.2_ | 80.7±1.9 | 24.1±7.6 | **84.5±0.5** |
|  | 5C | 63.9±0.4 | 62.6±0.7 | 55.4±0.5 | _65.1±0.2_ | 58.9±2.2 | 23.0±7.0 | **65.5±0.2** |
| 2003 | SP | 86.0±0.6 | 80.2±1.3 | 74.5±0.3 | _87.7±0.3_ | 87.6±0.3 | 47.7±8.0 | **89.1±0.3** |
|  | DP | 85.8±0.4 | 77.1±1.8 | 75.3±0.3 | _87.7±0.4_ | 85.8±2.5 | 50.3±5.9 | **89.2±0.5** |
|  | 3C | 58.4±0.5 | 55.7±1.1 | 51.0±0.6 | _60.2±0.3_ | _60.2±1.1_ | 33.4±2.4 | **62.7±0.4** |
|  | 4C | 80.3±0.5 | 78.2±1.3 | 68.5±0.8 | _81.9±0.5_ | 79.4±0.6 | 24.0±10.1 | **82.5±0.3** |
|  | 5C | 60.7±0.4 | 59.1±0.5 | 52.1±0.4 | _61.8±0.3_ | 58.5±1.3 | 23.7±5.8 | **62.3±0.4** |
| 2004 | SP | 85.2±0.3 | 74.0±2.4 | 71.8±0.8 | _86.8±0.3_ | 86.4±0.9 | 52.3±8.0 | **87.4±0.3** |
|  | DP | 85.4±0.4 | 76.3±2.2 | 71.9±0.8 | _87.2±0.2_ | 86.5±0.8 | 51.0±8.2 | **87.4±0.2** |
|  | 3C | 57.5±0.5 | 50.8±1.7 | 48.6±0.6 | _58.4±0.5_ | 57.0±2.2 | 33.9±3.7 | **60.1±0.7** |
|  | 4C | 76.8±0.5 | 72.5±1.7 | 61.8±0.7 | **78.6±0.3** | 73.0±2.9 | 20.1±6.2 | _78.3±0.3_ |
|  | 5C | 57.6±0.3 | 53.9±0.8 | 47.5±0.5 | **58.4±0.2** | 54.0±2.6 | 18.1±3.6 | _57.9±0.3_ |
| 2005 | SP | 83.8±0.3 | 73.4±1.1 | 71.0±1.0 | _85.4±0.4_ | 85.4±0.4 | 53.4±7.4 | **86.4±0.3** |
|  | DP | 83.7±0.6 | 73.0±2.2 | 71.8±0.5 | 85.4±0.6 | _85.6±0.7_ | 52.9±4.7 | **86.4±0.3** |
|  | 3C | 56.4±0.3 | 49.7±1.3 | 48.1±0.4 | 57.2±0.5 | _57.3±1.1_ | 32.4±1.9 | **59.5±0.6** |
|  | 4C | 77.3±0.3 | 72.9±1.0 | 65.8±0.4 | _79.2±0.5_ | 75.6±1.3 | 21.6±5.9 | **79.5±0.3** |
|  | 5C | 58.2±0.4 | 55.3±0.7 | 51.3±0.7 | **59.4±0.4** | 56.0±2.3 | 12.9±3.1 | _58.8±0.5_ |
| 2006 | SP | 87.7±0.3 | 77.2±3.2 | 73.9±1.2 | _89.2±0.4_ | 88.4±1.0 | 47.9±7.7 | **89.9±0.5** |
|  | DP | 87.6±0.4 | 76.6±2.3 | 74.1±0.7 | _89.1±0.3_ | _89.1±0.3_ | 57.0±11.9 | **90.1±0.4** |
|  | 3C | 59.8±0.4 | 52.5±0.7 | 51.0±0.4 | _61.1±0.5_ | 56.7±2.9 | 33.4±1.5 | **63.0±0.5** |
|  | 4C | 80.5±0.2 | 74.9±1.4 | 67.0±0.6 | **81.8±0.2** | 78.5±1.0 | 24.7±2.8 | _81.6±0.3_ |
|  | 5C | 60.7±0.5 | 55.6±0.8 | 51.4±0.6 | **61.7±0.3** | 58.0±0.8 | 21.7±6.9 | _60.8±0.6_ |
| 2007 | SP | 85.4±0.4 | 77.8±1.3 | 75.3±0.7 | _86.7±0.4_ | 85.7±1.4 | 58.3±7.1 | **88.0±0.2** |
|  | DP | 86.0±0.3 | 77.5±1.5 | 75.5±0.9 | _86.9±0.4_ | 85.9±1.6 | 55.1±10.4 | **88.0±0.3** |
|  | 3C | 59.1±0.6 | 56.4±0.9 | 53.4±0.8 | _61.0±0.2_ | 57.8±4.5 | 30.5±3.2 | **63.7±0.5** |
|  | 4C | 81.6±0.2 | 78.4±0.7 | 69.9±0.7 | _82.5±0.2_ | 79.5±0.7 | 23.1±9.2 | **83.0±0.2** |
|  | 5C | 63.1±0.3 | 60.5±0.5 | 54.9±0.5 | _63.7±0.5_ | 60.5±0.4 | 19.4±8.4 | **64.1±0.4** |
| 2008 | SP | 94.7±0.4 | 92.2±0.6 | 85.3±0.3 | _95.6±0.4_ | 95.1±0.4 | 53.5±15.7 | **96.5±0.3** |
|  | DP | 94.4±0.4 | 93.0±0.9 | 85.4±0.6 | _95.3±0.2_ | 94.6±1.7 | 34.1±11.4 | **96.6±0.2** |
|  | 3C | 74.1±0.3 | 74.1±0.1 | 67.8±0.5 | _75.3±0.3_ | 71.7±4.0 | 36.8±5.9 | **76.7±0.2** |
|  | 4C | 95.0±0.1 | 94.3±0.4 | 88.3±0.8 | **95.6±0.2** | 94.3±0.4 | 11.4±5.2 | _95.4±0.5_ |
|  | 5C | 78.4±0.3 | 77.6±0.3 | 70.6±0.6 | **79.4±0.1** | 76.1±2.7 | 13.6±6.6 | _78.9±0.7_ |
| 2009 | SP | 93.4±0.3 | 83.9±1.9 | 79.4±0.9 | _94.4±0.2_ | 93.7±1.1 | 47.6±9.4 | **95.2±0.2** |
|  | DP | 93.5±0.2 | 84.0±2.2 | 80.0±0.5 | _94.4±0.3_ | 93.7±0.5 | 42.0±11.9 | **95.2±0.2** |
|  | 3C | 67.6±0.3 | 62.6±1.5 | 56.1±0.4 | _69.2±0.3_ | 62.4±4.1 | 38.0±6.2 | **70.5±0.2** |
|  | 4C | 90.3±0.2 | 86.9±0.5 | 80.3±0.3 | _91.1±0.1_ | 88.4±1.4 | 30.5±13.0 | **91.4±0.3** |
|  | 5C | 72.5±0.4 | 68.5±0.4 | 62.7±0.5 | **73.3±0.2** | 68.0±1.4 | 21.7±8.9 | _73.2±0.2_ |
| 2010 | SP | 90.5±0.6 | 85.9±0.6 | 80.4±0.4 | _91.8±0.3_ | 90.3±1.1 | 46.3±11.7 | **92.5±0.3** |
|  | DP | 90.5±0.6 | 85.1±0.6 | 80.0±0.9 | _91.5±0.3_ | 90.4±1.2 | 48.7±6.7 | **92.5±0.4** |
|  | 3C | 63.9±0.4 | 62.6±0.3 | 57.8±0.6 | _65.4±0.3_ | 60.7±2.9 | 33.5±6.7 | **67.1±0.5** |
|  | 4C | 88.3±0.5 | 85.8±1.0 | 78.4±0.3 | _88.8±0.3_ | 85.5±0.8 | 21.5±3.9 | **89.3±0.3** |
|  | 5C | 69.2±0.6 | 67.3±1.0 | 60.6±0.7 | **69.9±0.5** | 65.8±0.7 | 19.8±5.2 | _69.9±0.3_ |

**Table 21:** Full link prediction test accuracy (%) comparison for directions (and signs) on *FiLL-OPCL* data sets on individual years 2011-2020. The best is marked in **bold red** and the second best is marked in underline blue. The link prediction tasks are introduced in Sec. 4.1.

| Year | Link Task | SGCN | SDGNN | SiGAT | SNEA | SSSNET | SigMaNet | MSGNN |
|------|-----------|------|-------|-------|------|--------|----------|-------|
| 2011 | SP | 94.6±0.2 | 92.2±0.6 | 84.2±0.2 | 95.2±0.3 | 95.4±1.1 | 39.3±10.0 | **96.2±0.3** |
|      | DP | 94.9±0.1 | 91.5±1.3 | 83.9±0.6 | 95.4±0.3 | 95.1±1.1 | 35.0±16.7 | **96.3±0.3** |
|      | 3C | 75.8±0.4 | 75.0±0.5 | 70.1±0.4 | 76.6±0.3 | 72.2±6.5 | 37.1±7.9 | **78.6±0.4** |
|      | 4C | 94.4±0.2 | 93.6±0.4 | 85.1±0.4 | 95.0±0.4 | 94.2±0.6 | 33.6±9.6 | **95.5±0.3** |
|      | 5C | 79.7±0.3 | 78.2±0.4 | 71.9±0.4 | 80.2±0.3 | 77.7±1.9 | 17.1±8.7 | **80.6±0.3** |
| 2012 | SP | 89.2±0.3 | 80.9±1.1 | 80.2±0.3 | 90.3±0.2 | 90.3±0.2 | 45.5±12.5 | **91.1±0.2** |
|      | DP | 89.4±0.4 | 82.4±1.0 | 80.2±0.7 | 90.3±0.2 | 89.8±0.5 | 50.1±6.9 | **91.2±0.2** |
|      | 3C | 61.8±0.3 | 57.0±1.2 | 56.2±0.5 | 62.5±0.2 | 62.3±0.9 | 34.5±5.6 | **64.4±0.5** |
|      | 4C | 80.3±0.4 | 75.9±0.6 | 71.2±0.6 | 81.0±0.3 | 79.1±0.8 | 28.9±9.9 | **81.5±0.4** |
|      | 5C | 60.9±0.3 | 57.1±0.5 | 54.5±0.4 | 61.4±0.3 | 57.3±4.0 | 20.1±6.5 | **61.5±0.3** |
| 2013 | SP | 86.5±0.6 | 82.3±1.0 | 79.7±0.3 | 88.1±0.3 | 88.0±0.9 | 52.8±10.0 | **89.5±0.3** |
|      | DP | 86.9±0.2 | 81.5±1.4 | 79.2±0.3 | 87.9±0.3 | 86.9±1.9 | 63.3±13.4 | **89.3±0.2** |
|      | 3C | 61.4±0.4 | 59.4±0.6 | 57.7±0.4 | 61.9±0.2 | 61.4±1.6 | 31.4±11.8 | **64.3±0.3** |
|      | 4C | 82.0±0.3 | 80.3±0.6 | 72.1±0.4 | **83.0±0.2** | 80.3±1.2 | 27.0±16.0 | 82.8±0.8 |
|      | 5C | 62.6±0.3 | 61.3±0.4 | 56.9±0.2 | 63.2±0.4 | 60.9±0.9 | 23.2±6.7 | **63.9±0.4** |
| 2014 | SP | 85.4±0.4 | 76.8±1.9 | 72.3±0.6 | 86.9±0.2 | 86.1±1.2 | 50.4±2.4 | **87.7±0.4** |
|      | DP | 85.2±0.5 | 76.5±0.7 | 72.2±0.4 | 86.7±0.3 | 84.7±2.5 | 51.2±6.0 | **87.9±0.4** |
|      | 3C | 58.3±0.5 | 54.1±1.8 | 50.3±0.2 | 59.7±0.8 | 59.5±0.8 | 37.2±3.9 | **61.9±0.2** |
|      | 4C | 79.4±0.2 | 76.0±0.6 | 68.0±1.0 | **81.3±0.2** | 78.9±0.7 | 21.9±9.0 | 81.2±0.2 |
|      | 5C | 60.4±0.5 | 57.9±0.4 | 53.1±0.4 | 61.7±0.2 | 58.7±1.0 | 18.9±4.5 | **61.8±0.3** |
| 2015 | SP | 87.0±0.4 | 81.5±0.6 | 78.5±0.7 | 88.6±0.3 | 87.0±2.8 | 41.9±7.1 | **89.8±0.3** |
|      | DP | 87.2±0.3 | 81.5±1.0 | 78.7±0.9 | 88.8±0.2 | 85.5±3.0 | 49.8±5.1 | **89.8±0.3** |
|      | 3C | 60.0±0.2 | 59.6±0.6 | 54.4±0.5 | 61.3±0.4 | 59.8±2.6 | 33.9±5.0 | **64.1±0.2** |
|      | 4C | 83.1±0.2 | 80.3±0.5 | 72.9±0.6 | 84.4±0.3 | 80.8±0.9 | 20.9±3.7 | **84.8±0.2** |
|      | 5C | 63.7±0.3 | 62.3±0.6 | 56.4±0.4 | **65.0±0.3** | 59.4±2.5 | 20.7±9.1 | 64.8±0.5 |
| 2016 | SP | 86.4±0.5 | 79.1±0.7 | 75.9±0.6 | 88.0±0.3 | 86.6±1.1 | 58.6±10.2 | **89.0±0.2** |
|      | DP | 86.5±0.5 | 78.2±1.0 | 76.3±0.4 | 88.2±0.5 | 86.6±2.3 | 53.0±2.9 | **88.9±0.3** |
|      | 3C | 59.6±0.6 | 54.1±0.6 | 52.6±0.5 | 60.6±0.3 | 59.0±1.5 | 31.5±3.9 | **62.2±0.5** |
|      | 4C | 74.9±0.4 | 71.0±1.0 | 64.2±0.3 | 76.5±0.3 | 71.5±2.0 | 24.9±3.8 | **76.7±0.5** |
|      | 5C | 56.5±0.4 | 54.0±0.7 | 49.8±0.2 | 57.0±0.2 | 50.7±2.1 | 20.4±6.0 | **58.1±0.3** |
| 2017 | SP | 86.4±0.2 | 79.3±1.9 | 75.8±0.3 | 88.9±0.2 | 87.9±1.0 | 53.2±7.6 | **89.3±0.3** |
|      | DP | 86.3±0.4 | 78.4±1.6 | 75.9±1.1 | 89.1±0.2 | 88.4±0.4 | 45.2±9.4 | **89.5±0.2** |
|      | 3C | 58.6±0.2 | 53.6±0.6 | 51.5±0.2 | 60.4±0.2 | 57.7±1.7 | 30.0±4.1 | **61.3±0.2** |
|      | 4C | 67.0±0.5 | 63.2±1.3 | 56.4±0.4 | 69.3±0.3 | 63.8±2.8 | 26.1±5.3 | **69.4±0.4** |
|      | 5C | 50.2±0.3 | 47.2±0.9 | 43.9±0.3 | 51.7±0.2 | 49.0±0.8 | 19.8±5.2 | **51.8±0.3** |
| 2018 | SP | 84.5±0.6 | 79.3±1.7 | 69.2±0.6 | 87.3±0.4 | 87.0±0.5 | 59.1±13.4 | **88.2±0.4** |
|      | DP | 84.7±0.5 | 77.9±1.0 | 70.3±0.6 | 87.4±0.5 | 87.1±0.5 | 50.8±6.3 | **88.1±0.4** |
|      | 3C | 59.6±0.3 | 55.1±1.3 | 48.6±0.7 | 61.4±0.5 | 59.0±3.0 | 33.2±2.6 | **64.0±0.6** |
|      | 4C | 80.2±0.7 | 76.3±1.4 | 63.4±0.7 | **83.2±0.6** | 78.9±1.2 | 30.4±7.7 | 83.0±0.7 |
|      | 5C | 62.4±0.3 | 58.1±0.7 | 50.5±0.5 | **63.9±0.3** | 58.4±4.9 | 23.2±6.9 | 63.7±0.5 |
| 2019 | SP | 86.4±0.3 | 80.8±1.0 | 77.3±0.3 | 88.5±0.3 | 85.8±2.8 | 41.7±9.4 | **89.3±0.4** |
|      | DP | 86.5±0.3 | 80.0±1.3 | 77.3±0.6 | 88.7±0.2 | 88.1±1.1 | 51.4±8.5 | **89.3±0.2** |
|      | 3C | 59.5±0.4 | 55.0±0.4 | 53.4±0.5 | 60.8±0.2 | 58.9±2.4 | 33.6±5.8 | **62.4±0.5** |
|      | 4C | 71.2±0.5 | 68.1±0.5 | 63.1±0.4 | **74.3±0.3** | 68.9±2.2 | 26.8±9.5 | 74.3±0.4 |
|      | 5C | 54.4±0.3 | 52.0±0.5 | 48.8±0.3 | **56.2±0.3** | 52.7±0.5 | 21.8±7.1 | 56.0±0.2 |
| 2020 | SP | 89.8±0.3 | 84.4±0.7 | 84.6±0.4 | 90.9±0.3 | 90.4±0.7 | 53.9±16.9 | **92.3±0.1** |
|      | DP | 90.1±0.2 | 85.4±0.7 | 84.7±0.4 | 91.0±0.3 | 89.6±1.2 | 53.5±14.0 | **92.1±0.1** |
|      | 3C | 66.8±0.4 | 62.3±0.4 | 62.4±0.4 | 67.8±0.2 | 63.6±4.1 | 41.5±4.6 | **69.2±0.4** |
|      | 4C | 84.0±0.4 | 81.5±0.5 | 78.5±0.4 | **85.1±0.3** | 82.6±1.2 | 16.3±9.3 | 84.8±0.4 |
|      | 5C | 66.8±0.5 | 63.6±0.6 | 61.1±0.2 | **68.1±0.4** | 64.2±1.3 | 24.4±10.1 | 67.3±0.6 |

**Table 22:** Link prediction test performance (accuracy in percentage) comparison for variants of MSGNN for individual years 2000-2010 of the *FiLL-pvCLCL* data set. Each variant is denoted by a $q$ value and a 2-tuple: (whether to include signed features, whether to include weighted features), where "T" and "F" stand for "True" and "False", respectively. "T" for weighted features means simply summing up entries in the adjacency matrix while "T'" means summing the absolute values of the entries. The best is marked in **bold red** and the second best is marked in underline blue.

| Year | Link Task | 0 (F, F) | 0 (F, T) | 0 (F, T') | 0 (T, F) | 0 (T, T) | $q_0 := 1/[2\max_{i,j}(\mathbf{A}_{i,j} - \mathbf{A}_{j,i})]$ (F, F) | (F, T) | (F, T') | (T, F) | (T, T) |
|---|---|---|---|---|---|---|---|---|---|---|---|
| 2000 | SP | 89.0±0.4 | 89.0±0.5 | 89.0±0.4 | 89.0±0.4 | 89.0±0.4 | 88.9±0.6 | 88.9±0.4 | 88.9±0.3 | 88.7±0.4 | 88.8±0.5 |
| | DP | 88.9±0.4 | 89.0±0.5 | 89.0±0.5 | 89.0±0.4 | 89.1±0.4 | 89.1±0.4 | 89.1±0.4 | 89.1±0.4 | 89.1±0.6 | 89.1±0.5 |
| | 3C | 60.7±0.6 | 61.1±0.7 | 61.0±0.5 | 60.7±0.6 | 61.3±0.4 | 60.4±0.4 | 61.0±0.4 | 61.4±0.8 | 60.6±0.6 | 61.5±0.7 |
| | 4C | 68.7±0.8 | 71.0±0.5 | 70.9±0.7 | 70.7±0.6 | 72.3±0.5 | 67.9±2.3 | 70.5±0.7 | 70.8±0.8 | 71.2±0.6 | 71.9±0.5 |
| | 5C | 49.9±2.3 | 53.1±0.2 | 52.7±0.7 | 51.7±0.5 | 54.0±0.4 | 50.4±0.7 | 52.9±0.4 | 52.6±0.5 | 51.4±2.5 | 53.9±0.4 |
| 2001 | SP | 90.6±0.4 | 90.7±0.3 | 90.7±0.1 | 90.3±0.2 | 90.7±0.2 | 90.4±0.2 | 90.5±0.2 | 90.5±0.2 | 90.0±0.5 | 90.7±0.2 |
| | DP | 90.6±0.4 | 90.5±0.2 | 90.6±0.2 | 90.4±0.3 | 90.7±0.2 | 90.7±0.4 | 90.7±0.3 | 90.7±0.2 | 90.6±0.3 | 90.7±0.1 |
| | 3C | 62.0±0.4 | 62.3±0.1 | 62.4±0.2 | 61.9±0.3 | 62.6±0.2 | 62.1±0.3 | 62.6±0.4 | 62.5±0.3 | 62.2±0.4 | 63.1±0.5 |
| | 4C | 71.9±1.0 | 74.6±0.4 | 74.6±0.5 | 74.6±0.6 | 75.8±0.3 | 72.2±1.2 | 74.5±0.7 | 74.4±0.5 | 74.9±0.6 | 75.7±0.3 |
| | 5C | 52.0±0.6 | 55.7±0.3 | 55.3±1.0 | 55.4±1.1 | 56.7±0.4 | 51.5±1.4 | 55.4±0.6 | 56.0±0.3 | 53.7±1.8 | 56.8±0.2 |
| 2002 | SP | 91.4±0.3 | 91.4±0.2 | 91.5±0.2 | 91.4±0.2 | 91.4±0.2 | 91.1±0.3 | 91.2±0.1 | 91.2±0.2 | 91.0±0.3 | 91.2±0.3 |
| | DP | 91.1±0.5 | 91.4±0.2 | 91.5±0.2 | 91.3±0.2 | 91.5±0.1 | 91.5±0.3 | 91.3±0.1 | 91.5±0.2 | 91.3±0.2 | 91.5±0.2 |
| | 3C | 64.7±0.3 | 65.5±0.5 | 65.5±0.5 | 64.6±0.6 | 66.2±0.4 | 64.2±0.7 | 65.7±0.3 | 65.4±0.4 | 65.2±0.1 | 66.2±0.2 |
| | 4C | 83.4±1.1 | 84.4±0.3 | 84.7±0.4 | 84.4±0.6 | 85.7±0.5 | 83.3±1.0 | 84.7±0.2 | 84.6±0.4 | 84.6±0.8 | 85.7±0.4 |
| | 5C | 60.8±4.3 | 65.5±0.4 | 65.6±0.3 | 65.6±0.5 | 66.8±0.4 | 63.7±1.1 | 65.8±0.3 | 65.2±0.2 | 64.8±0.8 | 66.7±0.4 |
| 2003 | SP | 89.3±0.5 | 89.3±0.3 | 89.4±0.3 | 89.3±0.4 | 89.5±0.4 | 89.3±0.3 | 89.3±0.3 | 89.3±0.4 | 88.9±0.4 | 89.3±0.4 |
| | DP | 89.4±0.4 | 89.4±0.4 | 89.5±0.4 | 89.4±0.5 | 89.6±0.4 | 89.5±0.5 | 89.4±0.2 | 89.5±0.4 | 89.6±0.3 | 89.6±0.4 |
| | 3C | 60.2±1.0 | 62.4±0.8 | 62.3±1.0 | 61.4±0.6 | 63.2±0.5 | 61.2±1.1 | 62.9±0.7 | 62.9±0.5 | 61.7±0.4 | 63.1±0.5 |
| | 4C | 80.3±0.6 | 81.9±0.2 | 81.9±0.3 | 81.5±0.6 | 82.6±0.4 | 79.7±0.8 | 81.8±0.5 | 81.6±0.2 | 81.4±0.9 | 82.7±0.4 |
| | 5C | 57.0±2.0 | 61.9±0.3 | 61.5±0.5 | 60.0±0.6 | 62.6±0.2 | 57.4±1.4 | 61.4±0.3 | 61.5±0.3 | 61.2±0.5 | 62.7±0.4 |
| 2004 | SP | 88.6±0.4 | 88.6±0.4 | 88.7±0.2 | 88.6±0.2 | 88.7±0.3 | 88.4±0.6 | 88.7±0.3 | 88.6±0.3 | 88.3±0.3 | 88.8±0.3 |
| | DP | 88.5±0.2 | 88.8±0.3 | 88.8±0.2 | 88.7±0.3 | 88.8±0.3 | 88.8±0.3 | 88.7±0.4 | 88.7±0.3 | 88.7±0.2 | 88.8±0.3 |
| | 3C | 60.2±0.7 | 61.4±0.4 | 61.3±0.6 | 60.5±0.3 | 61.6±0.5 | 59.4±1.0 | 61.6±0.5 | 61.3±0.3 | 60.4±0.5 | 61.6±0.4 |
| | 4C | 75.5±0.9 | 78.0±0.3 | 77.7±0.6 | 77.3±0.6 | 78.8±0.6 | 75.9±0.5 | 77.5±0.7 | 77.6±0.5 | 77.4±0.4 | 78.7±0.4 |
| | 5C | 55.5±0.7 | 58.1±0.6 | 58.0±0.3 | 55.8±2.8 | 58.7±0.3 | 53.8±0.8 | 57.3±0.5 | 57.5±0.3 | 55.5±2.4 | 58.7±0.6 |
| 2005 | SP | 87.7±0.4 | 87.7±0.5 | 87.6±0.5 | 87.5±0.5 | 87.8±0.4 | 87.5±0.4 | 87.7±0.6 | 87.6±0.5 | 87.3±0.5 | 87.7±0.4 |
| | DP | 87.6±0.5 | 87.8±0.4 | 87.7±0.6 | 87.5±0.5 | 87.9±0.4 | 87.3±0.8 | 87.8±0.4 | 87.8±0.5 | 87.8±0.4 | 87.8±0.4 |
| | 3C | 59.2±0.7 | 60.3±0.3 | 60.3±0.5 | 59.6±0.8 | 61.2±0.2 | 59.3±0.4 | 60.8±0.5 | 60.7±0.6 | 59.6±0.3 | 60.9±0.6 |
| | 4C | 78.2±0.4 | 79.7±0.3 | 79.7±0.4 | 79.0±1.2 | 80.8±0.2 | 78.1±0.4 | 79.1±0.5 | 79.4±0.3 | 80.3±0.3 | 80.5±0.2 |
| | 5C | 57.1±1.3 | 60.1±0.3 | 60.1±0.3 | 59.3±0.9 | 61.3±0.4 | 55.1±1.6 | 59.4±0.5 | 59.4±0.4 | 59.6±0.4 | 61.0±0.2 |
| 2006 | SP | 90.9±0.2 | 90.9±0.1 | 91.0±0.1 | 90.9±0.2 | 91.0±0.2 | 90.5±0.2 | 90.6±0.3 | 90.5±0.4 | 90.5±0.1 | 90.6±0.1 |
| | DP | 90.8±0.2 | 90.9±0.1 | 91.0±0.2 | 91.0±0.2 | 91.1±0.2 | 91.0±0.1 | 91.2±0.1 | 91.0±0.2 | 91.0±0.1 | 91.1±0.1 |
| | 3C | 63.2±0.3 | 63.4±0.4 | 63.5±0.3 | 63.2±0.3 | 64.0±0.3 | 61.8±2.0 | 64.1±0.4 | 64.0±0.4 | 63.0±0.4 | 64.1±0.3 |
| | 4C | 80.1±0.8 | 81.2±0.3 | 81.4±0.6 | 81.5±0.7 | 82.9±0.2 | 79.8±0.5 | 81.0±0.7 | 81.4±0.5 | 81.7±0.9 | 83.1±0.4 |
| | 5C | 58.1±0.7 | 61.6±0.3 | 61.6±0.5 | 61.3±0.7 | 62.6±0.3 | 59.4±1.1 | 61.1±0.7 | 61.4±0.3 | 61.5±0.7 | 62.8±0.3 |
| 2007 | SP | 90.2±0.4 | 90.3±0.3 | 90.3±0.4 | 90.4±0.3 | 90.4±0.5 | 89.7±0.1 | 89.9±0.2 | 90.0±0.2 | 89.6±0.4 | 90.0±0.4 |
| | DP | 90.3±0.3 | 90.4±0.3 | 90.2±0.3 | 90.3±0.4 | 90.4±0.3 | 90.2±0.4 | 90.4±0.3 | 90.3±0.3 | 90.4±0.4 | 90.6±0.4 |
| | 3C | 66.1±1.4 | 68.2±0.4 | 68.4±0.2 | 67.0±0.4 | 69.0±0.4 | 64.8±1.0 | 69.0±0.4 | 68.3±0.7 | 65.1±2.2 | 69.0±0.3 |
| | 4C | 85.3±0.5 | 87.3±0.3 | 86.9±0.5 | 87.4±0.6 | 88.1±0.2 | 85.5±0.4 | 87.3±0.1 | 86.7±0.7 | 86.9±0.8 | 88.4±0.2 |
| | 5C | 66.4±1.5 | 69.3±0.4 | 69.3±0.3 | 68.1±1.0 | 69.9±0.5 | 64.2±0.6 | 68.1±1.0 | 68.3±0.6 | 68.2±1.2 | 69.7±0.2 |
| 2008 | SP | 96.1±0.1 | 96.2±0.2 | 96.1±0.2 | 96.2±0.3 | 96.4±0.2 | 95.4±0.2 | 95.6±0.2 | 95.5±0.2 | 95.5±0.1 | 95.7±0.3 |
| | DP | 96.3±0.1 | 96.3±0.1 | 96.1±0.2 | 96.2±0.3 | 96.4±0.1 | 96.2±0.1 | 96.4±0.1 | 96.3±0.2 | 96.2±0.2 | 96.3±0.2 |
| | 3C | 76.7±1.3 | 78.8±0.3 | 78.7±0.5 | 77.7±0.7 | 79.3±0.7 | 76.0±1.5 | 78.2±0.3 | 78.4±0.6 | 76.9±0.7 | 78.9±0.2 |
| | 4C | 94.5±1.1 | 95.6±0.2 | 95.4±0.3 | 95.9±0.3 | 96.1±0.2 | 93.6±0.8 | 95.1±0.4 | 94.4±1.3 | 95.3±0.2 | 96.2±0.3 |
| | 5C | 78.7±0.5 | 81.1±0.5 | 80.8±0.5 | 79.3±1.0 | 82.4±0.4 | 78.1±0.8 | 79.4±0.8 | 79.7±0.5 | 79.9±0.7 | 82.0±0.4 |
| 2009 | SP | 97.5±0.2 | 97.5±0.2 | 97.6±0.2 | 97.7±0.2 | 97.8±0.2 | 96.9±0.1 | 96.7±0.2 | 96.6±0.7 | 97.0±0.4 | 97.1±0.3 |
| | DP | 97.5±0.2 | 97.6±0.1 | 97.6±0.2 | 97.6±0.1 | 97.8±0.1 | 97.5±0.2 | 97.6±0.2 | 97.6±0.2 | 97.5±0.2 | 97.6±0.1 |
| | 3C | 75.9±1.2 | 77.2±0.7 | 78.3±0.2 | 76.9±0.4 | 78.4±0.4 | 76.5±1.0 | 77.8±0.6 | 77.7±0.4 | 77.5±0.3 | 78.6±0.3 |
| | 4C | 93.5±0.5 | 94.6±0.2 | 94.7±0.2 | 94.0±0.5 | 95.2±0.3 | 93.7±0.3 | 94.6±0.2 | 94.4±0.3 | 94.2±0.2 | 95.0±0.2 |
| | 5C | 77.9±0.5 | 79.2±0.2 | 79.4±0.6 | 78.3±0.6 | 79.9±0.2 | 76.6±0.6 | 78.1±0.8 | 77.4±1.6 | 78.2±0.6 | 79.8±0.2 |
| 2010 | SP | 92.4±0.4 | 92.7±0.3 | 92.6±0.3 | 92.7±0.4 | 92.8±0.3 | 92.1±0.3 | 92.5±0.2 | 92.5±0.3 | 92.0±0.4 | 92.3±0.3 |
| | DP | 92.5±0.3 | 92.7±0.2 | 92.7±0.3 | 92.7±0.4 | 92.6±0.3 | 92.7±0.4 | 92.8±0.4 | 92.6±0.3 | 92.6±0.3 | 92.7±0.4 |
| | 3C | 66.3±0.4 | 67.9±0.3 | 67.6±0.6 | 66.3±1.0 | 68.3±0.2 | 66.2±0.7 | 67.9±0.2 | 67.7±0.5 | 65.7±1.6 | 68.5±0.4 |
| | 4C | 88.8±0.6 | 90.4±0.2 | 90.4±0.3 | 90.1±0.5 | 90.9±0.4 | 89.2±0.8 | 90.0±0.4 | 90.3±0.4 | 90.3±0.2 | 91.1±0.4 |
| | 5C | 67.5±1.3 | 71.4±0.3 | 71.4±0.5 | 70.1±1.2 | 72.5±0.4 | 68.5±2.3 | 71.1±0.5 | 71.1±0.6 | 71.5±0.2 | 72.4±0.1 |

**Table 23:** Link prediction test performance (accuracy in percentage) comparison for variants of MSGNN for individual years 2011-2020 of the *FiLL-pvCLCL* data set. Each variant is denoted by a $q$ value and a 2-tuple: (whether to include signed features, whether to include weighted features), where "T" and "F" stand for "True" and "False", respectively. "T" for weighted features means simply summing up entries in the adjacency matrix while "T'" means summing the absolute values of the entries. The best is marked in **bold red** and the second best is marked in underline blue.

| Year | Link Task | 0 | | | | | $q_0 := 1/[2\max_{i,j}(\mathbf{A}_{i,j} - \mathbf{A}_{j,i})]$ | | | | |
| --- | --- | (F, F) | (F, T) | (F, T') | (T, F) | (T, T) | (F, F) | (F, T) | (F, T') | (T, F) | (T, T) |
| 2011 | SP | 98.4±0.3 | **98.7±0.2** | 98.6±0.1 | 98.6±0.1 | **98.7±0.2** | 97.8±0.5 | 98.0±0.2 | 98.0±0.2 | 98.3±0.2 | 98.3±0.3 |
| | DP | 98.6±0.2 | 98.5±0.3 | 98.5±0.3 | 98.6±0.1 | 98.7±0.2 | 98.5±0.1 | 98.7±0.2 | 98.6±0.1 | **98.8±0.1** | 98.7±0.1 |
| | 3C | 84.5±1.4 | **86.7±0.2** | 86.4±0.4 | 84.0±1.2 | 86.5±0.2 | 83.8±1.1 | 85.5±1.5 | 85.9±0.4 | 84.2±0.8 | 86.2±0.3 |
| | 4C | 97.7±0.6 | 98.2±0.3 | 98.1±0.2 | 98.1±0.2 | **98.3±0.3** | 97.7±0.2 | 98.1±0.2 | 98.0±0.3 | 97.9±0.4 | **98.3±0.2** |
| | 5C | 85.3±0.8 | 87.0±0.4 | 87.4±0.4 | 86.0±0.9 | **87.4±0.3** | 83.2±2.3 | 86.6±0.4 | 86.8±0.5 | 85.6±0.8 | 87.2±0.4 |
| 2012 | SP | 92.4±0.3 | 92.6±0.3 | 92.6±0.4 | 92.5±0.4 | **92.7±0.3** | 92.4±0.4 | 92.5±0.3 | 92.4±0.3 | 92.2±0.5 | 92.4±0.3 |
| | DP | 92.5±0.4 | 92.4±0.4 | 92.5±0.4 | 92.5±0.4 | 92.6±0.2 | 92.5±0.2 | 92.5±0.4 | 92.5±0.4 | 92.4±0.2 | **92.6±0.3** |
| | 3C | 65.8±0.3 | 66.6±0.3 | 66.5±0.1 | 66.5±0.5 | 67.1±0.2 | 65.2±1.1 | 66.5±0.2 | 66.8±0.5 | 66.2±0.2 | **67.4±0.3** |
| | 4C | 84.9±0.7 | 85.9±0.7 | 85.9±0.4 | 85.6±0.8 | **87.0±0.5** | 84.4±0.7 | 86.1±0.3 | 86.1±0.6 | 85.8±0.6 | 86.9±0.4 |
| | 5C | 63.4±0.5 | 65.7±0.3 | 65.5±0.4 | 65.7±0.9 | 66.8±0.1 | 62.8±1.2 | 65.6±0.5 | 66.0±0.3 | 64.4±1.5 | **67.0±0.2** |
| 2013 | SP | 90.3±0.3 | 90.3±0.2 | 90.3±0.4 | 90.4±0.2 | **90.5±0.3** | 90.0±0.2 | 90.1±0.2 | 90.3±0.3 | 89.8±0.3 | 90.3±0.1 |
| | DP | 90.2±0.4 | 90.4±0.3 | 90.3±0.3 | 90.3±0.3 | 90.3±0.3 | 90.2±0.2 | 90.5±0.3 | 90.5±0.3 | 90.5±0.2 | 90.4±0.3 |
| | 3C | 64.7±0.7 | 66.3±0.2 | 66.5±0.3 | 65.4±0.5 | **66.9±0.2** | 62.0±2.6 | 65.9±0.5 | 66.0±0.5 | 64.4±0.9 | 66.1±0.3 |
| | 4C | 83.4±0.8 | 85.0±0.2 | 85.0±0.2 | 85.3±0.6 | 85.9±0.3 | 83.4±1.0 | 84.3±1.0 | 84.9±0.1 | 85.3±0.6 | **86.1±0.3** |
| | 5C | 62.5±1.9 | 66.2±0.3 | 66.3±0.3 | 65.3±0.7 | **67.0±0.3** | 62.1±0.7 | 65.6±0.4 | 65.6±0.6 | 64.5±1.0 | 66.8±0.3 |
| 2014 | SP | **87.3±0.3** | 87.3±0.3 | 87.3±0.2 | 87.2±0.2 | **87.3±0.2** | 86.9±0.3 | 87.1±0.2 | 87.0±0.2 | 86.8±0.4 | 87.1±0.2 |
| | DP | 87.1±0.4 | 87.1±0.3 | 87.2±0.3 | 87.2±0.2 | **87.3±0.2** | 87.1±0.2 | 87.0±0.1 | 87.2±0.2 | 87.2±0.3 | 87.2±0.3 |
| | 3C | 59.6±0.6 | 60.3±0.2 | 60.4±0.1 | 59.8±0.5 | **60.8±0.3** | 59.4±0.2 | 60.2±0.2 | 60.3±0.4 | 59.7±0.6 | 60.6±0.2 |
| | 4C | 77.5±0.7 | 79.0±0.2 | 79.0±0.2 | 79.0±0.8 | **80.2±0.3** | 77.5±0.6 | 79.1±0.3 | 79.0±0.4 | 79.0±0.4 | **80.2±0.2** |
| | 5C | 57.0±0.6 | 59.2±0.2 | 59.3±0.3 | 58.9±1.0 | **60.3±0.4** | 56.8±0.9 | 59.6±0.4 | 59.3±0.2 | 57.7±1.2 | **60.3±0.4** |
| 2015 | SP | 89.0±0.3 | **89.2±0.3** | 89.1±0.4 | 89.1±0.3 | 89.1±0.4 | 88.9±0.2 | 88.7±0.2 | 88.7±0.2 | 88.7±0.3 | 88.8±0.2 |
| | DP | 89.2±0.3 | 89.2±0.3 | 89.1±0.4 | **89.3±0.4** | 89.2±0.4 | 89.0±0.2 | 89.2±0.4 | 89.2±0.4 | 89.2±0.4 | **89.3±0.3** |
| | 3C | 62.3±0.4 | 63.4±0.3 | 63.2±0.3 | 63.0±0.7 | 63.5±0.5 | 62.5±0.4 | 63.3±0.2 | 63.4±0.3 | 62.9±0.3 | **63.8±0.4** |
| | 4C | 82.3±0.5 | 83.5±0.5 | 83.2±0.6 | 84.0±0.7 | **84.6±0.5** | 81.7±1.2 | 83.3±0.4 | 83.6±0.3 | 84.1±0.4 | 84.4±0.5 |
| | 5C | 61.2±1.5 | 64.3±0.4 | 63.8±0.5 | 63.2±1.2 | 65.6±0.4 | 61.4±0.8 | 64.2±0.4 | 64.0±0.4 | 63.8±0.8 | **65.7±0.4** |
| 2016 | SP | 90.0±0.4 | 89.9±0.4 | 90.0±0.4 | 90.1±0.4 | **90.2±0.5** | 89.9±0.2 | 89.2±1.2 | 89.7±0.4 | 89.8±0.3 | 89.7±0.5 |
| | DP | 89.9±0.6 | 90.0±0.4 | 89.9±0.4 | **90.1±0.5** | **90.1±0.5** | 90.1±0.5 | 90.0±0.4 | 90.0±0.5 | **90.1±0.4** | 90.0±0.5 |
| | 3C | 62.9±0.5 | 63.7±0.1 | 63.6±0.4 | 62.4±0.8 | **64.0±0.3** | 62.5±0.7 | 63.5±0.4 | 63.5±0.2 | 62.8±0.7 | 63.9±0.2 |
| | 4C | 78.9±0.5 | 80.7±0.5 | 80.5±0.6 | 80.9±0.3 | **82.0±0.5** | 77.9±2.2 | 80.8±0.5 | 80.6±0.6 | 81.0±1.1 | 82.1±0.5 |
| | 5C | 56.9±2.5 | 60.8±0.5 | 60.8±0.4 | 60.8±1.0 | **62.2±0.3** | 57.8±1.7 | 61.1±0.3 | 61.1±0.3 | 60.8±0.7 | 62.0±0.4 |
| 2017 | SP | 90.1±0.5 | 89.9±0.5 | 90.1±0.4 | **90.2±0.3** | **90.2±0.3** | 89.9±0.5 | 89.6±1.2 | 90.1±0.4 | 89.6±0.6 | 90.0±0.3 |
| | DP | 90.0±0.4 | 90.1±0.4 | 90.1±0.3 | 90.1±0.3 | 90.1±0.2 | **90.2±0.4** | **90.2±0.4** | 90.1±0.3 | **90.2±0.4** | **90.2±0.3** |
| | 3C | 61.4±0.6 | 62.0±0.4 | 62.1±0.3 | 61.6±0.2 | 62.0±0.1 | 61.1±0.8 | 62.3±0.4 | 62.4±0.5 | 62.2±0.5 | 62.3±0.4 |
| | 4C | 65.4±0.8 | 68.2±0.5 | 68.5±0.4 | 68.6±0.3 | **70.1±0.5** | 66.4±1.3 | 68.7±0.4 | 68.8±0.2 | 68.7±0.7 | 69.6±0.4 |
| | 5C | 50.1±0.5 | 52.3±0.4 | 52.8±0.4 | 51.5±0.7 | **53.4±0.2** | 46.0±4.7 | 52.6±0.5 | 52.6±0.2 | 51.3±0.8 | 53.2±0.1 |
| 2018 | SP | 86.9±0.4 | 86.8±0.5 | 86.9±0.4 | 86.8±0.4 | **87.0±0.4** | 86.6±0.7 | 86.5±0.3 | 86.5±0.4 | 86.5±0.5 | 86.6±0.8 |
| | DP | 86.9±0.5 | 86.8±0.4 | 86.8±0.4 | 86.7±0.4 | 86.9±0.4 | 86.7±0.3 | 86.9±0.4 | 86.9±0.5 | 86.8±0.5 | **87.0±0.4** |
| | 3C | 59.7±1.3 | 60.9±0.5 | 61.0±0.6 | 60.3±0.5 | **61.5±0.4** | 60.0±0.5 | 61.3±0.3 | 61.5±0.5 | 60.8±0.3 | 61.4±0.3 |
| | 4C | 75.8±1.2 | 78.4±0.5 | 78.5±0.6 | 78.4±1.2 | **79.7±0.4** | 75.4±3.1 | 78.2±0.4 | 78.1±0.6 | 78.0±0.9 | 79.5±0.5 |
| | 5C | 57.5±0.8 | 59.2±0.4 | 59.3±0.2 | 58.7±1.9 | **60.9±0.4** | 56.2±1.4 | 59.4±0.2 | 59.3±0.8 | 59.1±0.5 | 60.5±0.3 |
| 2019 | SP | 90.8±0.3 | 90.8±0.4 | 90.8±0.3 | **90.9±0.3** | 90.8±0.3 | 90.6±0.5 | 90.6±0.4 | 90.6±0.4 | 90.4±0.1 | 90.6±0.4 |
| | DP | 90.8±0.3 | 90.8±0.3 | 90.8±0.2 | 90.7±0.3 | **90.9±0.3** | 90.6±0.3 | 90.8±0.3 | 90.7±0.3 | **90.9±0.3** | **90.9±0.3** |
| | 3C | 63.1±0.7 | 63.9±0.3 | 64.0±0.2 | 63.4±1.2 | **64.6±0.2** | 63.5±0.3 | 64.2±0.2 | 64.2±0.2 | 63.4±0.6 | 64.3±0.2 |
| | 4C | 74.3±0.8 | 76.1±0.3 | 76.1±0.3 | 77.1±0.6 | **77.9±0.3** | 75.0±0.4 | 76.3±0.5 | 76.5±0.4 | 77.1±0.3 | 77.5±0.4 |
| | 5C | 55.2±1.8 | 58.0±0.3 | 57.9±0.5 | 57.8±0.8 | **59.0±0.2** | 54.5±2.9 | 57.8±0.4 | 57.6±0.5 | 55.6±1.9 | 58.7±0.2 |
| 2020 | SP | 97.1±0.2 | 97.1±0.2 | 97.2±0.1 | **97.3±0.2** | **97.3±0.2** | 96.4±0.2 | 96.4±0.1 | 96.4±0.2 | 96.7±0.1 | 96.6±0.2 |
| | DP | **97.3±0.1** | 97.2±0.1 | 97.1±0.1 | **97.3±0.2** | **97.3±0.2** | 97.2±0.1 | 97.1±0.2 | 97.1±0.2 | 97.1±0.1 | 97.2±0.1 |
| | 3C | 81.8±0.6 | **83.2±0.1** | 83.1±0.2 | 81.6±1.5 | 82.8±1.3 | 80.1±1.4 | 82.8±0.6 | 82.6±0.6 | 82.2±0.4 | 82.9±0.5 |
| | 4C | 96.2±0.2 | 96.2±0.2 | **96.5±0.2** | 96.4±0.5 | 96.4±0.2 | 96.0±0.2 | 96.1±0.6 | 96.1±0.2 | 96.2±0.2 | **96.5±0.1** |
| | 5C | 80.3±1.6 | 82.9±0.3 | 82.9±0.3 | 82.1±0.5 | **83.2±0.4** | 78.8±3.8 | 82.6±0.6 | 82.7±0.2 | 82.1±0.5 | 82.8±0.4 |

**Table 24:** Link prediction test performance (accuracy in percentage) comparison for variants of MSGNN for individual years 2000-2010 of the *FiLL-OPCL* data set. Each variant is denoted by a $q$ value and a 2-tuple: (whether to include signed features, whether to include weighted features), where "T" and "F" stand for "True" and "False", respectively. "T" for weighted features means simply summing up entries in the adjacency matrix while "T'" means summing the absolute values of the entries. The best is marked in **bold red** and the second best is marked in underline blue.

| Year | Link Task | (F, F) | (F, T) | 0 (F, T') | (T, F) | (T, T) | (F, F) | $q_0 := 1/[2\max_{i,j}(\mathbf{A}_{i,j} - \mathbf{A}_{j,i})]$ (F, T) | (F, T') | (T, F) | (T, T) |
|---|---|---|---|---|---|---|---|---|---|---|---|
| 2000 | SP | 87.5±0.5 | 87.6±0.5 | 87.7±0.5 | 87.7±0.6 | 87.9±0.5 | 87.2±0.5 | 87.6±0.6 | 87.6±0.5 | 87.4±0.5 | 87.6±0.5 |
|  | DP | 87.6±0.6 | 87.6±0.6 | 87.5±0.5 | 87.7±0.5 | 87.9±0.5 | 87.6±0.6 | 87.8±0.5 | 87.9±0.6 | 87.7±0.4 | 87.9±0.6 |
|  | 3C | 59.5±0.9 | 60.0±0.4 | 60.2±0.5 | 59.6±0.8 | 60.7±0.3 | 59.4±1.2 | 60.6±0.6 | 60.6±0.4 | 59.7±0.5 | 60.7±0.4 |
|  | 4C | 68.4±1.2 | 69.9±0.4 | 70.1±0.3 | 70.3±0.8 | 71.3±0.3 | 68.1±0.8 | 70.0±0.6 | 70.0±0.5 | 70.5±0.5 | 71.3±0.3 |
|  | 5C | 49.8±2.1 | 52.3±0.4 | 52.4±0.5 | 51.1±1.0 | 53.5±0.4 | 49.0±1.6 | 52.4±0.6 | 52.1±0.3 | 51.6±1.0 | 53.4±0.5 |
| 2001 | SP | 89.7±0.4 | 89.9±0.4 | 89.9±0.3 | 89.9±0.3 | 90.0±0.5 | 89.7±0.5 | 89.8±0.4 | 89.8±0.3 | 89.4±0.4 | 89.9±0.4 |
|  | DP | 89.9±0.3 | 89.9±0.4 | 90.0±0.3 | 89.9±0.3 | 90.0±0.3 | 89.8±0.4 | 90.1±0.3 | 90.0±0.4 | 89.9±0.3 | 90.2±0.3 |
|  | 3C | 60.4±0.5 | 61.6±0.4 | 61.5±0.4 | 61.2±0.6 | 61.9±0.3 | 61.1±0.3 | 61.8±0.5 | 62.0±0.4 | 61.2±0.5 | 62.2±0.4 |
|  | 4C | 71.9±0.6 | 74.3±0.7 | 74.1±0.6 | 74.5±0.9 | 75.5±0.3 | 71.9±1.4 | 74.3±0.7 | 74.0±0.9 | 73.6±1.4 | 75.4±0.6 |
|  | 5C | 52.1±0.7 | 54.7±0.5 | 54.6±0.4 | 53.7±0.6 | 55.8±0.4 | 49.4±2.0 | 54.8±0.7 | 54.6±0.6 | 54.7±0.8 | 55.8±0.4 |
| 2002 | SP | 90.3±0.2 | 90.4±0.3 | 90.4±0.2 | 89.9±0.3 | 90.5±0.2 | 89.7±0.7 | 90.2±0.3 | 90.2±0.3 | 89.6±0.3 | 90.0±0.4 |
|  | DP | 90.2±0.3 | 90.3±0.3 | 90.3±0.4 | 90.3±0.2 | 90.3±0.2 | 90.4±0.3 | 90.4±0.3 | 90.4±0.2 | 90.3±0.2 | 90.5±0.3 |
|  | 3C | 63.0±1.1 | 64.7±0.3 | 64.5±0.5 | 64.4±0.4 | 65.1±0.7 | 62.8±1.0 | 64.6±0.6 | 64.6±0.7 | 64.0±0.4 | 65.0±0.4 |
|  | 4C | 82.3±0.7 | 83.0±0.5 | 83.3±0.6 | 83.5±0.6 | 84.5±0.5 | 82.4±0.8 | 83.4±0.5 | 83.2±0.7 | 83.4±0.5 | 84.5±0.5 |
|  | 5C | 61.8±1.1 | 64.2±0.3 | 64.4±0.4 | 64.4±0.7 | 65.4±0.3 | 60.4±3.0 | 64.5±0.3 | 64.7±0.2 | 64.4±1.1 | 65.5±0.2 |
| 2003 | SP | 88.8±0.2 | 89.1±0.4 | 88.9±0.2 | 88.9±0.2 | 89.1±0.3 | 88.7±0.1 | 88.9±0.2 | 88.9±0.2 | 88.6±0.3 | 88.9±0.3 |
|  | DP | 89.1±0.2 | 89.1±0.4 | 89.0±0.2 | 88.8±0.4 | 89.2±0.5 | 88.9±0.1 | 89.0±0.5 | 89.1±0.4 | 89.0±0.4 | 89.2±0.5 |
|  | 3C | 60.9±0.3 | 61.9±0.2 | 62.0±0.2 | 61.3±0.5 | 62.5±0.5 | 59.7±2.3 | 62.4±0.4 | 62.4±0.4 | 61.2±0.7 | 62.7±0.4 |
|  | 4C | 80.1±0.8 | 81.8±0.3 | 81.6±0.4 | 81.4±0.7 | 82.2±0.4 | 80.6±0.5 | 81.3±0.6 | 81.6±0.6 | 81.5±0.7 | 82.5±0.3 |
|  | 5C | 57.6±1.8 | 61.5±0.5 | 61.6±0.4 | 60.4±1.5 | 62.4±0.3 | 55.8±4.4 | 61.2±0.5 | 61.4±0.3 | 61.1±0.7 | 62.3±0.4 |
| 2004 | SP | 87.5±0.4 | 87.4±0.3 | 87.4±0.3 | 87.5±0.5 | 87.4±0.3 | 87.4±0.4 | 87.5±0.4 | 87.4±0.3 | 87.2±0.3 | 87.4±0.4 |
|  | DP | 87.4±0.2 | 87.2±0.5 | 87.5±0.3 | 87.3±0.3 | 87.4±0.3 | 87.4±0.3 | 87.4±0.3 | 87.4±0.3 | 87.3±0.4 | 87.4±0.2 |
|  | 3C | 59.4±0.3 | 59.9±0.4 | 60.0±0.3 | 59.2±0.8 | 60.3±0.4 | 59.0±0.5 | 60.3±0.6 | 60.2±0.3 | 59.6±0.5 | 60.1±0.7 |
|  | 4C | 76.3±0.6 | 77.7±0.4 | 77.7±0.5 | 77.3±1.1 | 78.3±0.3 | 75.9±0.6 | 77.6±0.3 | 76.9±0.4 | 77.7±0.5 | 78.3±0.3 |
|  | 5C | 55.2±0.8 | 57.1±0.3 | 57.4±0.2 | 56.6±0.7 | 57.9±0.4 | 55.3±0.6 | 56.7±0.8 | 56.7±0.5 | 55.9±0.4 | 57.9±0.3 |
| 2005 | SP | 86.2±0.4 | 86.3±0.4 | 86.3±0.3 | 86.1±0.7 | 86.4±0.3 | 85.9±0.6 | 86.2±0.4 | 86.3±0.4 | 85.7±0.4 | 86.1±0.4 |
|  | DP | 86.3±0.4 | 86.5±0.3 | 86.5±0.4 | 86.1±0.4 | 86.4±0.3 | 86.5±0.4 | 86.3±0.4 | 86.4±0.4 | 86.3±0.4 | 86.4±0.3 |
|  | 3C | 58.3±0.8 | 59.0±0.4 | 59.2±0.5 | 58.7±0.6 | 59.5±0.6 | 58.5±0.5 | 59.7±0.4 | 59.3±0.4 | 58.6±0.4 | 59.5±0.6 |
|  | 4C | 76.2±1.1 | 78.6±0.2 | 78.5±0.2 | 78.2±0.6 | 79.6±0.2 | 77.3±0.9 | 78.5±0.4 | 78.4±0.6 | 78.8±0.4 | 79.5±0.3 |
|  | 5C | 54.3±1.8 | 58.1±0.4 | 58.0±0.4 | 57.0±0.6 | 59.0±0.4 | 53.8±2.1 | 57.8±0.5 | 57.8±0.3 | 57.1±0.8 | 58.8±0.5 |
| 2006 | SP | 89.8±0.4 | 89.6±0.4 | 89.9±0.4 | 89.8±0.4 | 89.9±0.5 | 89.5±0.4 | 89.6±0.4 | 89.7±0.4 | 89.3±0.6 | 89.7±0.4 |
|  | DP | 89.7±0.4 | 89.6±0.5 | 89.8±0.4 | 89.6±0.7 | 89.9±0.4 | 89.8±0.5 | 89.6±0.4 | 90.1±0.5 | 89.8±0.3 | 90.1±0.4 |
|  | 3C | 61.5±0.5 | 62.1±0.3 | 62.2±0.4 | 61.9±0.5 | 62.8±0.6 | 61.6±0.4 | 62.5±0.6 | 62.6±0.4 | 60.7±2.6 | 63.0±0.5 |
|  | 4C | 78.1±0.9 | 80.7±0.3 | 80.6±0.6 | 80.9±0.5 | 81.6±0.2 | 78.5±0.9 | 80.0±0.4 | 80.0±0.5 | 80.1±0.4 | 81.6±0.3 |
|  | 5C | 57.5±0.7 | 59.9±0.5 | 60.0±0.4 | 59.3±1.4 | 60.9±0.4 | 56.4±2.2 | 59.3±0.5 | 59.7±0.4 | 59.5±0.5 | 60.8±0.6 |
| 2007 | SP | 87.8±0.3 | 88.0±0.3 | 87.9±0.3 | 87.8±0.1 | 88.0±0.2 | 87.7±0.3 | 87.8±0.3 | 87.9±0.3 | 87.4±0.2 | 87.8±0.2 |
|  | DP | 87.7±0.4 | 87.8±0.2 | 87.9±0.2 | 87.8±0.2 | 88.0±0.2 | 88.0±0.1 | 88.0±0.2 | 87.8±0.2 | 87.9±0.3 | 88.0±0.3 |
|  | 3C | 61.6±0.6 | 63.2±0.6 | 62.9±0.9 | 62.3±0.4 | 63.4±0.6 | 61.4±0.4 | 63.2±0.1 | 63.4±0.6 | 61.9±0.7 | 63.7±0.5 |
|  | 4C | 79.6±0.6 | 81.7±0.6 | 81.8±0.7 | 81.7±0.8 | 83.1±0.2 | 79.1±1.0 | 82.0±0.6 | 82.0±0.3 | 82.1±0.5 | 83.0±0.2 |
|  | 5C | 60.4±1.0 | 62.8±0.3 | 62.6±0.3 | 61.8±0.6 | 63.7±0.4 | 60.2±1.5 | 62.9±0.5 | 62.9±0.6 | 62.5±0.5 | 64.1±0.4 |
| 2008 | SP | 96.1±0.3 | 96.4±0.3 | 96.2±0.4 | 96.4±0.3 | 96.5±0.3 | 95.6±0.2 | 95.5±0.3 | 95.5±0.4 | 95.5±0.5 | 95.7±0.2 |
|  | DP | 96.4±0.3 | 96.2±0.4 | 96.3±0.3 | 96.5±0.3 | 96.6±0.3 | 96.2±0.3 | 96.5±0.3 | 96.4±0.3 | 96.3±0.4 | 96.6±0.2 |
|  | 3C | 75.4±1.0 | 76.8±0.3 | 77.0±0.2 | 75.4±1.5 | 77.6±0.1 | 73.2±1.0 | 75.9±0.8 | 76.0±0.2 | 74.5±0.9 | 76.7±0.2 |
|  | 4C | 93.9±0.3 | 94.8±0.4 | 95.1±0.2 | 95.0±0.5 | 95.7±0.2 | 92.7±0.6 | 94.0±0.4 | 94.3±0.2 | 94.5±0.4 | 95.4±0.5 |
|  | 5C | 74.7±4.0 | 78.8±0.3 | 78.7±0.4 | 77.8±0.8 | 79.8±0.3 | 74.8±0.9 | 77.4±0.8 | 76.6±2.2 | 76.7±1.2 | 78.9±0.7 |
| 2009 | SP | 94.9±0.2 | 95.1±0.1 | 95.0±0.2 | 95.0±0.2 | 95.2±0.2 | 94.7±0.1 | 94.7±0.1 | 94.7±0.3 | 94.5±0.3 | 94.5±0.2 |
|  | DP | 94.9±0.1 | 95.1±0.1 | 95.0±0.2 | 95.0±0.2 | 95.3±0.1 | 95.1±0.1 | 95.1±0.1 | 95.1±0.2 | 95.1±0.2 | 95.2±0.2 |
|  | 3C | 68.1±0.6 | 70.4±0.1 | 70.8±0.3 | 69.9±0.6 | 70.8±0.4 | 69.2±0.5 | 70.3±0.4 | 70.5±0.4 | 70.0±0.4 | 70.5±0.2 |
|  | 4C | 88.9±1.2 | 90.5±0.4 | 90.8±0.3 | 91.2±0.4 | 91.6±0.3 | 89.9±0.2 | 90.8±0.4 | 90.6±0.4 | 90.9±0.3 | 91.4±0.3 |
|  | 5C | 69.7±0.8 | 72.5±0.3 | 72.4±0.3 | 72.2±0.3 | 73.5±0.4 | 69.9±0.6 | 72.1±0.2 | 72.4±0.3 | 71.9±0.4 | 73.2±0.2 |
| 2010 | SP | 92.3±0.4 | 92.4±0.4 | 92.4±0.3 | 92.3±0.3 | 92.5±0.3 | 92.0±0.5 | 92.2±0.4 | 92.1±0.4 | 91.9±0.3 | 92.2±0.2 |
|  | DP | 92.1±0.5 | 92.4±0.4 | 92.4±0.3 | 92.3±0.3 | 92.4±0.4 | 92.4±0.3 | 92.5±0.3 | 92.5±0.3 | 92.3±0.4 | 92.5±0.4 |
|  | 3C | 63.7±1.8 | 66.5±0.3 | 66.4±0.4 | 66.0±0.4 | 67.2±0.3 | 64.1±0.8 | 66.7±0.7 | 66.6±0.5 | 65.5±0.4 | 67.1±0.5 |
|  | 4C | 87.5±1.0 | 88.3±0.5 | 88.6±0.4 | 88.4±0.6 | 89.4±0.4 | 86.9±1.1 | 88.1±0.3 | 88.1±0.4 | 88.7±0.4 | 89.3±0.3 |
|  | 5C | 67.1±0.7 | 68.8±0.5 | 68.9±0.1 | 67.8±1.0 | 69.8±0.2 | 66.1±1.4 | 68.6±0.6 | 68.6±0.3 | 67.7±0.8 | 69.9±0.3 |

**Table 25:** Link prediction test performance (accuracy in percentage) comparison for variants of MSGNN for individual years 2011-2020 of the *FiLL-OPCL* data set. Each variant is denoted by a $q$ value and a 2-tuple: (whether to include signed features, whether to include weighted features), where "T" and "F" stand for "True" and "False", respectively. "T" for weighted features means simply summing up entries in the adjacency matrix while "T'" means summing the absolute values of the entries. The best is marked in **bold red** and the second best is marked in underline blue.

| Year | Link Task | $q$ value = 0 (F, F) | (F, T) | (F, T') | (T, F) | (T, T) | $q_0 := 1/[2\max_{i,j}(\mathbf{A}_{i,j} - \mathbf{A}_{j,i})]$ (F, F) | (F, T) | (F, T') | (T, F) | (T, T) |
|---|---|---|---|---|---|---|---|---|---|---|---|
| 2011 | SP | 96.0±0.4 | 96.1±0.3 | 96.1±0.3 | **96.2±0.2** | **96.2±0.3** | 95.4±0.4 | 95.5±0.5 | 95.5±0.3 | 95.7±0.3 | 95.9±0.2 |
|  | DP | 95.5±0.9 | 96.0±0.4 | 96.0±0.2 | **96.3±0.2** | **96.3±0.2** | 96.0±0.2 | 96.1±0.3 | 96.2±0.2 | 96.2±0.3 | **96.3±0.3** |
|  | 3C | 77.2±0.4 | 78.3±0.6 | 78.7±0.1 | 77.9±0.4 | **79.0±0.2** | 76.8±0.8 | 78.5±0.6 | 78.7±0.4 | 76.7±1.8 | 78.6±0.4 |
|  | 4C | 94.3±0.4 | 94.9±0.3 | 95.0±0.2 | 95.0±0.3 | 95.4±0.2 | 93.9±0.6 | 94.8±0.2 | 94.7±0.2 | 95.1±0.2 | **95.5±0.3** |
|  | 5C | 77.5±1.3 | 80.2±0.6 | 80.2±0.4 | 79.3±0.5 | **80.9±0.4** | 77.6±1.7 | 79.7±0.6 | 79.8±0.5 | 79.8±0.4 | 80.6±0.3 |
| 2012 | SP | 91.0±0.5 | **91.1±0.3** | **91.1±0.2** | 90.9±0.4 | **91.1±0.2** | 90.6±0.4 | 90.7±0.6 | 90.8±0.4 | 90.6±0.2 | 90.8±0.5 |
|  | DP | 90.9±0.4 | 91.1±0.3 | 91.0±0.2 | 91.0±0.3 | **91.2±0.3** | 91.0±0.3 | **91.2±0.3** | 91.1±0.4 | 91.0±0.2 | **91.2±0.2** |
|  | 3C | 63.1±0.8 | 64.1±0.4 | 63.6±0.4 | 63.4±0.4 | 64.2±0.4 | 63.4±0.3 | 64.0±0.7 | 64.2±0.6 | 63.3±1.2 | **64.4±0.5** |
|  | 4C | 78.6±0.9 | 80.6±0.2 | 80.6±0.3 | 80.9±0.6 | **81.7±0.4** | 78.0±1.5 | 80.6±0.4 | 80.4±0.3 | 80.5±0.5 | 81.5±0.4 |
|  | 5C | 57.2±1.3 | 60.6±0.0 | 60.7±0.5 | 60.0±0.7 | **61.6±0.5** | 55.8±3.1 | 60.5±0.5 | 60.3±0.3 | 60.2±0.5 | 61.5±0.3 |
| 2013 | SP | 89.1±0.2 | 89.3±0.2 | 89.3±0.3 | 89.3±0.3 | **89.5±0.3** | 88.9±0.3 | 88.9±0.3 | 89.0±0.2 | 88.9±0.2 | 89.4±0.2 |
|  | DP | 89.0±0.3 | 89.3±0.2 | 89.2±0.3 | 89.1±0.3 | **89.4±0.3** | 89.1±0.2 | 89.3±0.2 | 89.2±0.3 | 89.2±0.2 | 89.3±0.2 |
|  | 3C | 63.3±0.2 | 63.8±0.4 | 63.9±0.2 | 63.2±0.4 | **64.3±0.4** | 62.8±0.4 | 63.9±0.3 | 64.1±0.3 | 63.0±0.5 | **64.3±0.3** |
|  | 4C | 79.8±1.6 | 82.0±0.4 | 81.8±0.4 | 82.6±0.6 | **83.3±0.3** | 80.8±1.4 | 82.3±0.3 | 82.0±0.5 | 81.7±1.0 | 82.8±0.8 |
|  | 5C | 61.5±0.5 | 63.1±0.4 | 63.0±0.5 | 61.9±1.6 | **64.2±0.3** | 60.7±0.8 | 62.5±0.7 | 63.1±0.6 | 62.4±0.8 | 63.9±0.4 |
| 2014 | SP | 87.7±0.3 | **87.8±0.4** | 87.6±0.3 | 87.6±0.3 | 87.7±0.4 | 87.5±0.4 | 87.3±0.7 | 87.5±0.3 | 87.3±0.5 | 87.5±0.3 |
|  | DP | 87.6±0.4 | 87.6±0.3 | 87.7±0.3 | 87.7±0.5 | 87.7±0.5 | 87.6±0.3 | 87.6±0.3 | 87.6±0.4 | 87.7±0.5 | **87.9±0.4** |
|  | 3C | 60.5±0.6 | 61.2±0.5 | 61.2±0.7 | 60.9±0.3 | 61.8±0.7 | 60.6±0.5 | 61.7±0.4 | 61.6±0.2 | 61.1±0.3 | **61.9±0.2** |
|  | 4C | 78.3±1.6 | 80.1±0.2 | 80.4±0.3 | 80.3±0.4 | **81.3±0.3** | 78.5±0.3 | 80.6±0.2 | 80.4±0.3 | 80.9±0.4 | 81.2±0.2 |
|  | 5C | 58.4±1.1 | 61.0±0.4 | 61.2±0.4 | 60.5±0.5 | **61.9±0.3** | 58.2±1.0 | 60.5±0.3 | 60.9±0.3 | 59.0±1.6 | 61.8±0.3 |
| 2015 | SP | 89.5±0.4 | 89.6±0.3 | 89.7±0.3 | 89.5±0.4 | **89.8±0.3** | 89.1±0.2 | 89.1±0.4 | 89.2±0.5 | 89.0±0.5 | 89.6±0.4 |
|  | DP | 89.5±0.2 | 89.7±0.3 | 89.5±0.3 | 89.7±0.3 | **89.9±0.3** | 89.6±0.2 | 89.6±0.3 | 89.7±0.3 | 89.6±0.2 | 89.8±0.3 |
|  | 3C | 62.2±0.6 | 63.5±0.6 | 63.7±0.5 | 62.7±0.8 | **64.1±0.4** | 62.0±1.0 | 63.8±0.3 | 63.7±0.5 | 62.8±0.2 | **64.1±0.2** |
|  | 4C | 82.1±0.5 | 83.6±0.4 | 83.5±0.2 | 83.7±0.6 | 84.7±0.3 | 82.8±0.4 | 83.6±0.4 | 83.6±0.3 | 83.8±0.5 | **84.8±0.2** |
|  | 5C | 61.9±0.5 | 63.8±0.5 | 63.9±0.6 | 63.5±0.7 | **64.9±0.4** | 61.5±0.5 | 63.7±0.5 | 63.7±0.8 | 63.4±1.1 | 64.8±0.5 |
| 2016 | SP | 88.9±0.4 | 88.9±0.3 | 88.9±0.3 | 88.8±0.4 | **89.0±0.2** | 88.7±0.2 | 88.8±0.3 | 88.5±0.3 | 88.7±0.4 | 88.7±0.3 |
|  | DP | 88.9±0.3 | 88.7±0.3 | 88.6±0.4 | 88.8±0.4 | **88.9±0.2** | 88.6±0.2 | **88.9±0.2** | 88.8±0.3 | 88.7±0.3 | **88.9±0.3** |
|  | 3C | 61.4±0.6 | 62.4±0.6 | 61.9±0.3 | 61.6±0.5 | **62.5±0.2** | 61.5±0.6 | 61.8±0.5 | 62.0±0.4 | 61.4±0.5 | 62.2±0.5 |
|  | 4C | 72.6±1.7 | 75.2±0.4 | 74.4±0.2 | 76.2±0.6 | **76.8±0.2** | 73.6±1.0 | 75.0±0.7 | 74.9±0.1 | 75.6±0.6 | 76.7±0.5 |
|  | 5C | 54.7±1.0 | 56.8±0.7 | 56.6±0.5 | 56.4±1.0 | 57.9±0.3 | 51.4±1.6 | 56.9±0.8 | 57.1±0.7 | 56.2±0.6 | **58.1±0.3** |
| 2017 | SP | 89.2±0.2 | **89.4±0.2** | 89.2±0.2 | 89.2±0.2 | 89.3±0.3 | 89.1±0.2 | 89.3±0.2 | 89.3±0.3 | 88.9±0.3 | **89.4±0.3** |
|  | DP | 89.3±0.2 | 89.2±0.3 | 89.2±0.3 | 89.4±0.1 | 89.3±0.3 | 89.3±0.3 | 89.4±0.2 | 89.3±0.3 | 89.3±0.2 | **89.5±0.2** |
|  | 3C | 60.4±0.6 | 61.0±0.3 | 61.1±0.2 | 60.7±0.3 | 61.3±0.3 | 60.9±0.5 | **61.4±0.3** | 61.3±0.2 | 60.7±0.3 | 61.3±0.2 |
|  | 4C | 65.8±0.4 | 68.4±0.6 | 68.3±0.3 | 68.9±0.8 | **69.5±0.5** | 64.9±2.9 | 68.4±0.8 | 68.6±0.5 | 68.1±0.7 | 69.4±0.4 |
|  | 5C | 49.2±0.5 | 51.0±0.3 | 51.1±0.6 | 49.9±1.3 | **52.0±0.4** | 47.9±1.2 | 51.1±0.6 | 51.4±0.4 | 49.7±0.8 | 51.8±0.3 |
| 2018 | SP | 88.0±0.5 | 88.0±0.4 | 88.0±0.4 | 88.0±0.3 | **88.2±0.4** | 87.3±0.5 | 87.8±0.5 | 87.8±0.6 | 87.4±0.6 | 87.7±0.6 |
|  | DP | 87.7±0.7 | 88.0±0.6 | 88.0±0.4 | **88.1±0.4** | **88.1±0.4** | 88.0±0.6 | **88.1±0.5** | 88.0±0.4 | 87.8±0.6 | **88.1±0.4** |
|  | 3C | 61.1±0.4 | 63.1±0.5 | 63.1±0.5 | 62.5±0.6 | 63.7±0.3 | 61.3±1.3 | 63.8±0.4 | 63.8±0.5 | 62.0±0.7 | **64.0±0.6** |
|  | 4C | 80.6±0.5 | 82.1±0.3 | 82.0±0.4 | 82.1±0.6 | **83.0±0.4** | 79.6±0.6 | 81.1±0.5 | 81.3±0.3 | 81.6±1.2 | **83.0±0.7** |
|  | 5C | 61.1±0.9 | 62.8±0.5 | 62.7±0.5 | 62.2±0.5 | **63.8±0.5** | 58.3±5.1 | 62.3±0.4 | 62.4±0.5 | 61.9±0.9 | 63.7±0.5 |
| 2019 | SP | 89.2±0.2 | 89.1±0.3 | 89.2±0.5 | 89.1±0.4 | **89.3±0.4** | 88.6±1.1 | 89.1±0.2 | 89.0±0.2 | 88.6±0.3 | 89.1±0.5 |
|  | DP | 89.0±0.4 | 89.2±0.2 | 89.3±0.2 | 89.1±0.2 | **89.4±0.2** | 89.3±0.3 | 89.3±0.2 | 89.3±0.4 | 89.2±0.4 | 89.3±0.2 |
|  | 3C | 61.3±0.3 | 62.0±0.4 | 61.9±0.2 | 61.6±0.3 | 62.2±0.2 | 60.4±0.8 | 62.3±0.5 | 62.2±0.5 | 62.1±0.5 | **62.4±0.5** |
|  | 4C | 70.8±1.0 | 72.4±0.4 | 72.8±0.3 | 73.2±0.5 | **74.5±0.2** | 71.6±0.8 | 72.9±0.3 | 72.9±0.2 | 73.1±0.6 | 74.3±0.4 |
|  | 5C | 51.4±2.0 | 55.0±0.3 | 55.2±0.2 | 53.7±0.9 | **56.0±0.3** | 52.4±1.3 | 55.1±0.3 | 55.2±0.2 | 53.3±2.3 | **56.0±0.2** |
| 2020 | SP | 92.0±0.1 | 91.8±0.4 | 91.8±0.2 | 92.2±0.2 | **92.3±0.1** | 91.5±0.2 | 91.4±0.2 | 91.3±0.2 | 91.6±0.2 | 91.5±0.1 |
|  | DP | 91.8±0.2 | 91.7±0.3 | 92.0±0.3 | **92.2±0.1** | **92.2±0.1** | 91.7±0.3 | 92.0±0.2 | 91.9±0.1 | 92.0±0.2 | 92.1±0.1 |
|  | 3C | 66.5±0.7 | 69.0±0.3 | 68.6±0.5 | 68.2±1.0 | **69.2±0.3** | 66.0±1.2 | **69.2±0.3** | 69.0±0.4 | 67.3±0.7 | **69.2±0.4** |
|  | 4C | 83.0±0.9 | 84.0±0.4 | 84.0±0.3 | 84.8±0.3 | **85.4±0.2** | 81.4±1.4 | 83.1±0.5 | 83.1±0.7 | 84.1±1.0 | 84.8±0.4 |
|  | 5C | 64.2±1.7 | 66.5±0.5 | 66.6±0.7 | 66.3±0.6 | **67.8±0.2** | 62.5±0.5 | 65.1±1.0 | 65.6±1.4 | 65.3±1.2 | 67.3±0.6 |

**Table 26:** Link prediction test performance (accuracy in percentage) comparison for MSGNN with different $q$ values for individual years 2000-2010 of the *FiLL-pvCLCL* data set. The best is marked in **bold red** and the second best is marked in underline blue.

| Year | Link Task | $q = 0$ | $q = 0.2q_0$ | $q = 0.4q_0$ | $q = 0.6q_0$ | $q = 0.8q_0$ | $q = q_0$ |
|---|---|---|---|---|---|---|---|
| 2000 | SP | 89.0±0.4 | **89.1±0.3** | 88.9±0.5 | 88.9±0.4 | 88.7±0.6 | 88.8±0.5 |
|  | DP | **89.1±0.4** | **89.1±0.4** | **89.1±0.3** | **89.1±0.4** | **89.1±0.4** | **89.1±0.5** |
|  | 3C | 61.3±0.4 | 61.3±0.4 | 61.3±0.5 | 61.3±0.5 | **61.5±0.5** | **61.5±0.7** |
|  | 4C | **72.3±0.5** | 72.1±0.5 | 72.1±0.7 | 72.1±0.5 | 72.2±0.6 | 71.9±0.5 |
|  | 5C | **54.0±0.4** | 53.9±0.4 | **54.0±0.5** | 53.9±0.6 | 53.8±0.5 | 53.9±0.4 |
| 2001 | SP | **90.7±0.2** | 90.5±0.3 | 90.4±0.2 | 90.6±0.3 | 90.4±0.3 | **90.7±0.2** |
|  | DP | **90.7±0.2** | **90.7±0.1** | **90.7±0.2** | **90.7±0.2** | **90.7±0.2** | **90.7±0.1** |
|  | 3C | 62.6±0.2 | 62.6±0.2 | 62.6±0.3 | 62.7±0.2 | 62.6±0.2 | **63.1±0.5** |
|  | 4C | 75.8±0.3 | 75.8±0.4 | 75.7±0.3 | **76.1±0.3** | 75.9±0.4 | 75.7±0.3 |
|  | 5C | 56.7±0.4 | **56.8±0.2** | 56.7±0.3 | **56.8±0.3** | **56.8±0.3** | **56.8±0.2** |
| 2002 | SP | **91.4±0.2** | 91.0±0.3 | 91.1±0.2 | 90.9±0.4 | 91.2±0.3 | 91.2±0.3 |
|  | DP | **91.5±0.1** | **91.5±0.1** | 91.4±0.3 | **91.5±0.2** | 91.4±0.1 | **91.5±0.2** |
|  | 3C | **66.2±0.4** | 66.0±0.2 | 66.0±0.4 | 66.1±0.2 | 65.9±0.4 | **66.2±0.2** |
|  | 4C | **85.7±0.5** | **85.7±0.5** | 85.5±0.3 | 85.5±0.3 | 85.6±0.4 | **85.7±0.4** |
|  | 5C | **66.8±0.4** | **66.8±0.4** | 66.7±0.5 | 66.7±0.3 | **66.8±0.3** | 66.7±0.4 |
| 2003 | SP | **89.5±0.4** | 89.0±0.4 | 89.4±0.3 | 89.2±0.4 | 89.2±0.3 | 89.3±0.4 |
|  | DP | **89.6±0.4** | 89.5±0.4 | 89.5±0.4 | 89.5±0.4 | **89.6±0.4** | **89.6±0.4** |
|  | 3C | 63.2±0.5 | 62.9±0.5 | **63.3±0.3** | 63.2±0.4 | 63.2±0.5 | 63.1±0.5 |
|  | 4C | 82.6±0.4 | 82.8±0.3 | **82.9±0.2** | 82.8±0.2 | **82.9±0.2** | 82.7±0.4 |
|  | 5C | 62.6±0.2 | 62.7±0.3 | **62.8±0.3** | **62.8±0.4** | 62.4±0.4 | 62.7±0.4 |
| 2004 | SP | 88.7±0.3 | 88.2±0.8 | 88.7±0.3 | 88.7±0.3 | 88.6±0.2 | **88.8±0.3** |
|  | DP | **88.8±0.3** | 88.7±0.3 | 88.7±0.4 | **88.8±0.4** | 88.7±0.2 | **88.8±0.3** |
|  | 3C | 61.6±0.5 | 61.7±0.5 | **61.8±0.5** | 61.7±0.2 | **61.8±0.3** | 61.6±0.4 |
|  | 4C | 78.8±0.6 | 78.7±0.4 | **78.9±0.5** | 78.6±0.4 | 78.7±0.5 | 78.7±0.4 |
|  | 5C | 58.7±0.3 | 58.7±0.4 | **58.8±0.5** | **58.8±0.5** | **58.8±0.3** | 58.7±0.6 |
| 2005 | SP | **87.8±0.4** | 87.4±0.6 | 87.4±0.4 | 87.7±0.5 | 87.7±0.3 | 87.7±0.4 |
|  | DP | **87.9±0.4** | 87.7±0.5 | 87.8±0.5 | 87.8±0.4 | 87.8±0.4 | 87.8±0.4 |
|  | 3C | **61.2±0.2** | **61.2±0.2** | 61.0±0.4 | 60.9±0.3 | 61.1±0.2 | 60.9±0.6 |
|  | 4C | **80.8±0.2** | 80.7±0.4 | **80.8±0.2** | **80.8±0.2** | 80.6±0.2 | 80.5±0.2 |
|  | 5C | **61.3±0.4** | **61.3±0.3** | 61.2±0.3 | **61.3±0.2** | 61.1±0.2 | 61.0±0.2 |
| 2006 | SP | **91.0±0.2** | 90.7±0.4 | 90.7±0.2 | **91.0±0.3** | 90.6±0.1 | 90.6±0.1 |
|  | DP | **91.1±0.2** | 91.0±0.2 | **91.1±0.1** | 91.0±0.1 | 90.9±0.2 | **91.1±0.1** |
|  | 3C | 64.0±0.3 | **64.3±0.4** | 64.2±0.4 | 64.0±0.4 | 64.0±0.2 | 64.1±0.3 |
|  | 4C | 82.9±0.2 | 82.8±0.2 | 83.0±0.3 | 82.9±0.2 | 82.9±0.3 | **83.1±0.4** |
|  | 5C | 62.6±0.3 | 62.6±0.2 | 62.6±0.4 | 62.6±0.2 | **63.0±0.3** | 62.8±0.3 |
| 2007 | SP | **90.4±0.5** | 90.3±0.4 | 90.0±0.3 | 90.3±0.3 | 90.1±0.3 | 90.0±0.4 |
|  | DP | 90.4±0.3 | 90.5±0.4 | 90.4±0.4 | 90.4±0.3 | 90.5±0.4 | **90.6±0.4** |
|  | 3C | 69.0±0.4 | 69.0±0.7 | **69.2±0.3** | 69.1±0.3 | 69.0±0.2 | 69.0±0.3 |
|  | 4C | 88.1±0.2 | 88.3±0.4 | 88.2±0.4 | 88.1±0.3 | **88.4±0.2** | **88.4±0.2** |
|  | 5C | 69.9±0.5 | 69.9±0.6 | **70.0±0.5** | 69.7±0.4 | 69.7±0.5 | 69.7±0.2 |
| 2008 | SP | **96.4±0.2** | 95.8±0.2 | 95.9±0.1 | 95.7±0.3 | 95.5±0.4 | 95.7±0.3 |
|  | DP | 96.4±0.1 | **96.5±0.2** | **96.5±0.1** | **96.5±0.1** | 96.3±0.4 | 96.3±0.2 |
|  | 3C | **79.3±0.7** | 79.0±0.2 | 79.2±0.3 | 79.1±0.1 | 78.5±0.3 | 78.9±0.2 |
|  | 4C | 96.1±0.2 | **96.5±0.2** | 96.3±0.3 | 96.2±0.2 | 96.1±0.5 | 96.2±0.3 |
|  | 5C | **82.4±0.4** | 82.2±0.6 | 82.2±0.7 | 82.2±0.6 | 82.1±0.4 | 82.0±0.4 |
| 2009 | SP | **97.8±0.2** | 97.2±0.1 | 97.3±0.2 | 97.3±0.2 | 97.1±0.2 | 97.1±0.3 |
|  | DP | **97.8±0.1** | 97.7±0.1 | **97.8±0.2** | **97.8±0.2** | **97.8±0.2** | 97.6±0.1 |
|  | 3C | 78.4±0.4 | **78.6±0.4** | 78.5±0.2 | 78.5±0.5 | 78.4±0.4 | **78.6±0.3** |
|  | 4C | 95.2±0.3 | **95.3±0.3** | 95.1±0.4 | 95.1±0.3 | 94.8±0.3 | 95.0±0.2 |
|  | 5C | **79.9±0.2** | **79.9±0.3** | 79.8±0.2 | **79.9±0.4** | 79.6±0.6 | 79.8±0.4 |
| 2010 | SP | **92.8±0.3** | 92.0±0.2 | 92.5±0.2 | 92.4±0.3 | 92.4±0.3 | 92.3±0.3 |
|  | DP | 92.6±0.3 | 92.7±0.3 | 92.7±0.3 | 92.6±0.4 | **92.8±0.4** | 92.7±0.4 |
|  | 3C | 68.3±0.2 | 68.4±0.2 | 68.4±0.4 | 68.4±0.3 | **68.6±0.3** | 68.5±0.4 |
|  | 4C | 90.9±0.4 | 90.9±0.3 | 91.0±0.4 | 90.9±0.3 | 91.0±0.4 | **91.1±0.4** |
|  | 5C | **72.5±0.4** | **72.5±0.3** | 72.4±0.3 | **72.5±0.3** | 72.4±0.3 | 72.4±0.1 |

**Table 27:** Link prediction test performance (accuracy in percentage) comparison for MSGNN with different $q$ values for individual years 2011-2020 of the *FiLL-pvCLCL* data set. The best is marked in **bold red** and the second best is marked in underline blue.

| Year | Link Task | $q = 0$ | $q = 0.2q_0$ | $q = 0.4q_0$ | $q = 0.6q_0$ | $q = 0.8q_0$ | $q = q_0$ |
|------|-----------|---------|--------------|--------------|--------------|--------------|-----------|
| 2011 | SP | **98.7±0.2** | 98.4±0.3 | 98.3±0.2 | 98.5±0.2 | 98.2±0.5 | 98.3±0.3 |
|      | DP | **98.7±0.2** | **98.7±0.1** | **98.7±0.1** | **98.7±0.1** | **98.7±0.1** | **98.7±0.1** |
|      | 3C | 86.5±0.2 | **86.6±0.1** | 86.1±0.5 | 86.4±0.2 | 86.2±0.3 | 86.2±0.3 |
|      | 4C | 98.3±0.3 | **98.4±0.1** | **98.4±0.2** | 98.3±0.4 | 98.3±0.2 | 98.3±0.2 |
|      | 5C | 87.4±0.3 | 87.3±0.3 | **87.5±0.3** | 87.3±0.4 | 87.2±0.6 | 87.2±0.4 |
| 2012 | SP | **92.7±0.3** | 92.1±0.8 | 92.4±0.3 | 92.4±0.4 | 92.5±0.3 | 92.4±0.3 |
|      | DP | 92.6±0.2 | 92.6±0.2 | **92.7±0.3** | **92.7±0.4** | **92.7±0.3** | 92.6±0.3 |
|      | 3C | 67.1±0.2 | **67.5±0.4** | 67.1±0.2 | 67.2±0.4 | 67.1±0.3 | 67.4±0.3 |
|      | 4C | 87.0±0.5 | 86.9±0.4 | **87.1±0.5** | 87.0±0.5 | 87.0±0.5 | 86.9±0.4 |
|      | 5C | 66.8±0.1 | 66.9±0.1 | 66.8±0.3 | 66.8±0.3 | 66.9±0.1 | **67.0±0.2** |
| 2013 | SP | **90.5±0.3** | 90.0±0.2 | 90.2±0.4 | 90.1±0.2 | 90.3±0.3 | 90.3±0.1 |
|      | DP | 90.3±0.3 | **90.5±0.2** | 90.4±0.1 | 90.3±0.3 | 90.4±0.2 | 90.4±0.3 |
|      | 3C | **66.9±0.2** | 66.3±0.4 | 66.3±0.4 | 66.4±0.4 | 66.3±0.4 | 66.1±0.3 |
|      | 4C | 85.9±0.3 | **86.1±0.2** | **86.1±0.3** | **86.1±0.2** | 86.0±0.3 | **86.1±0.3** |
|      | 5C | 67.0±0.3 | 66.9±0.2 | 67.0±0.4 | **67.1±0.2** | 67.0±0.1 | 66.8±0.3 |
| 2014 | SP | **87.3±0.2** | 86.8±0.2 | 87.2±0.2 | 87.0±0.3 | 86.9±0.3 | 87.1±0.2 |
|      | DP | **87.3±0.2** | 87.2±0.2 | 87.2±0.2 | 87.2±0.3 | 87.2±0.2 | 87.2±0.3 |
|      | 3C | **60.8±0.3** | 60.5±0.2 | **60.8±0.1** | **60.8±0.2** | **60.8±0.2** | 60.6±0.2 |
|      | 4C | 80.2±0.3 | 80.1±0.2 | 80.2±0.3 | **80.3±0.3** | 80.1±0.4 | 80.2±0.2 |
|      | 5C | 60.3±0.4 | 60.4±0.2 | **60.5±0.2** | 60.4±0.4 | 60.3±0.3 | 60.3±0.4 |
| 2015 | SP | **89.1±0.4** | 87.7±1.0 | 88.9±0.1 | 88.8±0.4 | 88.8±0.1 | 88.8±0.2 |
|      | DP | 89.2±0.4 | 89.2±0.4 | 89.2±0.3 | 89.2±0.4 | **89.4±0.4** | 89.3±0.3 |
|      | 3C | 63.5±0.5 | 63.9±0.3 | **64.1±0.2** | 63.7±0.3 | 63.7±0.4 | 63.8±0.4 |
|      | 4C | 84.6±0.5 | 84.5±0.5 | 84.6±0.3 | 84.5±0.3 | **84.7±0.4** | 84.4±0.5 |
|      | 5C | 65.6±0.4 | **65.9±0.3** | 65.8±0.2 | 65.5±0.4 | 65.4±0.3 | 65.7±0.4 |
| 2016 | SP | **90.2±0.5** | 89.7±0.4 | 89.6±0.4 | 89.8±0.5 | 89.7±0.5 | 89.7±0.5 |
|      | DP | 90.1±0.5 | **90.2±0.4** | **90.2±0.4** | **90.2±0.4** | 90.0±0.4 | 90.0±0.5 |
|      | 3C | 64.0±0.3 | 64.0±0.3 | 63.9±0.2 | 63.9±0.3 | **64.1±0.3** | 63.9±0.2 |
|      | 4C | 82.0±0.5 | 81.8±0.5 | 81.7±0.6 | **82.2±0.6** | 81.8±0.6 | 82.1±0.5 |
|      | 5C | **62.2±0.3** | 62.1±0.3 | 62.0±0.4 | 62.1±0.5 | 61.9±0.4 | 62.0±0.4 |
| 2017 | SP | **90.2±0.3** | 89.8±0.4 | 89.8±0.9 | 89.9±0.4 | 90.0±0.3 | 90.0±0.3 |
|      | DP | 90.1±0.2 | **90.2±0.3** | **90.2±0.3** | 90.0±0.4 | 90.1±0.3 | **90.2±0.3** |
|      | 3C | 62.0±0.1 | 62.4±0.4 | 62.4±0.3 | **62.5±0.4** | 62.4±0.2 | 62.3±0.4 |
|      | 4C | **70.1±0.5** | 70.0±0.5 | **70.1±0.5** | 70.0±0.2 | 69.9±0.5 | 69.6±0.4 |
|      | 5C | **53.4±0.2** | 53.2±0.2 | **53.4±0.1** | 53.3±0.2 | 53.3±0.3 | 53.2±0.1 |
| 2018 | SP | **87.0±0.4** | 86.7±0.4 | 85.8±1.8 | 86.8±0.6 | 86.7±0.3 | 86.6±0.8 |
|      | DP | 86.9±0.4 | **87.0±0.5** | 86.8±0.4 | **87.0±0.5** | **87.0±0.5** | **87.0±0.4** |
|      | 3C | 61.5±0.4 | 61.5±0.6 | **61.8±0.3** | **61.8±0.5** | 61.7±0.3 | 61.4±0.3 |
|      | 4C | 79.7±0.4 | **79.8±0.4** | 79.5±0.4 | 79.7±0.4 | 79.7±0.5 | 79.5±0.5 |
|      | 5C | **60.9±0.4** | 60.8±0.5 | 60.7±0.5 | 60.7±0.7 | 60.6±0.5 | 60.5±0.3 |
| 2019 | SP | **90.8±0.3** | 90.4±0.2 | 90.3±0.2 | 90.6±0.3 | 90.4±0.3 | 90.6±0.4 |
|      | DP | **90.9±0.3** | **90.9±0.2** | 90.8±0.3 | 90.8±0.2 | 90.8±0.2 | **90.9±0.3** |
|      | 3C | **64.6±0.2** | 64.5±0.3 | 64.3±0.2 | 64.3±0.2 | 64.4±0.2 | 64.3±0.2 |
|      | 4C | **77.9±0.3** | 77.9±0.3 | **77.9±0.3** | 77.8±0.5 | 77.6±0.7 | 77.5±0.4 |
|      | 5C | **59.0±0.2** | 58.8±0.2 | **59.0±0.4** | 58.9±0.4 | **59.0±0.3** | 58.7±0.2 |
| 2020 | SP | **97.3±0.2** | 96.8±0.2 | 96.7±0.2 | 96.5±0.2 | 96.6±0.1 | 96.6±0.2 |
|      | DP | 97.3±0.2 | **97.4±0.2** | 97.3±0.1 | 97.3±0.2 | 97.3±0.1 | 97.2±0.1 |
|      | 3C | 82.8±1.3 | 82.7±1.0 | **83.2±0.5** | 82.7±0.8 | 82.8±0.5 | 82.9±0.5 |
|      | 4C | 96.4±0.2 | **96.6±0.2** | 96.5±0.1 | 96.4±0.1 | 96.4±0.3 | 96.5±0.1 |
|      | 5C | **83.2±0.4** | 83.0±0.3 | 82.9±0.5 | 83.1±0.2 | 83.0±0.4 | 82.8±0.4 |

**Table 28:** Link prediction test performance (accuracy in percentage) comparison for MSGNN with different $q$ values for individual years 2000-2010 of the *FiLL-OPCL* data set. The best is marked in **bold red** and the second best is marked in underline blue.

| Year | Link Task | $q = 0$ | $q = 0.2q_0$ | $q = 0.4q_0$ | $q = 0.6q_0$ | $q = 0.8q_0$ | $q = q_0$ |
|---|---|---|---|---|---|---|---|
| 2000 | SP | **87.9±0.5** | 87.4±0.6 | 87.6±0.5 | 87.5±0.5 | 87.6±0.4 | 87.6±0.5 |
|  | DP | 87.9±0.5 | 87.9±0.4 | 87.7±0.7 | **88.0±0.5** | 87.9±0.6 | 87.9±0.6 |
|  | 3C | 60.7±0.3 | **60.8±0.4** | 60.7±0.3 | 60.2±0.4 | 60.7±0.5 | 60.7±0.4 |
|  | 4C | 71.3±0.3 | 71.3±0.4 | 71.3±0.4 | **71.5±0.3** | 71.4±0.4 | 71.3±0.3 |
|  | 5C | **53.5±0.4** | 53.2±0.5 | 53.4±0.4 | 53.4±0.4 | 53.4±0.5 | 53.4±0.5 |
| 2001 | SP | **90.0±0.5** | 89.9±0.3 | 89.9±0.4 | 89.6±0.5 | 89.8±0.5 | 89.9±0.4 |
|  | DP | 90.0±0.3 | 90.1±0.4 | **90.2±0.2** | 90.1±0.4 | 90.1±0.4 | **90.2±0.3** |
|  | 3C | 61.9±0.3 | 62.1±0.7 | 61.9±0.5 | **62.2±0.4** | 62.0±0.5 | **62.2±0.4** |
|  | 4C | 75.5±0.3 | 75.2±0.3 | **75.6±0.5** | 75.5±0.3 | **75.6±0.6** | 75.4±0.6 |
|  | 5C | **55.8±0.4** | **55.8±0.4** | 55.6±0.4 | 55.7±0.3 | **55.8±0.4** | **55.8±0.4** |
| 2002 | SP | **90.5±0.2** | 90.1±0.1 | 90.2±0.3 | 90.0±0.2 | 90.0±0.3 | 90.0±0.4 |
|  | DP | 90.3±0.2 | **90.5±0.3** | 90.4±0.3 | 90.4±0.2 | **90.5±0.3** | **90.5±0.3** |
|  | 3C | 65.1±0.7 | 65.1±0.5 | 65.3±0.3 | 65.4±0.5 | **65.5±0.3** | 65.0±0.4 |
|  | 4C | **84.5±0.5** | 84.3±0.4 | 84.3±0.5 | **84.5±0.4** | 84.4±0.3 | **84.5±0.5** |
|  | 5C | 65.4±0.3 | **65.6±0.5** | 65.4±0.7 | 65.5±0.5 | 65.4±0.4 | 65.5±0.2 |
| 2003 | SP | **89.1±0.3** | 89.0±0.3 | **89.1±0.3** | **89.1±0.3** | 88.8±0.3 | 88.9±0.3 |
|  | DP | 89.2±0.5 | 89.2±0.3 | 89.0±0.2 | **89.3±0.4** | 89.1±0.4 | 89.2±0.5 |
|  | 3C | 62.5±0.5 | **62.7±0.3** | 62.5±0.5 | 62.5±0.5 | 62.5±0.5 | **62.7±0.4** |
|  | 4C | 82.2±0.4 | 82.3±0.3 | 82.4±0.5 | 82.3±0.4 | 82.3±0.4 | **82.5±0.3** |
|  | 5C | 62.4±0.3 | 62.3±0.4 | 62.4±0.4 | 62.3±0.4 | **62.6±0.2** | 62.3±0.4 |
| 2004 | SP | 87.4±0.3 | 87.0±0.4 | 87.3±0.2 | 87.4±0.4 | **87.5±0.3** | 87.4±0.4 |
|  | DP | 87.4±0.3 | **87.5±0.3** | 87.3±0.3 | **87.5±0.4** | **87.5±0.3** | 87.4±0.2 |
|  | 3C | 60.3±0.4 | 60.6±0.5 | 60.5±0.5 | 60.0±0.4 | **60.7±0.3** | 60.1±0.7 |
|  | 4C | 78.3±0.3 | **78.6±0.2** | 78.3±0.3 | 78.4±0.4 | 78.2±0.2 | 78.3±0.3 |
|  | 5C | 57.9±0.4 | 57.9±0.4 | **58.1±0.3** | 57.8±0.1 | 57.9±0.5 | 57.9±0.3 |
| 2005 | SP | **86.4±0.3** | 86.3±0.3 | 86.0±0.4 | 86.2±0.3 | 86.2±0.5 | 86.1±0.4 |
|  | DP | 86.4±0.3 | **86.5±0.3** | 86.4±0.3 | **86.5±0.3** | 86.4±0.3 | 86.4±0.3 |
|  | 3C | 59.5±0.6 | **59.6±0.3** | **59.6±0.6** | **59.6±0.3** | 59.5±0.4 | 59.5±0.6 |
|  | 4C | 79.6±0.2 | 79.5±0.4 | 79.5±0.3 | 79.6±0.3 | **79.7±0.4** | 79.5±0.3 |
|  | 5C | 59.0±0.4 | 59.0±0.2 | 59.0±0.4 | **59.1±0.3** | 59.0±0.4 | 58.8±0.5 |
| 2006 | SP | **89.9±0.5** | 89.5±0.4 | 89.2±1.2 | 89.6±0.6 | 89.7±0.5 | 89.7±0.4 |
|  | DP | 89.9±0.4 | 89.9±0.4 | **90.1±0.4** | 90.0±0.4 | **90.1±0.4** | **90.1±0.4** |
|  | 3C | 62.8±0.6 | 62.7±0.4 | 62.7±0.6 | 62.7±0.4 | **63.0±0.5** | **63.0±0.5** |
|  | 4C | **81.6±0.2** | 81.4±0.4 | 81.5±0.3 | 81.2±0.4 | 81.4±0.3 | **81.6±0.3** |
|  | 5C | 60.9±0.4 | 60.9±0.6 | 60.9±0.5 | **61.0±0.4** | 60.9±0.4 | 60.8±0.6 |
| 2007 | SP | **88.0±0.2** | 87.7±0.2 | 87.8±0.3 | 87.9±0.2 | 87.8±0.2 | 87.8±0.2 |
|  | DP | 88.0±0.2 | 87.9±0.3 | 88.0±0.3 | 88.0±0.3 | **88.1±0.3** | 88.0±0.3 |
|  | 3C | 63.4±0.6 | 63.8±0.4 | 63.7±0.6 | **63.9±0.5** | 63.8±0.4 | 63.7±0.5 |
|  | 4C | 83.1±0.2 | **83.4±0.4** | 83.1±0.3 | 83.3±0.3 | 83.1±0.4 | 83.0±0.2 |
|  | 5C | 63.7±0.4 | 64.0±0.2 | 64.0±0.4 | 64.0±0.4 | 63.8±0.4 | **64.1±0.4** |
| 2008 | SP | **96.5±0.3** | 96.0±0.4 | 95.9±0.3 | 95.8±0.4 | 95.7±0.4 | 95.7±0.2 |
|  | DP | **96.6±0.3** | 96.5±0.2 | 96.5±0.2 | 96.4±0.2 | 96.4±0.3 | **96.6±0.2** |
|  | 3C | **77.6±0.1** | 76.6±0.6 | 76.7±0.4 | 77.0±0.3 | 76.6±0.5 | 76.7±0.2 |
|  | 4C | **95.7±0.2** | 95.6±0.2 | **95.7±0.2** | 95.6±0.2 | **95.7±0.3** | 95.4±0.5 |
|  | 5C | **79.8±0.3** | 79.7±0.4 | **79.8±0.3** | 79.5±0.2 | 79.4±0.1 | 78.9±0.7 |
| 2009 | SP | **95.2±0.2** | 94.8±0.2 | 94.6±0.2 | 94.7±0.2 | 94.5±0.4 | 94.5±0.2 |
|  | DP | **95.3±0.1** | 95.1±0.1 | 95.2±0.1 | 95.2±0.2 | 95.1±0.2 | 95.2±0.2 |
|  | 3C | 70.8±0.4 | **70.9±0.4** | 70.5±0.3 | 70.6±0.4 | 70.7±0.2 | 70.5±0.2 |
|  | 4C | 91.6±0.3 | 91.5±0.3 | 91.6±0.2 | 91.6±0.1 | **91.7±0.3** | 91.4±0.3 |
|  | 5C | **73.5±0.4** | 73.4±0.3 | 73.5±0.2 | **73.5±0.5** | 73.3±0.4 | 73.2±0.2 |
| 2010 | SP | **92.5±0.3** | 92.2±0.4 | 92.1±0.3 | 91.3±1.0 | 92.2±0.3 | 92.2±0.2 |
|  | DP | 92.4±0.4 | **92.5±0.3** | **92.5±0.3** | **92.5±0.3** | 92.4±0.4 | **92.5±0.4** |
|  | 3C | **67.2±0.3** | 67.0±0.4 | 67.0±0.5 | 67.0±0.4 | 67.0±0.3 | 67.1±0.5 |
|  | 4C | **89.4±0.4** | 89.2±0.4 | 89.0±0.2 | 88.9±0.4 | 89.2±0.4 | 89.3±0.3 |
|  | 5C | 69.8±0.2 | 69.8±0.3 | **70.0±0.4** | 69.9±0.3 | 69.7±0.4 | 69.9±0.3 |

**Table 29:** Link prediction test performance (accuracy in percentage) comparison for MSGNN with different $q$ values for individual years 2011-2020 of the *FiLL-OPCL* data set. The best is marked in **bold red** and the second best is marked in underline blue.

| Year | Link Task | $q=0$ | $q=0.2q_0$ | $q=0.4q_0$ | $q=0.6q_0$ | $q=0.8q_0$ | $q=q_0$ |
|---|---|---|---|---|---|---|---|
| 2011 | SP | 96.2±0.3 | 96.0±0.2 | 96.0±0.2 | 95.9±0.3 | 96.0±0.2 | 95.9±0.2 |
|  | DP | 96.3±0.2 | 96.4±0.3 | 96.3±0.2 | 96.3±0.3 | 96.3±0.2 | 96.3±0.3 |
|  | 3C | 79.0±0.2 | 78.8±0.4 | 79.1±0.3 | 79.0±0.3 | 79.0±0.2 | 78.6±0.4 |
|  | 4C | 95.4±0.2 | 95.4±0.3 | 95.5±0.2 | 95.5±0.3 | 95.4±0.2 | 95.5±0.3 |
|  | 5C | 80.9±0.4 | 81.0±0.6 | 80.8±0.3 | 80.8±0.3 | 80.7±0.6 | 80.6±0.3 |
| 2012 | SP | 91.1±0.2 | 91.0±0.4 | 90.8±0.3 | 90.8±0.4 | 90.8±0.5 | 90.8±0.5 |
|  | DP | 91.2±0.3 | 91.2±0.3 | 91.2±0.3 | 91.2±0.2 | 91.2±0.4 | 91.2±0.2 |
|  | 3C | 64.2±0.4 | 64.6±0.5 | 64.6±0.5 | 64.5±0.4 | 64.5±0.3 | 64.4±0.5 |
|  | 4C | 81.7±0.4 | 81.6±0.5 | 81.7±0.5 | 81.4±0.4 | 81.5±0.5 | 81.5±0.4 |
|  | 5C | 61.6±0.5 | 61.7±0.4 | 61.6±0.4 | 61.7±0.3 | 61.6±0.3 | 61.5±0.3 |
| 2013 | SP | 89.5±0.3 | 88.8±0.4 | 89.2±0.4 | 89.3±0.2 | 89.3±0.2 | 89.4±0.2 |
|  | DP | 89.4±0.3 | 89.4±0.1 | 89.3±0.2 | 89.4±0.2 | 89.4±0.2 | 89.3±0.2 |
|  | 3C | 64.3±0.4 | 64.4±0.2 | 64.4±0.1 | 64.2±0.2 | 64.4±0.3 | 64.3±0.3 |
|  | 4C | 83.3±0.3 | 83.4±0.1 | 83.4±0.3 | 83.4±0.4 | 83.5±0.1 | 82.8±0.8 |
|  | 5C | 64.2±0.3 | 64.1±0.2 | 64.0±0.3 | 64.1±0.4 | 64.0±0.3 | 63.9±0.4 |
| 2014 | SP | 87.7±0.4 | 87.0±0.6 | 87.5±0.5 | 87.7±0.4 | 87.4±0.5 | 87.5±0.3 |
|  | DP | 87.7±0.5 | 87.9±0.5 | 87.9±0.4 | 87.8±0.3 | 87.8±0.4 | 87.9±0.4 |
|  | 3C | 61.8±0.7 | 62.0±0.5 | 61.8±0.3 | 61.8±0.3 | 61.7±0.3 | 61.9±0.2 |
|  | 4C | 81.3±0.3 | 81.5±0.2 | 81.4±0.2 | 81.4±0.3 | 81.3±0.2 | 81.2±0.2 |
|  | 5C | 61.9±0.3 | 62.0±0.3 | 61.9±0.3 | 61.8±0.4 | 61.9±0.4 | 61.8±0.3 |
| 2015 | SP | 89.8±0.3 | 89.3±0.4 | 89.4±0.5 | 89.4±0.3 | 89.4±0.2 | 89.6±0.4 |
|  | DP | 89.9±0.3 | 89.7±0.3 | 89.6±0.5 | 89.7±0.4 | 89.7±0.4 | 89.8±0.3 |
|  | 3C | 64.1±0.4 | 64.0±0.4 | 64.3±0.2 | 64.2±0.2 | 64.2±0.3 | 64.1±0.2 |
|  | 4C | 84.7±0.3 | 84.6±0.4 | 84.8±0.2 | 84.8±0.1 | 84.7±0.2 | 84.8±0.2 |
|  | 5C | 64.9±0.4 | 64.8±0.6 | 64.8±0.5 | 64.6±0.7 | 64.8±0.5 | 64.8±0.5 |
| 2016 | SP | 89.0±0.2 | 88.7±0.2 | 88.7±0.3 | 88.8±0.5 | 88.5±0.8 | 88.7±0.3 |
|  | DP | 88.9±0.2 | 89.0±0.3 | 88.9±0.4 | 88.8±0.3 | 89.0±0.3 | 88.9±0.3 |
|  | 3C | 62.5±0.2 | 62.2±0.5 | 62.4±0.5 | 62.4±0.3 | 62.4±0.3 | 62.2±0.5 |
|  | 4C | 76.8±0.2 | 76.9±0.4 | 76.8±0.4 | 76.6±0.4 | 76.7±0.5 | 76.7±0.5 |
|  | 5C | 57.9±0.3 | 58.0±0.4 | 58.0±0.6 | 58.0±0.7 | 57.9±0.5 | 58.1±0.3 |
| 2017 | SP | 89.3±0.3 | 89.1±0.3 | 89.3±0.2 | 89.2±0.3 | 89.4±0.2 | 89.4±0.3 |
|  | DP | 89.3±0.3 | 89.4±0.2 | 89.4±0.4 | 89.4±0.2 | 89.4±0.1 | 89.5±0.2 |
|  | 3C | 61.3±0.3 | 61.3±0.3 | 61.2±0.4 | 61.4±0.4 | 61.3±0.3 | 61.3±0.2 |
|  | 4C | 69.5±0.5 | 69.5±0.5 | 69.4±0.5 | 69.3±0.7 | 69.3±0.6 | 69.4±0.4 |
|  | 5C | 52.0±0.4 | 51.8±0.5 | 51.8±0.4 | 51.6±0.5 | 51.9±0.6 | 51.8±0.3 |
| 2018 | SP | 88.2±0.4 | 87.6±0.6 | 87.8±0.6 | 87.8±0.6 | 87.9±0.4 | 87.7±0.6 |
|  | DP | 88.1±0.4 | 88.0±0.4 | 88.1±0.4 | 88.2±0.4 | 88.1±0.4 | 88.1±0.4 |
|  | 3C | 63.7±0.3 | 64.1±0.6 | 64.2±0.7 | 63.7±0.7 | 64.0±0.5 | 64.0±0.6 |
|  | 4C | 83.0±0.4 | 82.7±0.4 | 83.0±0.4 | 83.1±0.4 | 83.0±0.5 | 83.0±0.7 |
|  | 5C | 63.8±0.5 | 63.7±0.4 | 63.7±0.6 | 63.6±0.5 | 63.6±0.5 | 63.7±0.5 |
| 2019 | SP | 89.3±0.4 | 88.7±0.4 | 89.0±0.3 | 88.7±0.6 | 89.1±0.3 | 89.1±0.5 |
|  | DP | 89.4±0.2 | 89.3±0.4 | 89.4±0.2 | 89.2±0.3 | 89.2±0.1 | 89.3±0.2 |
|  | 3C | 62.2±0.2 | 62.4±0.5 | 62.3±0.7 | 62.2±0.6 | 62.4±0.3 | 62.4±0.5 |
|  | 4C | 74.5±0.2 | 74.5±0.4 | 74.1±0.3 | 74.4±0.2 | 74.2±0.3 | 74.3±0.4 |
|  | 5C | 56.0±0.3 | 55.9±0.3 | 56.1±0.4 | 56.1±0.4 | 55.8±0.3 | 56.0±0.2 |
| 2020 | SP | 92.3±0.1 | 91.8±0.1 | 91.5±0.3 | 91.7±0.2 | 91.6±0.1 | 91.5±0.1 |
|  | DP | 92.2±0.1 | 92.2±0.1 | 92.2±0.1 | 92.1±0.2 | 92.2±0.1 | 92.1±0.1 |
|  | 3C | 69.2±0.3 | 69.6±0.6 | 69.8±0.3 | 68.8±0.4 | 69.0±0.7 | 69.2±0.4 |
|  | 4C | 85.4±0.2 | 85.5±0.2 | 85.2±0.8 | 85.2±0.5 | 85.1±0.4 | 84.8±0.4 |
|  | 5C | 67.8±0.2 | 67.8±0.6 | 67.5±0.7 | 67.4±0.5 | 67.3±0.6 | 67.3±0.6 |

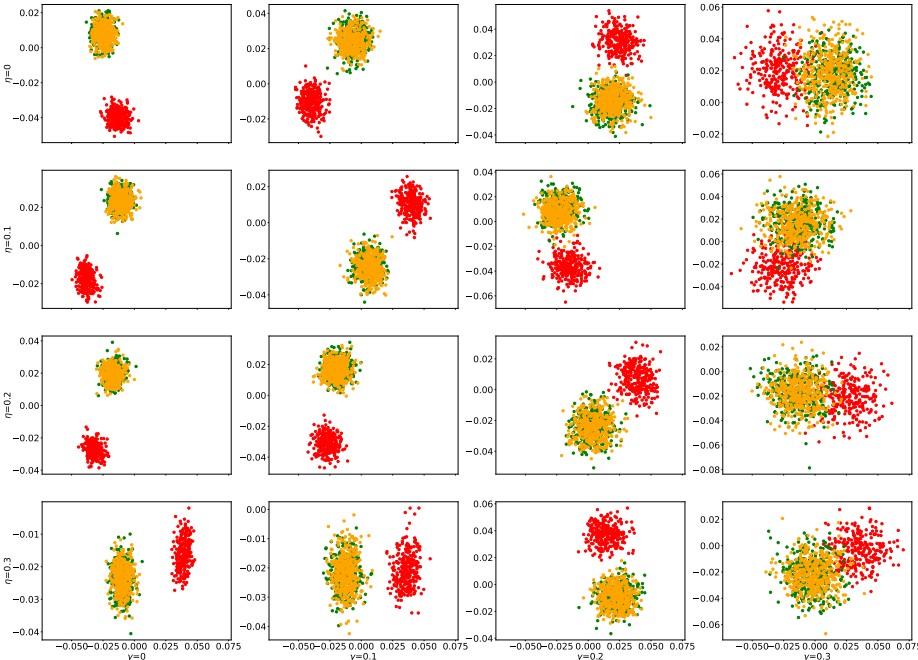

**Figure 3:** Real (x-axis) and imaginary parts (y-axis) of the top eigenvector of the symmetric-normalized magnetic signed Laplacian with $q = 0.25$ versus clusters labels on SDSBM($\mathbf{F}_1(\gamma), n = 1000$,
$p = 0.1, \rho = 1.5, \eta$) with various $\gamma$ and $\eta$ values, where red, green, and orange denote $\mathcal{C}_0, \mathcal{C}_1$, and $\mathcal{C}_2$, respectively.

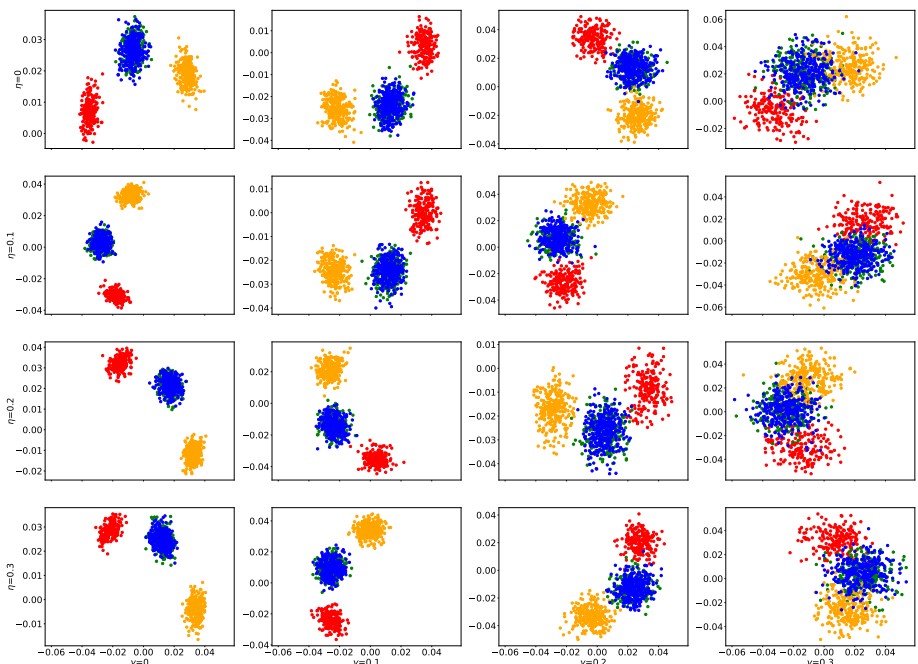

**Figure 4:** Real (x-axis) and imaginary parts (y-axis) of the top eigenvector of the symmetric-normalized magnetic signed Laplacian with $q = 0.25$ versus clusters labels on SDSBM($\mathbf{F}_2(\gamma), n = 1000$,

$p = 0.1, \rho = 1.5, \eta$) with various $\gamma$ and $\eta$ values, where red, green, orange, and blue denote $\mathcal{C}_0, \mathcal{C}_1, \mathcal{C}_2$, and $\mathcal{C}_3$, respectively.

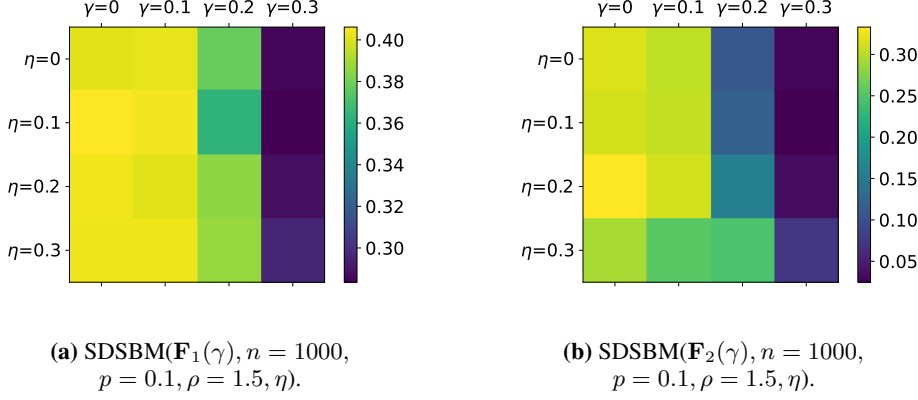

**(a)** SDSBM($\mathbf{F}_1(\gamma), n = 1000$,
$p = 0.1, \rho = 1.5, \eta$).

**(b)** SDSBM($\mathbf{F}_2(\gamma), n = 1000$,
$p = 0.1, \rho = 1.5, \eta$).

**Figure 5:** ARI using the proposed magnetic signed Laplacian's top eigenvectors and K-means.

