# OpenReview forum: "MSGNN: A Spectral Graph Neural Network Based on a Novel Magnetic Signed Laplacian"
_logconference.io/LOG/2022/Conference — LoG 2022 Poster_

### Official Review · Reviewer_WE2B · 2022-10-20

**Overall Score:** 8
**Confidence:** 4

**Review:**

The paper addresses machine learning tasks on signed, directed networks by proposing a spectral graph neural network. The model design is based on a generalisation of the graph Laplacian to signed and directed graphs. The proposed Laplacian matrix has many desirable properties, including that it reduces to the original graph Laplacian for unsigned, undirected networks. The neural network design follows existing spectral graph neural networks but using the proposed graph Laplacian. Experiments on node clustering and link prediction show competitive performance against strong baselines.

Pros:
* The paper is very well written and easy to follow. It should be easy to understand even for people with little background in graph signal processing as it is fairly self-contained. Overall, it is an enjoyable read.
* The experiments are extensive and in particular including experiments on synthetic data and ablation studies is commendable.

Cons:
* A minor weakness of the work is its limited novelty as the main difference to existing neural network architectures like ChebNet is the use of the modified Laplacian, which is itself similar to previous works. Nonetheless, the paper proposes a general method for signed and directed graphs and has a thorough evaluation, which makes it a useful read for the community.

Suggestions:
* Given the paper proposes a spectral rather than spatial GNN, it would be interesting to study some of the properties of the proposed Laplacian in more detail (even if this has to be moved to the Appendix). In particular, the following questions are of interest:
	* The original graph Laplacian for undirected, unsigned graphs effectively computes the second derivative of a graph signal. What can be said about the expression Lx where x is a graph signal and L is the proposed Laplacian?
	* What do the eigenvalues and eigenvectors of the proposed graph Laplacian look like? Can eigenvalues be interpreted as frequencies, meaning corresponding eigenvectors show different levels of smoothness? Can we observe clusters in the eigenvalues of the proposed Laplacian and if so, do they have a clear relationship with the structure of the graph?

I believe discussing the above questions could significantly improve the work. I would recommend a score of 7/10 and would be happy to raise my score if the authors address the above suggestions.

Update after rebuttal: The authors have addressed my concerns and I am happy to raise my score to 8, clearly recommending acceptance.

---

### Official Review · Reviewer_xPuh · 2022-10-21

**Overall Score:** 8
**Confidence:** 5

**Review:**

(1) The article studies introduces a new version of the Laplacian operator which is adapted to the cases of directed and signed graphs. After a theoretical study of its properties, the authors show how to use it for convolutions on graphs (using the now classical spectral equivalent) before plugging that un a Graph Neural Network architecture based on graph convolutions. Then, a long and detailed study of numerical experiments are reported that show the benefits  of considering this signed magnetic Laplacian for learning task with conv. GNN. For these experiments, the authors introduce, as a additional contribution, a novel SBM-based model for signed and directed model of random networks with groups. Globally, the authors show that the proposed method behaves well, especially that it improves on existing methods when the features of sign and/or directions are important for the studied data, for two tasks: node clustering and link prediction.


(2) This proposed article is strong. The proposed Magnetic signed Laplacian is a novel (albeit built on ideas already considered for signed graphs or directed graphs, usually independently) and its is correctly used in a convolutional GNN framework.

Strong points:
- a novel and relevant  Magnetic signed Laplacian
- a good analysis of its properties and of its difference w.r.t. the existing (and concurrent) SigMaNet (Fiorini et al., 2022, arXiv:2205.13459), including some weaknesses in the Laplacian proposed in the latter.
- an interesting Signed Directed SBM which complements the existing benchmarks
- a robust and trustable numerical study, including both synthetic and real-world data, with comparison to SOTA and ablation study.

Weak point:
- I find mostly one weak point in the presentation (see also additional suggestions underneath) :
the discussion of the present work relative to  SigMaNet at the end of Section 2 looks not adequately placed. In this Section 2 about related work, the fact that there is a concurrent solution arrives late = it should be stated in the Introduction already ; then the technical details about     the differences between SigMaNet and the proposed MSL are awkward to be discussed there because the reader has not already read about the proposition that comes after in 3.2.
My major feedback is that the paragraphs from lines 105 to 125 should be split in 3 places : i) announce the work of  SigMaNet in the introduction to say that a comparison is necessary ; ii) keep some non-technical element in 2 to discuss it a related world ; ii) devote a full discussion after 3.2 to discuss the technical differences between the proposed MSLaplacian and the one from SigMaNet. Maybe this could be one in a novel 3.3, between the current subsections, as, basically, the rest is of the same flavor in the two works (the present work uses a ConvNN of Defferrard and al. while SigMaNet builds on the simplified version GCN of Kipf&Welling, but this does not not really change much).

(4) Given all the strong points and the fact that the weak point i easy to revise, I strongly suggest acceptation.

(5) and (6): some questions or secondary suggestions:

- page 4: the choice of the charge parameter $q$ could be more discussed upfront or with more insight. Is there a way to estimate it ? Set it according to typical (or maximal) values of $A_{ij} - A_{ji}$. What type of pre-processing normalization would be adapted to set $q$ ?
- in 3.3: I think the authors should refer to [39] then [38] before [13, 36] in lines 168-169 to say that they used "similar methods" to existing ones. Basically, the spectral convolution is the one coming from [39], then proposed for GNN in [38].
- lines 180-184: the comments missed a little bit the point : the purpose of using a graph filters is both to reduce the number of parameters and to have an efficient spatial implementation still coherent with the spectral definition.
- about the financial data in 4.3:  the proposed dataset is interesting. Still, I think that the reference to [4] at the end is unnecessary. One would more expect a general reference about inference of dependence in financial data (be it by Granger causality, directed entropy, this method of [4] or whatever). If the authors want to tell that they are more advanced ways to evaluate directed dependences, it's fine ; however one expects a general reference on that and not a specific work (even if the work is interesting, it is not relevant on the  whole to the present article).
- Figure 1 is barely legible. Please consider to use larger markers and lines

(7) this is a 9 page article which deserves to be published in the 9 page track.

---

### Official Review · Reviewer_idTr · 2022-10-21

**Overall Score:** 6
**Confidence:** 3

**Review:**

Summary:

The authors present an extension to spectral GNNs in order to support undirected and signed edge weights. The method is based on the Magnetic Signed Laplacian that is designed to fulfill the requirements from a proper convolution operator. The authors provide a theoretical analysis of their proposition followed by extensive experiments and ablations studies, often times showing improvement over existing methods.

Strong points:

1. The authors provide a solid background of spectral GNNs and the recent effort in enabling the usage of signed and directed graph using spectral GNNs. Especially, I liked the comparison to the recent pre-print SigMaNet which addresses the very same problem although using a slightly different mechanism.

2.The authors provide a solid theoretical analysis of the method.

3. The authors conduct sufficient amount of experiments and ablation studies in the Appendix. The experiments read improvements over existing methods in many of the considered datasets.

Weak points:

1. The authors propose both symmetrized and non-symmetrized Magnetic Signed Laplacians. However, from the text it was not clear to me which one is used when, and I think that an ablation study of the choice of normalization is important here.

2. The authors only consider two layers of their MSGNN. Given the well-known oversmoothing phenomenon, I would like to see experiments with various number of layers.

3.Given the large and impressive number of experiments in the ablation study, I think it would be useful if the authors can also state how long it took to run the models on the specific datasets (that is besides the already reported single run time of the model)

4.Figure 1 can be improved both from a clarity standpoint and print quality.

5. It is not clear to me from the text how many channels (features) are utilized in the proposed network compared to other methods, and what is the number of required parameters.

Recommendation:

I recommend accepting the paper given the clear writing and direct comparison with existing methods and current pre-prints, and also based on the experimental results and theoretical discussion.

Question to the authors: It seems that in most settings, your method outperforms the rest of the methods. However on the FiLL dataset it seems that your method is consistently outperformed by others. Can you please discuss why?

---

### Meta-Review · Area_Chair_T6nH · 2022-11-21

**Confidence:** 5
**Recommendation:** Accept

**Meta Review:**

Reviewers are enthusiastic about this submission. Discussion among PC members showed a strong consensus that:
* S1: the new approach describes a novel and relevant magnetic signed Laplacian,
* S2: theoretical analyses are solid and empirical experiments extensive,
* S3: the paper is easy to follow and understand.

Minor questions and points of clarification were raised during the review, for which authors provided an extensive rebuttal with additional insights and experiments.

---

### Decision · Program_Chairs · 2022-11-22

Accept (Poster)